# An artificial protein modulator reprogramming neuronal protein functions

Peihua Lin [1,2,9], Bo Zhang [1,3,9], Hongli Yang [2], Shengfei Yang[2], Pengpeng Xue[2], Ying Chen[2], Shiyi Yu [2], Jichao Zhang[4], Yixiao Zhang[5], Liwei Chen[5,6], Chunhai Fan [1], Fangyuan Li [2,7,8] ✉ & Daishun Ling [1,3] ✉

Reversible protein phosphorylation, regulated by protein phosphatases, fine-tunes target protein function and plays a vital role in biological processes. Dysregulation of this process leads to aberrant post-translational modifications (PTMs) and contributes to disease development. Despite the widespread use of artificial catalysts as enzyme mimetics, their direct modulation of proteins remains largely unexplored. To address this gap and enable the reversal of aberrant PTMs for disease therapy, we present the development of artificial protein modulators (APROMs). Through atomic-level engineering of heterogeneous catalysts with asymmetric catalytic centers, these modulators bear structural similarities to protein phosphatases and exhibit remarkable ability to destabilize the bridging $\mu_3$-hydroxide. This activation of catalytic centers enables spontaneous hydrolysis of phospho-substrates, providing precise control over PTMs. Notably, APROMs, with protein phosphatase-like characteristics, catalytically reprogram the biological function of α-synuclein by directly hydrolyzing hyperphosphorylated α-synuclein. Consequently, synaptic function is reinforced in Parkinson's disease. Our findings offer a promising avenue for reprogramming protein function through de novo PTMs strategy.

Proteins execute a wide variety of biological functions ranging from catalysts to structural basis and signaling messengers in living systems[1]. Among these processes, reversible protein phosphorylation stands out as a major form of post-translational modification (PTM) that orchestrates protein function[2,3]. Protein phosphatases, working in collaboration with kinases, precisely regulate this dynamic phosphorylation-dephosphorylation cycle[4,5]. However, in the pathological microenvironment, the catalytic activity of protein phosphatases becomes compromised[6,7], leading to irreversible phosphorylation of neuronal proteins, including α-synuclein (α-syn)[8] and tau protein[9]. This dysregulation ultimately disrupts the structure and biological function of these neuronal proteins, severely impairing synaptic function and contributing to the onset and progression of neurodegenerative diseases[10–12]. Despite efforts to recruit protein phosphatases for protein dephosphorylation[13] and to disentangle[14] or eliminate[15,16] hyperphosphorylated protein aggregates, the imbalance in the kinase and protein phosphatase system[6,7] persists as an inevitable challenge, irreversibly exacerbating aberrant PTMs of neuronal proteins.

[1]Frontiers Science Center for Transformative Molecules, School of Chemistry and Chemical Engineering, School of Biomedical Engineering, National Center for Translational Medicine, State Key Laboratory of Oncogenes and Related Genes, Shanghai Jiao Tong University, Shanghai 200240, China. [2]Institute of Pharmaceutics, College of Pharmaceutical Sciences, Zhejiang University, Hangzhou 310058, China. [3]World Laureates Association (WLA) Laboratories, Shanghai 201210, China. [4]Shanghai Synchrotron Radiation Facility, Shanghai Advanced Research Institute, Chinese Academy of Sciences, Shanghai 201204, China. [5]In-situ Center for Physical Sciences, School of Chemistry and Chemical Engineering, Shanghai Electrochemical Energy Device Research Center (SEED), Shanghai Jiao Tong University, Shanghai 200240, China. [6]Future Battery Research Center, Global Institute of Future Technology, Shanghai Jiao Tong University, Shanghai 200240, China. [7]Songjiang Research Institute, Songjiang Hospital, Shanghai Jiao Tong University School of Medicine, Shanghai 201600, China. [8]Key Laboratory of Precision Diagnosis and Treatment for Hepatobiliary and Pancreatic Tumor of Zhejiang Province, Hangzhou 310009, China. [9]These authors contributed equally: Peihua Lin, Bo Zhang. ✉e-mail: lfy@shsmu.edu.cn; dsling@sjtu.edu.cn

Artificial catalysts with enzymatic activities[17–19] hold great potential as alternative solutions to modulate proteins by compensating for compromised endogenous catalytic activities[20–22]. However, their efficacy as substitutes for natural enzymes is hindered by the lack of highly efficient catalytic centers[23], particularly in the context of phosphatase mimetics. A critical aspect contributing to the exceptional catalytic performance of natural enzymes is the presence of the asymmetric unit at catalytic centers[24–26]. While small-molecule asymmetric catalysts can selectively generate favorable intermediates for efficient catalytic reactions[27,28], their practical biomedical applications are hindered by inherent challenges associated with small molecules[29,30], such as off-target side effects[31] and untargeted biodistribution[29,32]. In this regard, nanomaterials based heterogeneous catalysts with asymmetric catalytic centers show promise in overcoming the inherent limitations of small-molecule asymmetric catalysts due to their modifiable surface[29,33]. Although progress has been made in this area, the rational design of heterogeneous catalysts with natural enzyme-like asymmetric catalytic centers remains elusive, primarily due to the difficulty in precisely doping the catalytic centers with heterogeneous metals at the atomic level.

Here, we report the development of the artificial protein modulator (APROM) that exhibits protein phosphatase-like characteristics through atomic-level engineering of heterogeneous catalysts with asymmetric catalytic centers (Fig. 1). Our strategy involves precisely incorporating site-specific single manganese atoms into heterogeneous ceria nanoparticular catalysts ($Mn/CeO_{2-x}$) through a self-aggravating surface oxygen vacancy (SOV)-driven cation exchange reaction. The resulting APROM

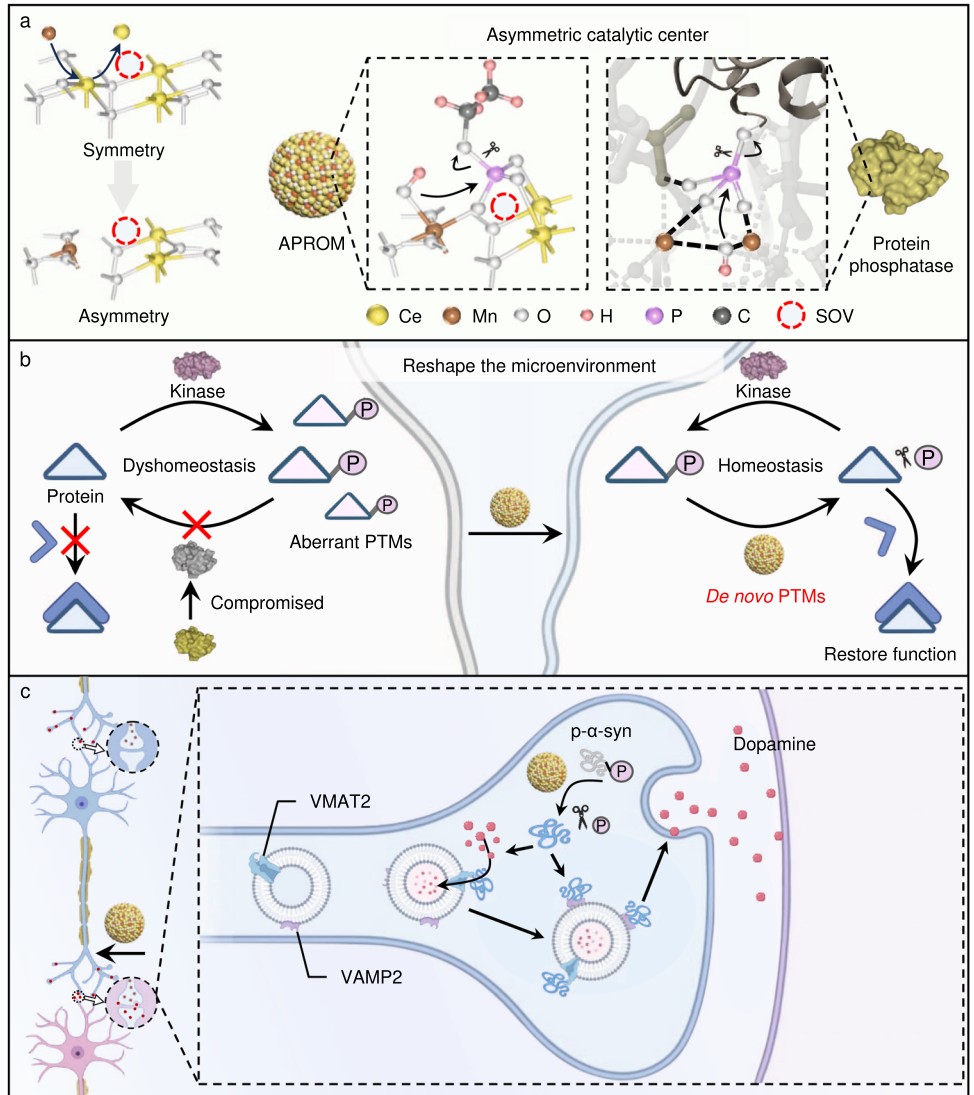

**Fig. 1 | Designing the artificial protein modulator (APROM) through atomic-level engineering of heterogeneous catalysts with asymmetric catalytic centers to reprogram neuronal protein functions via the de novo post-translational modification (PTM) strategy. a** The scheme illustrates the unique features of the APROM, which possesses asymmetric catalytic centers similar to natural protein phosphatases through precisely incorporating site-specific single manganese atoms into heterogeneous ceria nanoparticular catalysts. These catalytic centers facilitate the conversion of the asymmetric bridging $\mu_3$-hydroxide into Mn-bonded OH, resulting in the activation of active catalytic centers capable of binding phospho-substrates and initiating spontaneous hydrolysis. **b** The APROM, characterized by protein phosphatase-like characteristics, plays a vital role in reversing aberrant PTM processes by compensating for compromised protein phosphatases through de novo PTMs. This mechanism ultimately leads to the restoration of protein function. **c** Notably, APROM₂ directly modulates phosphorylated α-syn (p-α-syn) by cleaving the phosphate monoester bond, enabling α-syn to regain its biological functions of binding to vesicular monoamine transporter 2 (VMAT2) and vesicle-associated membrane protein 2 (VAMP2). This restoration of α-syn function catalytically fuels synaptic activity, thereby contributing to the enhancement of synaptic plasticity in PD. SOV, surface oxygen vacancy. Structure of protein phosphatase and kinase is from PDB ID 1S95 and 4RS6, respectively. Panels **a**–**c** were created with BioRender.com.

possesses asymmetric catalytic centers comprising Mn, Ce, and SOVs, which exhibit a unique steric effect that activates the catalytic centers for spontaneous hydrolysis of phospho-substrates, thereby reversing aberrant PTMs of neuronal proteins. Notably, the APROM demonstrates protein phosphatase-like behavior and has the ability to restore the biological function of phosphorylated α-syn (p-α-syn) through de novo PTMs, effectively fueling synaptic function. Moreover, the APROM exhibits antioxidant activity, enabling the restoration of mitochondrial function to maintain energy homeostasis which is crucial for synaptic activity. These findings highlight the potential for developing asymmetric heterogeneous catalysts as artificial protein modulators, and provide a promising platform for reprogramming neuronal protein functions.

## Results

### Synthesis and characterization of APROMs with asymmetric catalytic centers

To synthesize the APROM with asymmetric catalytic centers, we developed a facile self-aggravating SOV-driven cation exchange strategy (Fig. 2a), resulting in tunable Mn-to-Ce ratios denoted as $APROM_1$, $APROM_2$, and $APROM_3$ (Supplementary Fig. 1). Transmission electron microscopy (TEM) and high-resolution TEM (HRTEM) images confirm the successful synthesis of ultrafine APROMs (Fig. 2b). For comparison purposes, we also synthesized ceria nanoparticles (CeNPs) with symmetric catalytic centers using a previously reported method[34] (Fig. 2b). X-ray diffraction (XRD) patterns of APROMs closely match those of CeNPs, without characteristic peaks corresponding to Mn detected (Supplementary Fig. 2). To investigate the elemental composition and chemical state of APROMs, we performed X-ray photoelectron spectroscopy (XPS) analysis. The high-resolution XPS spectra of Mn $2p$, Ce $3d$, and O $1s$ demonstrate the presence of Mn, Ce, and O elements in APROMs (Fig. 2c–e), consistent with the energy-dispersive X-ray spectroscopy (EDS) mapping results (Supplementary Fig. 3), confirming the successful doping of Mn into the ceria lattice. Importantly, we observed an increase in the number of SOVs, considered as catalytic centers of APROMs[35], accompanied by an increase in the $Ce^{3+}$-to-$Ce^{4+}$ ratio. Notably, $APROM_2$, with a Ce-to-Mn ratio of approximately 3, exhibits the highest abundance of SOVs and the highest ratio of $Ce^{3+}$-to-$Ce^{4+}$ (Fig. 2d, e, Supplementary Table 1). To further investigate the local coordination environment of APROMs, we conducted structure-sensitive X-ray absorption fine structure (XAFS) measurements. Fourier-transform extended XAFS (FT-EXAFS) spectra at the Ce-L3-edge display peaks at approximately 1.8 and 3.6 Å, corresponding to Ce-O and Ce-M (M = Ce or Mn) bonds, respectively (Fig. 2f). Notably, the amplitudes of the second Ce-M shell of APROMs are reduced due to the disordering effect[36] induced by Mn substitution, confirming the formation of asymmetric catalytic centers. FT-EXAFS spectra at the Mn-K-edge reveal an intense first Mn-O coordination shell for APROMs, while the characteristic metallic Mn-Mn interaction at approximately 2.4 Å is not observed (Fig. 2g). Moreover, wavelet transform (WT) analysis of Mn-K-edge EXAFS oscillations further confirms the atomically dispersed nature of Mn species in APROMs (Fig. 2h–l). To gain further insights into the enhanced generation of asymmetric catalytic centers, density functional theory (DFT) calculations were performed. Three models, including $CeO_2$, $Ce_{0.75}Mn_{0.25}O_2$, and $Ce_{0.5}Mn_{0.5}O_2$, were constructed to simulate the formation of SOVs. The calculated SOV formation energy ($\Delta E_{SOV}$) of $Ce_{0.75}Mn_{0.25}O_2$ decreases to 1.95 eV, while that of $Ce_{0.5}Mn_{0.5}O_2$ increases to 3.93 eV, as compared to $CeO_2$ (3.90 eV) (Fig. 2m). Moreover, the bond length calculation reveals that the Mn-O bond (-2.105 Å) is relatively shorter than the Ce-O bond (-2.399 Å) in $Ce_{0.75}Mn_{0.25}O_2$, indicating the migration of adjacent oxygen towards Mn. As a result, the interaction between oxygen and Ce is weakened, making oxygen susceptible to detach from the fluorite lattice. However, with increased Mn substitution, the unit cell of

$Ce_{0.5}Mn_{0.5}O_2$ contracts (Fig. 2b, Supplementary Fig. 2b), allowing for stable bonding between oxygen and Ce as well as Mn. Consequently, catalytic centers favor the substitution of only one Ce in the $CeO_2$ unit cell, as supported by both experimental and theoretical results. Charge density diagrams reveal that the charge of oxygen inclined towards adjacent Mn in the $Ce_{0.75}Mn_{0.25}O_{2-x}$ model, resulting in the reduction of Ce and an uneven distribution of electrons in asymmetric catalytic centers (Fig. 2n, o, Supplementary Fig. 4). Collectively, these findings indicate that appropriate Mn substitution facilitates structural reconstruction for the generation of asymmetric catalytic centers.

### The evolution of asymmetric catalytic centers via a self-aggravating SOV-driven cation exchange strategy

Control experiments using $APROM_2$ as a model were conducted to provide detailed insights into the atomic-scale evolution process. Initially, $CeO_{2-x}$ nanocrystals with a small amount of symmetric catalytic centers are formed, as observed in Supplementary Fig. 5. The driving forces from the composing SOVs[37] enable preferential absorption of Mn at the as-constructed symmetric catalytic centers. Subsequently, the much larger radius of SOVs (1.38 Å) compared to Mn (ionic radii of $Mn^{2+}$, $Mn^{3+}$ and $Mn^{4+}$ are 0.83 Å, 0.65 Å and 0.53 Å, respectively) and Ce (ionic radii of $Ce^{3+}$ and $Ce^{4+}$ are 1.14 Å and 0.97 Å, respectively) allows SOVs to act as sub-nano reaction containers in the cation exchange reaction, where Mn substitutes surrounding Ce. Moreover, SOVs play a crucial role as excellent transporters of Mn through the ceria lattice, resulting in the uniform dispersion of Mn within the heterogeneous structure and successful formation of asymmetric catalytic centers (Fig. 2a, p–r, Supplementary Figs. 6 and 7). This mechanism prevents the formation of homogeneous, core-shell, or Janus-like structures[38–40]. Importantly, Mn doping facilitates SOV formation by decreasing $\Delta E_{SOV}$ (Fig. 1m), further establishing a positive feedback loop of cation exchange mediated by SOVs. High-angle annular dark-field scanning transmission electron microscopy (HAADF-STEM) images reveal that Mn preferentially substitutes Ce around the SOVs, confirming the formation of asymmetric catalytic centers (Fig. 2p, q, Supplementary Figs. 6 and 7). This observation strongly suggests that the presence of SOVs drives the cation exchange reaction. Consequently, $APROM_2$ with a molar ratio of approximately 25% Mn is successfully prepared, as quantified by inductively coupled plasma mass spectrometry (ICP-MS) (Fig. 2s). As expected, during the reconstruction of catalytic centers, only cubic fluorite structures are formed without any invisible Mn-related phases (Supplementary Fig. 8). Additionally, based on the positive correlation between luminance and atomic number, the dark points representing low-atomic-number Mn indicate that Mn is monatomically dispersed in the ceria lattice without forming aggregates (Fig. 2q, Supplementary Fig. 7).

In order to gain a clear insight into the evolution of asymmetric catalytic centers at the atomic level, DFT calculations were performed. The calculation results reveal that Mn is more accessible to the subsurface of the nanoparticle, particularly at SOV sites, compared to Ce, due to a lower energy barrier ($\Delta E_{mo}$) (Fig. 2t, u, Supplementary Table 2). Importantly, the energetically favored Mn is the one adjacent to the SOV, as evidenced by its lower formation energy ($E_f$) compared to the non-SOV-adjacent Mn (Fig. 2v). These findings are consistent with the observations from HAADF-STEM images, indicating that Mn preferentially exchanges with Ce around SOV sites to form asymmetric catalytic centers in terms of kinetics. Moreover, the calculations show that the doping of the second Mn into the Mn/$CeO_{2-x}$ unit cell requires a larger $E_f$ compared to doping into other unit cells, suggesting uniform Mn substitution. Additionally, the dissociation of lattice oxygen species into SOVs is facilitated by Mn substitution, promoting cation exchange reactions through a self-aggravating SOV-driven mechanism. Subsequently, Mn dopants competitively migrate inward due to a significantly lower migration

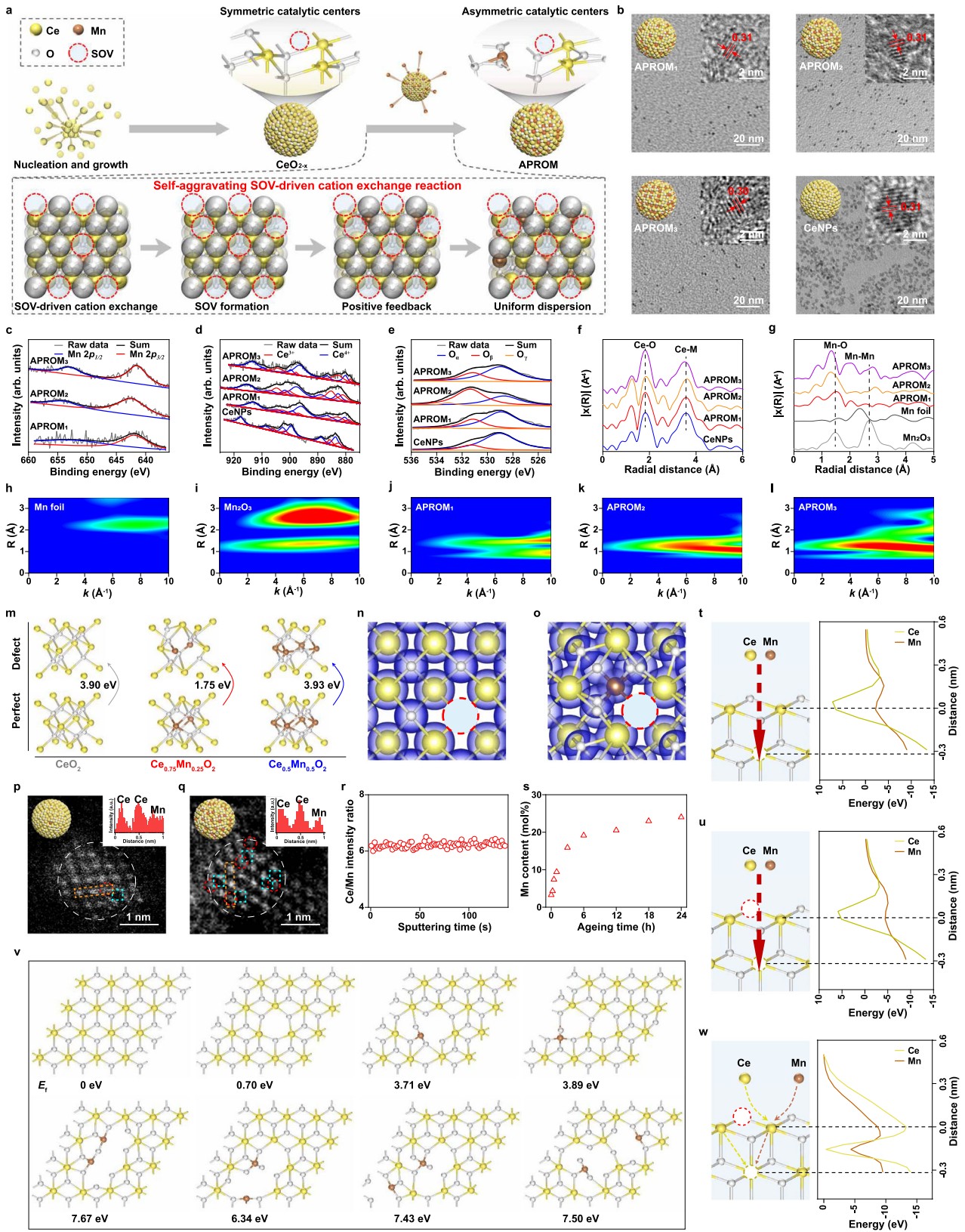

energy barrier ($\Delta E_{mi}$) compared to Ce (Fig. 2w, Supplementary Table 2), resulting in the uniform distribution of Mn dopants. Ultimately, the precise incorporation of site-specific Mn into the ceria lattice leads to the formation of asymmetric catalytic centers. Notably, the self-aggravating SOV-driven cation exchange strategy exhibits unusual phenomena, including a slow exchange rate, an SOV-mediated positive feedback loop of cation exchange, and the atomic-level uniform distribution of guest cations, which enable the controllable construction of asymmetric catalytic centers, distinguishing itself from traditional cation exchange methods[38–41].

**Fig. 2 | Synthesis, and characterization of APROMs, as well as the evolution of asymmetric catalytic centers in APROMs. a** Schematic illustration of the synthesis of the APROM with asymmetric catalytic centers through the self-aggravating SOV-driven cation exchange strategy. **b** TEM image of APROMs and CeNPs. Inset, schematic illustration (left) and HRTEM (right). High-resolution XPS spectra of Mn 2$p$ peaks (**c**), Ce 3$d$ peaks (**d**) and O 1$s$ peaks (**e**) of APROMs and CeNPs. Peaks of $O_\alpha$ correspond to lattice oxygen species, peaks of $O_\beta$ correspond to surface oxygen species derived from defective sites, and peaks of $O_\gamma$ correspond to chemisorbed water and carbonates. **f** Fourier-transform Ce-L3-edge EXAFS spectra of APROMs and CeNPs. **g** Fourier-transform Mn-K-edge EXAFS spectra of Mn foil, $Mn_2O_3$ and APROMs. WT images of the Mn-K-edge of Mn foil (**h**), $Mn_2O_3$ (**i**), and APROMs (**j–l**). **m** Unit cell configurations and corresponding $\Delta E_{SOV}$ of $CeO_2$, $Ce_{0.75}Mn_{0.25}O_2$, and $Ce_{0.5}Mn_{0.5}O_2$. Simulated charge-density isosurface plots of $CeO_{2-x}$ (**n**) and $Ce_{0.75}Mn_{0.25}O_{2-x}$ (**o**). The red dashed circle represents SOVs, and the blue area represents a charge density of 0.05 e/bohr. Atomic-resolution HAADF-STEM images of APROM$_2$ obtained after aging for 15 min (**p**) and 24 h (**q**). Inset, schematic illustration of nanoparticles (left) and line intensity profile in the orange rectangle highlighting the positions of single Ce atom and Mn atom (right). The red dashed circle represents Mn, and the cyan dashed square represents SOVs. **r** Time-of-flight secondary ion mass spectrometry depth profile of APROM$_2$, showing the in-depth distribution of the Ce-to-Mn ratio. **s** The Mn content of APROM$_2$ at different aging times. Schematic illustration and computational energy of the movement of Ce and Mn toward the subsurface of $CeO_2$ (**t**) and $CeO_{2-x}$ (**u**). **v** Structural configurations and corresponding $E_f$ of $CeO_2$, $CeO_{2-x}$, and different $Mn/CeO_{2-x}$. **w** Schematic illustration and corresponding computational energy of self-aggravating SOV-driven cation exchange in $CeO_{2-x}$ unit cell. Source data are provided as a Source Data file.

## Asymmetric catalytic centers endow APROMs with protein phosphatase-like characteristics

The well-defined asymmetric catalytic centers in the atomic-level engineered APROMs motivate us to investigate relevant protein phosphatase-like properties. In CeNPs, the OH adsorbed on the SOV acts as a stable bridging $\mu_3$-hydroxide due to the high symmetry of the catalytic center (Fig. 3a). In contrast, the smaller Mn ionic radius and Mn-O bond length in the asymmetric catalytic center of the APROM allow the bridging $\mu_3$-hydroxide to lean towards Mn, thereby destabilizing the OH-Ce interaction (Fig. 3b). Additionally, the asymmetric catalytic center exhibits a stronger electron-induced effect on the O atom of OH, leading to the easy breakage of Ce-O bonds, as evidenced by differential charge density maps of OH absorbed CeNPs and APROM (Fig. 3a, b). The formed bridging $\mu_3$-hydroxide further influences the subsequent hydrolysis of phospho-substrates that contain phosphate monoester bonds. Specifically, steric hindrance induced by the stable bridging $\mu_3$-hydroxide in CeNPs requires additional energy (0.19 eV) for phospho-substrate adsorption (Fig. 3c). In contrast, when the phospho-substrate is bound to the surface of the APROM, the bridging $\mu_3$-hydroxide is quickly replaced by the oxygen of the phosphate group, which is then transferred to the Mn site in the APROM. This is supported by the reaction energy of −1.03 eV, indicating that the APROM can efficiently compete in the spontaneous hydrolysis of phospho-substrates. Differential charge density analysis further explores the $SN_2$ reaction, revealing that the R-O bond of the phospho-substrate is more vulnerable in the APROM compared to CeNPs (Fig. 3d, e). Moreover, the Bader charge of the O atom of OH absorbed in the APROM increases after phospho-substrate absorption, facilitating the nucleophilic attack (Fig. 3f). Additionally, electronic structure analysis based on the density of states (DOS) shows that CeNPs possess a wide band gap of approximately 0.8 eV, which hinders electron conduction (Fig. 3g). However, Mn substitution significantly reduces the band gap in the APROM, enhancing the electron transfer capacity and resulting in the crossing of DOS over the Fermi level. Further analysis of the projected density of states (PDOS) reveals that the DOS near the Fermi level on the surface of the APROM is mainly contributed by the d orbital of Mn (Fig. 3h, i). Overall, thanks to the presence of asymmetric catalytic centers, the APROM exhibits strong catalytic activity in phospho-substrate hydrolysis, resembling the function of protein phosphatases. To further investigate the protein phosphatase-like behavior of APROMs, the as-synthesized APROMs were modified with citric acid via a ligand exchange method, followed by PEGylation, resulting in water-dispersible and stable APROMs (Supplementary Fig. 9). As a proof-of-concept, the protein phosphatase-mimetic activity of APROMs was tested using O-phospho-L-serine (P-Ser) as the phospho-substrate, as phosphorylation at Ser-129 is a predominant pathological PTM of p-α-syn[42]. Comparative analysis reveals that APROMs exhibit superior protein phosphatase-mimetic activity (Fig. 3j, Supplementary Fig. 10), with APROM$_2$, containing the highest number of asymmetric catalytic centers, exhibiting the most potent

protein phosphatase-mimetic activity. According to the Michaelis–Menten plot and Lineweaver–Burk plot, the Michaelis–Menten constant ($K_m$), maximum velocity ($V_{max}$) and turnover number ($k_{cat}$) of APROM$_2$ are 39.17 mM, 0.49 μM s$^{-1}$, and $2.47 \times 10^2$ s$^{-1}$, respectively, indicating APROM$_2$ can effectively dephosphorylate P-Ser (Supplementary Fig. 11, Supplementary Table 3). Raman spectra further confirm that the dephosphorylation of P-Ser occurred on asymmetric catalytic centers (Fig. 3k), validating the enhancement of catalytic performance by asymmetric catalytic centers. Asymmetric catalytic centers of APROM$_2$ exhibit exceptional stability, as evidenced by the constant Ce-to-Mn molar ratio and fluorite lattice throughout the catalytic reaction (Fig. 3l, m), indicating that APROM$_2$ can sustainably and efficiently dephosphorylate the phospho-substrate. Inspired by the remarkable dephosphorylation capability of APROM$_2$, we further investigated its protein phosphatase-mimetic activity using p-α-syn as the phospho-substrate. As expected, APROM$_2$ readily dephosphorylates p-α-syn in a concentration-dependent manner (Fig. 3n), and the $K_m$, $V_{max}$, $k_{cat}$ of APROM$_2$ are 0.175 mM, 0.0015 μM s$^{-1}$, and 0.764 s$^{-1}$ (Fig. 3o, p, Supplementary Table 3), respectively. This suggests that APROM$_2$ can efficiently dephosphorylate p-α-syn, converting it into α-syn through de novo PTMs. Moreover, APROM$_2$ exhibits pronounced dephosphorylation activity towards O-phospho-L-tyrosine (P-Tyr) and O-phospho-L-threonine (P-Thr), effectively cleaving the phosphate monoester bonds (Fig. 3q). This collective evidence solidifies the notion that APROM$_2$ faithfully emulates the functionality of protein phosphatases, encompassing both protein serine/threonine phosphatases and protein tyrosine phosphatases.

## Asymmetric catalytic centers endow APROMs with exceptional antioxidant activity

Ascribed from the reversible $Ce^{3+}/Ce^{4+}$ redox pair in asymmetric catalytic centers[43] (Fig. 2d), APROMs simultaneously possess antioxidant properties, including catalase (CAT)-mimetic and superoxide dismutase (SOD)-mimetic activities, corresponding to the multi-enzymatic activities achieved by artificial catalysts instead of natural enzymes so as to exert synergistic effect[21]. Remarkably, the SOD-mimetic activity of APROMs is enhanced, with APROM$_2$ performing best in scavenging superoxide radicals ($O_2^-$), consistent with the high $Ce^{3+}$-to-$Ce^{4+}$ ratio induced by asymmetric catalytic centers (Fig. 4a). More importantly, the SOD-mimetic activity of APROM$_2$ is superior to that of natural SOD (Fig. 4b, c, Supplementary Table 4). Notably, different from the downregulated CAT-mimetic activity affected by decreasing surface $Ce^{4+}$ levels[44], APROMs exhibit superior CAT-mimetic activity in decomposing $H_2O_2$ into $H_2O$ and $O_2$ compared to CeNPs, despite a lower $Ce^{4+}$-to-$Ce^{3+}$ ratio, which possibly due to the presence of asymmetric catalytic centers (Fig. 4d). It's worth mentioning that APROM$_2$ demonstrates a CAT-mimetic activity approximately 14.8-fold higher than that of CeNPs, with an affinity for $H_2O_2$ approximately 3.34-fold higher than that of natural CAT enzyme

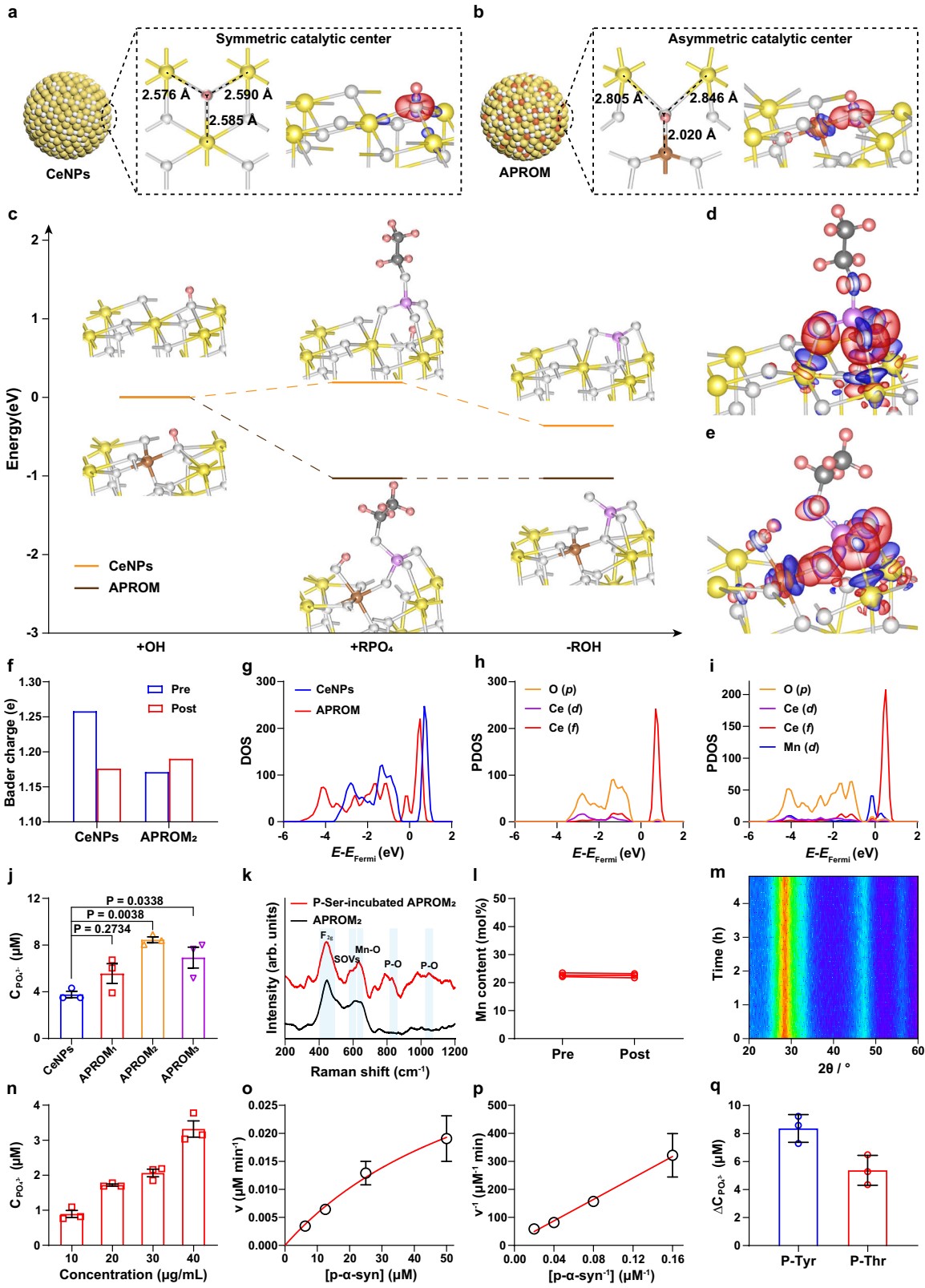

(Fig. 4e, f, Supplementary Figs. 12 and 13, Supplementary Table 5), as confirmed by steady-state kinetics. Furthermore, APROMs do not generate hydroxyl radicals (·OH) through Fenton-like reactions in the presence of $H_2O_2$ (Supplementary Fig. 14). Therefore, APROM$_2$ exhibits exceptional antioxidant activity, capitalizing on its asymmetric catalytic centers.

To gain a deeper understanding of the relationship between enhanced CAT-mimetic activity and asymmetric catalytic centers, DFT calculations were performed to investigate the catalytic mechanism. The key intermediates along the reaction pathway of $H_2O_2$ decomposition, along with their respective free energies, are depicted in Fig. 4e, confirming that both CeNPs and the APROM can

**Fig. 3 | Asymmetric catalytic centers confer protein phosphatase-like characteristics to APROMs.** Local coordination environments and corresponding differential charge density maps of OH absorbed CeNPs (**a**) and APROM (**b**). Red and blue represent accumulation and depletion charge areas, respectively. Ce, yellow; Mn, brown; O, white; H, pink; C, dark gray. **c** Free-energy diagram of phospho-substrate hydrolysis on the APROM and CeNPs. Differential charge density maps of OH and phospho-substrate absorbed CeNPs (**d**) and APROM (**e**). **f** Bader charge of the O atom of OH before and after phospho-substrate absorption. A positive value indicates that the atom gains electrons. **g** DOS of CeNPs and the APROM. $E$ = energy level, $E_{Fermi}$ = Fermi level. PDOS of CeNPs (**h**) and the APROM (**i**). **j** Protein phosphatases-mimetic activity of APROMs and CeNPs by using P-Ser as the phospho-substrate ($n = 3$ independent experiments). **k** Raman spectra of APROM$_2$ with and without incubation of P-Ser. The P-Ser is used as the phospho-substrate for the dephosphorylation reaction. $F_{2g}$, the Raman-active vibrational mode of the cubic fluorite structure. **l** The Ce-to-Mn ratio of APROM$_2$ before and after catalyzing phospho-substrate dephosphorylation. **m** In situ XRD of APROM$_2$ under dephosphorylation reaction conditions. **n** Protein phosphatases-mimetic activity of APROM$_2$ by using p-$\alpha$-syn as the phospho-substrates ($n = 3$ independent experiments). Michaelis–Menten kinetics (**o**) and Lineweaver–Burk plotting (**p**) of APROM$_2$ obtained by adding different concentrations of p-$\alpha$-syn ($n = 3$ independent experiments). **q** Protein phosphatases-mimetic activity of APROM$_2$ by using P-Tyr and P-Thr as the phospho-substrates ($n = 3$ independent experiments). All the data are presented as means ± s.e.m. Statistical significance was analyzed by one-way ANOVA with multiple comparisons test. Source data are provided as a Source Data file.

thermodynamically catalyze the decomposition of $H_2O_2$. The differential charge density analysis reveals that the APROM promotes the transfer of electrons to the $H_2O_2$ molecule more effectively than CeNPs, facilitating the activation and decomposition of $H_2O_2$ (Fig. 4f, g). Interestingly, in the APROM, the desorption of $O_2$, which is a rate-determining step from a thermodynamic perspective, is facilitated through two steps. Firstly, $O_2$ generated at the Ce site is transferred to the Mn site, allowing for the regeneration of the Ce site for $H_2O_2$ decomposition. Subsequently, $O_2$ at the Mn site can readily desorb with a low energy barrier. As a result, the CAT-mimetic activity of the APROM is significantly enhanced due to the presence of asymmetric catalytic centers (Fig. 4h).

## APROM$_2$ reprograms the biological function of α-syn to fuel synaptic function via de novo PTMs

Building upon the remarkable protein phosphatase-like characteristics of APROM$_2$, we further investigated its capacity to reprogram neuronal protein functions through de novo PTMs (Fig. 5a). As a proof-of-concept, we chose α-syn as the neuronal protein of interest. APROM$_2$ exhibits efficient cellular internalization in a time-dependent manner[45] (Supplementary Fig. 15) without inducing significant cytotoxicity (Supplementary Fig. 16). In subsequent experiments, we established a primary neuronal model with abnormal PTMs by treating neurons with the neurotoxin 1-methyl-4-phenylpyridinium (MPP$^+$). Upon MPP$^+$ exposure, the intracellular p-α-syn level significantly increases (Fig. 5b, c, Supplementary Fig. 17). The C-terminal region of α-syn, which contains motifs of charged amino acids, is capable of binding to metal ions[46]. Notably, phosphorylation of Ser 129 within the C-terminal region significantly enhances this affinity[47]. In line with these observations, APROM$_2$ remarkably dephosphorylates p-α-syn with decreasing the p-α-syn level approximate to the physiological level via effective interaction with p-α-syn, thereby rebalancing the aberrant PTMs. In contrast, although CeNPs demonstrate some reduction in the p-α-syn level in primary neurons, aberrant PTMs of α-syn still persist. Notably, while the antioxidant N-acetylcysteine (NAC) efficiently scavenges reactive oxygen species (ROS) (Supplementary Fig. 18), it only minimally reduces p-α-syn levels (Fig. 5b, c, Supplementary Fig. 19), suggesting that the protein phosphatase-mimetic activity of APROM$_2$, rather than its antioxidant properties, plays a crucial role in reducing p-α-syn levels. Subsequently, the impact of APROM$_2$ on synaptic transmission was assessed using the fluorescent dye FM1-43, which is incorporated into synaptic vesicles during endocytosis and released during vesicle-membrane fusion (Fig. 5d, e). Surprisingly, APROM$_2$ significantly enhances synaptic transmission. Given that α-syn is essential for maintaining synaptic homeostasis[11,48], we hypothesize that the dephosphorylation of p-α-syn by APROM$_2$ through de novo PTMs attributes to the restoration of synaptic transmission. Emerging evidence suggests that α-syn, primarily localized at presynaptic terminals, binds to vesicular monoamine transporter 2 (VMAT2)[11], facilitating the incorporation of dopamine into synaptic vesicles, and interacts with vesicle-associated membrane protein 2 (VAMP2)[48], promoting vesicle-

membrane fusion for dopamine release. To test our hypothesis, the colocalization of α-syn with VMAT2 and VAMP2 was examined (Fig. 5f–i). The ability of α-syn to bind with VMAT2 and VAMP2 is significantly impaired following MPP$^+$ exposure, which is substantially restored by treatment with APROM$_2$, confirming APROM$_2$'s ability to reprogram the biological function of α-syn in MPP$^+$-induced cell model of Parkinson's disease (PD). Recent studies have highlighted synaptic dysfunction as a key initial event in the progressive degeneration of axons and terminals[49], which is implicated in PD. Importantly, APROM$_2$ promotes the length and number of neurites, enhancing neuronal connectivity and improving the synaptic plasticity of primary neurons (Fig. 5j, Supplementary Fig. 20).

It is widely recognized that synaptic activity, as a highly energy-demanding process, relies heavily on mitochondrial function[50]. However, in the pathological microenvironment, mitochondrial dysfunction caused by excessive ROS production disrupts presynaptic energy homeostasis. Encouragingly, APROM$_2$ with antioxidant activities effectively alleviates oxidative stress (Fig. 5k) and exhibits remarkable mitochondrial protection (Fig. 5l, m). This suggests that APROM$_2$ has the potential to maintain presynaptic energy homeostasis. Furthermore, APROM$_2$ treatment alleviates MPP$^+$-induced neurotoxicity and significantly enhances cell viability (Fig. 5n). Collectively, APROM$_2$ fuels synaptic function through de novo PTMs of p-α-syn and protects mitochondria in MPP$^+$-induced cell model of PD, thereby safeguarding neurons against degeneration (Fig. 5o).

## APROM$_2$ protects dopaminergic neurons against neurodegeneration in PD

PD is currently incurable, as available treatments only provide temporary relief of motor symptoms[51], necessitating the development of more effective therapies. Increasing evidence suggests that p-α-syn contributes to synaptic dysfunction, leading to the onset and progression of PD[8,11,12]. Therefore, the PD mouse model was built to evaluate the therapeutic effect of APROM$_2$ in vivo (Fig. 6a). Since PD is characterized by progressive motor dysfunction in clinical, behavioral evaluations were assessed, including the rotarod test, the pole test, and the wire hang test (Fig. 6b–d). Motor coordination deficits emerge in PD mice that manifest as increased pole test time spent and decreased rotarod and wire hang test time spent. Dyskinesia in APROM$_2$-treated PD mice is significantly rescued, while CeNPs-treated mice still have motor coordination deficits.

Tyrosine hydroxylase (TH), a rate-limiting enzyme for dopamine synthesis, is regarded as the key pathologic hallmark of PD. Focusing on the substantia nigra (SN) where dopaminergic neurons reside, TH-positive cells are sparse and scattered in PD mice, reflecting the degeneration of dopaminergic neurons in the nigrostriatal pathway (Fig. 6e, f). Consistently, the 1-methyl-4-phenyl-1,2,3,6-tetrahydropyridine (MPTP)-induced neuronal death in the SN was further verified by terminal deoxynucleotidyl transferase dUTP nick end labeling (Supplementary Fig. 21). Based on these findings, APROM$_2$ remarkably rescues dopaminergic neurons. In contrast, disappointing

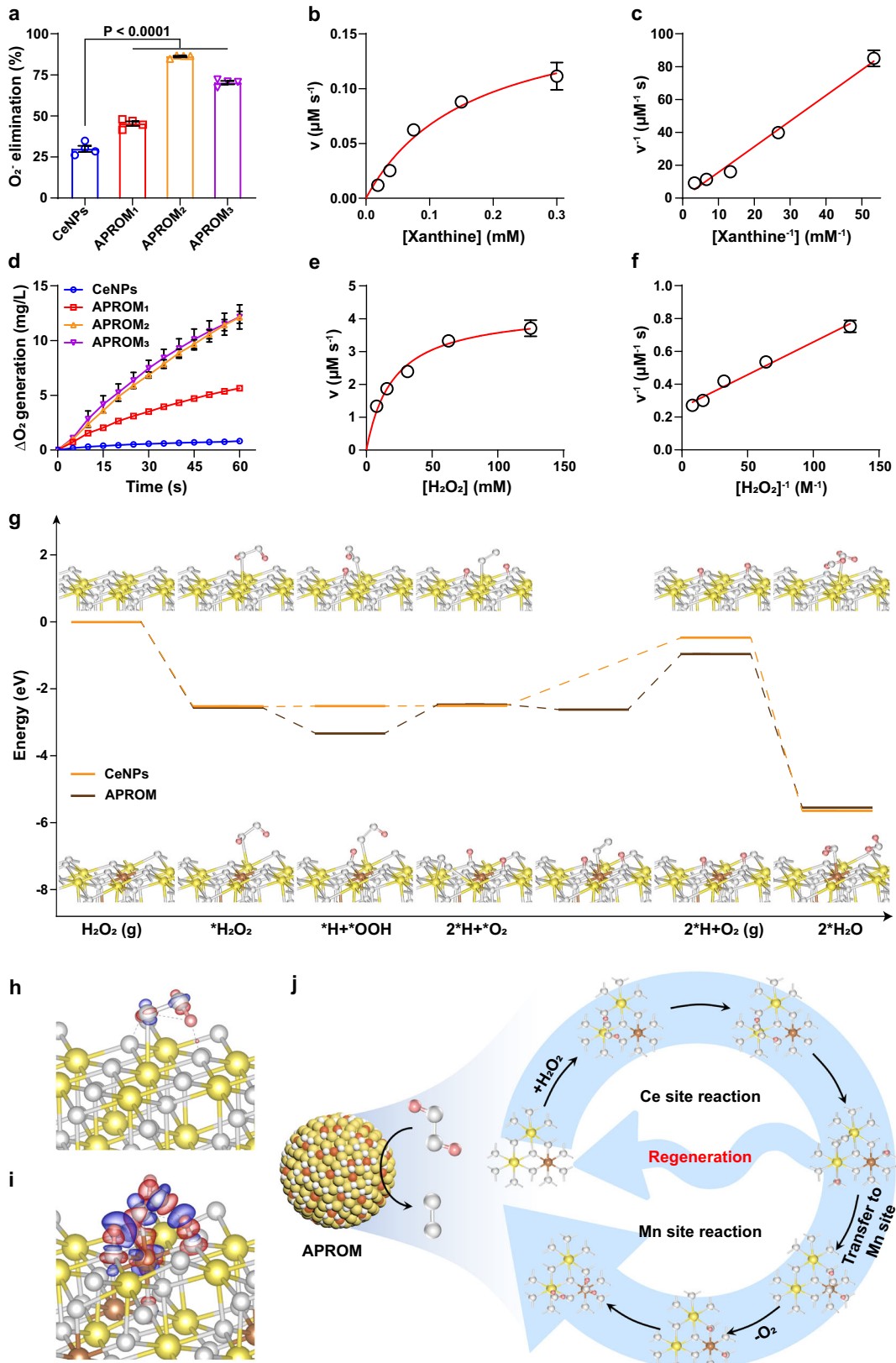

neurodegeneration develops in the SN of CeNPs-treated PD mice, despite the protective effect of CeNPs on dopaminergic neurons. Likewise, the loss of striatal dopaminergic fibers provoked by MPTP is rescued to a great extent upon the treatment of $APROM_2$ (Fig. 6g). Overall, these results suggest that $APROM_2$ can restore the motor function of PD by rescuing dopaminergic neurodegeneration.

## $APROM_2$ promotes synaptic plasticity in PD

To unravel the underlying mechanism of $APROM_2$'s capability to rescue dopaminergic neurons (Fig. 6h), we evaluated the phosphorylation levels of α-syn, a key pathological hallmark of PD, in midbrain tissues. $APROM_2$ effectively reverses the hyperphosphorylation state of α-syn at Ser-129 in both the SN and striatum (ST) (Fig. 6i, Supplementary

**Fig. 4 | Asymmetric catalytic centers endow APROMs with exceptional anti-oxidant activity. a** SOD-mimetic activity of APROMs and CeNPs ($n = 4$ independent experiments). Michaelis–Menten kinetics (**b**) and Lineweaver–Burk plotting (**c**) of APROM$_2$ obtained by adding different concentrations of xanthine ($n = 3$ independent experiments). **d** CAT-mimetic activity of APROMs and CeNPs ($n = 3$ independent experiments). Michaelis–Menten kinetics (**e**) and Lineweaver–Burk plotting (**f**) of APROM$_2$ obtained by adding different concentrations of H$_2$O$_2$ ($n = 3$ independent experiments). **g** The free-energy diagrams for H$_2$O$_2$ decomposition on the APROM and CeNPs. Ce, yellow; Mn, brown; O, white; H, pink. Differential charge density maps of H$_2$O$_2$ adsorbed CeNPs (**h**) and APROM (**i**). Red and blue represent accumulation and depletion charge areas, respectively. The isosurface level is 0.005 arb. units. **j** Schematic illustration of the catalytic mechanism of H$_2$O$_2$ decomposition in the asymmetric catalytic center. H$_2$O$_2$ is decomposed at the Ce site of the asymmetric catalytic center, and then the formed O$_2$ is transformed to the Mn site, so that the Ce site is regenerated for a new round of H$_2$O$_2$ decomposition. Moreover, O$_2$ can be readily desorbed at the Mn site with a low energy barrier. All the data are presented as means ± s.e.m. Statistical significance was analyzed by one-way ANOVA with multiple comparisons test. Source data are provided as a Source Data file.

Figs. 22 and 23). This ability highlights the capacity of APROM$_2$ to rebalance the abnormal PTMs of α-syn and consequently inhibit the subsequent formation of toxic α-syn inclusions (Fig. 6j, Supplementary Fig. 22). More importantly, through APROM$_2$'s de novo PTM strategy, the biological function of α-syn is regained, as verified by the promoted colocalization of α-syn with VMAT2 in dopaminergic neurons that is dependent on dephosphorylation (Fig. 6k–m, Supplementary Fig. 24), as well as colocalization of α-syn with VAMP2 (Fig. 6n). Moreover, synaptic vesicles are significantly recovered with the treatment of APROM$_2$ (Fig. 6o), and the dopamine level of PD mice return to normal physiological levels (Supplementary Fig. 25). This finding indicates the reestablishment of synaptic function. Additionally, consistent with previous studies demonstrating the association between α-syn dysfunction, MPTP-induced mitochondrial dysfunction, and oxidative stress in PD[11], we observed an increased level of 4-Hydroxynonenal (4-HNE), an indicator of oxidative damage, in PD mice, which is efficiently reduced by APROM$_2$ (Fig. 6p). Furthermore, APROM$_2$-treated mice exhibit downregulated expression of ionized calcium-binding adapter molecule 1 (IBA-1), a microglia marker, suggesting that APROM$_2$ can inhibit neuroinflammation (Supplementary Fig. 26). These collective results strongly support the notion that APROM$_2$ can promote the synaptic plasticity of dopaminergic neurons in the MPTP mouse model of PD. To evaluate the in vivo toxicity of APROM$_2$, we conducted a biosafety assessment. No visible hemolysis was observed when different concentrations of APROM$_2$ were tested, confirming its excellent blood compatibility (Supplementary Fig. 27). Morphological analysis of the SN and ST reveals no discernible differences following APROM$_2$ treatment (Fig. 6q, Supplementary Fig. 28). Moreover, APROM$_2$ does not induce any noticeable burden on major organs, including the heart, liver, lung, spleen, and kidney (Supplementary Fig. 29). As indicated in phosphoproteomic analysis, the phosphorylation levels of some metabolism-related proteins in the midbrain of APROM$_2$-treated PD mice are lower than those in normal mice, probably attributed to the pleiotropic effects of APROM$_2$, damage caused by MPTP or other factors (Supplementary Figs. 30–34). However, it is worth noting that, compared to PD mice, there is no discernible reduction in phosphorylation levels of aforementioned proteins in APROM$_2$-treated PD mice. Above all, these results demonstrate that APROM$_2$ exhibits excellent biocompatibility and does not induce local or systemic toxicity.

## Discussion

Reversible protein phosphorylation, the predominant PTM in eukaryotes, plays a pivotal role in reprogramming protein function across various cellular processes[4,5]. However, the catalytic activity of protein phosphatases, which regulate this process, is often compromised in pathological conditions[6,7], resulting in protein hyperphosphorylation, disruption of proteostasis, and disease development[8,9]. Despite clinical trials of small-molecule drugs targeting specific protein phosphatases[52], their limited regulatory efficacy arises from the complex nature of reversible protein phosphorylation involving multiple protein phosphatases[53].

In this study, we present a strategical approach by developing asymmetric catalytic center-engineered APROMs with protein phosphatase-like characteristics. Through a self-aggravating SOV-driven cation exchange strategy, site-specific single manganese atoms are precisely incorporated into ceria nanoparticles, creating asymmetric catalytic centers. These APROMs enable efficient rebalancing of aberrant PTMs of phospho-proteins that possess phosphate monoester bonds at amino acid residues like serine, tyrosine and threonine, through a de novo PTM strategy. As a proof-of-concept, we found the restoration of α-syn biological function using APROMs, resulting in enhanced synaptic plasticity in MPP$^+$-treated primary neurons and the rescue of dopaminergic neurons in MPTP mouse model of PD. While our data support the hypothesis that the dephosphorylation of p-α-syn can restore its biological function in MPP$^+$/MPTP models of PD, a more in-depth investigation in other PD models is still needed, given the ongoing controversy regarding the pathogenic relevance of p-α-syn[54–56].

Our findings present a promising avenue for reprogramming protein function through the de novo PTM strategy, which involves the dephosphorylation of hyperphosphorylated proteins by APROMs. While our preliminary in vitro and in vivo data demonstrate the direct reprogramming of α-syn by APROMs, future advancements in developing specific protein-targeted APROMs, capitalizing on their modifiable characteristics, are essential. These advancements will enable precise targeting and rebalancing of aberrant PTMs, particularly those involving phosphate monoester bonds at amino acid residues like serine, tyrosine, and threonine within affected areas. This targeted approach holds great promise for selectively reprogramming protein function and maintaining proteostasis, offering potential therapeutic strategies for various diseases. Moreover, the utilization of probe-modified APROMs will facilitate the study of protein-to-protein interaction patterns and the investigation of biological functions associated with reverse-modified proteins in cellular homeostasis.

## Methods

### Ethical statement

Animal experiments were performed according to institutional guidelines and were approved by the Institutional Animal Care and Use Committee of Zhejiang University School of Medicine and Shanghai Jiao Tong University School of Medicine.

### Materials

All reagents and solvents were obtained commercially and used without further purification. Cerium acetylacetonate hydrate, oleylamine, xylene, 1-ethyl-3-(3-dimethly-aminopropyl) carbodiimide (EDC), N-hydroxysuccinimide (NHS), methylene blue, and DNase I were purchased from Sigma-Aldrich. Manganese acetylacetonate, cerium acetate hydrate, citric acid (CA), 2-morpholinoethanesulfonic acid (MES), and N, N-dimethylformamide (DMF) were purchased from Aladdin. TrypLE™ Express, Dulbecco's Modified Eagle Medium F-12 (DMEM/F12), Neurobasal-A medium, fetal bovine serum (FBS), and B27 were purchased from Abcam. Poly-L-lysine (PLL) was purchased from Solarbio. Acetone, chloroform (CHCl$_3$), ethanol, and ethyl ether were purchased from Sinopharm. mPEG$_{2000}$-NH$_2$ and NH$_2$-PEG$_{2000}$-FITC were purchased from Ponsure Biotechnology Company.

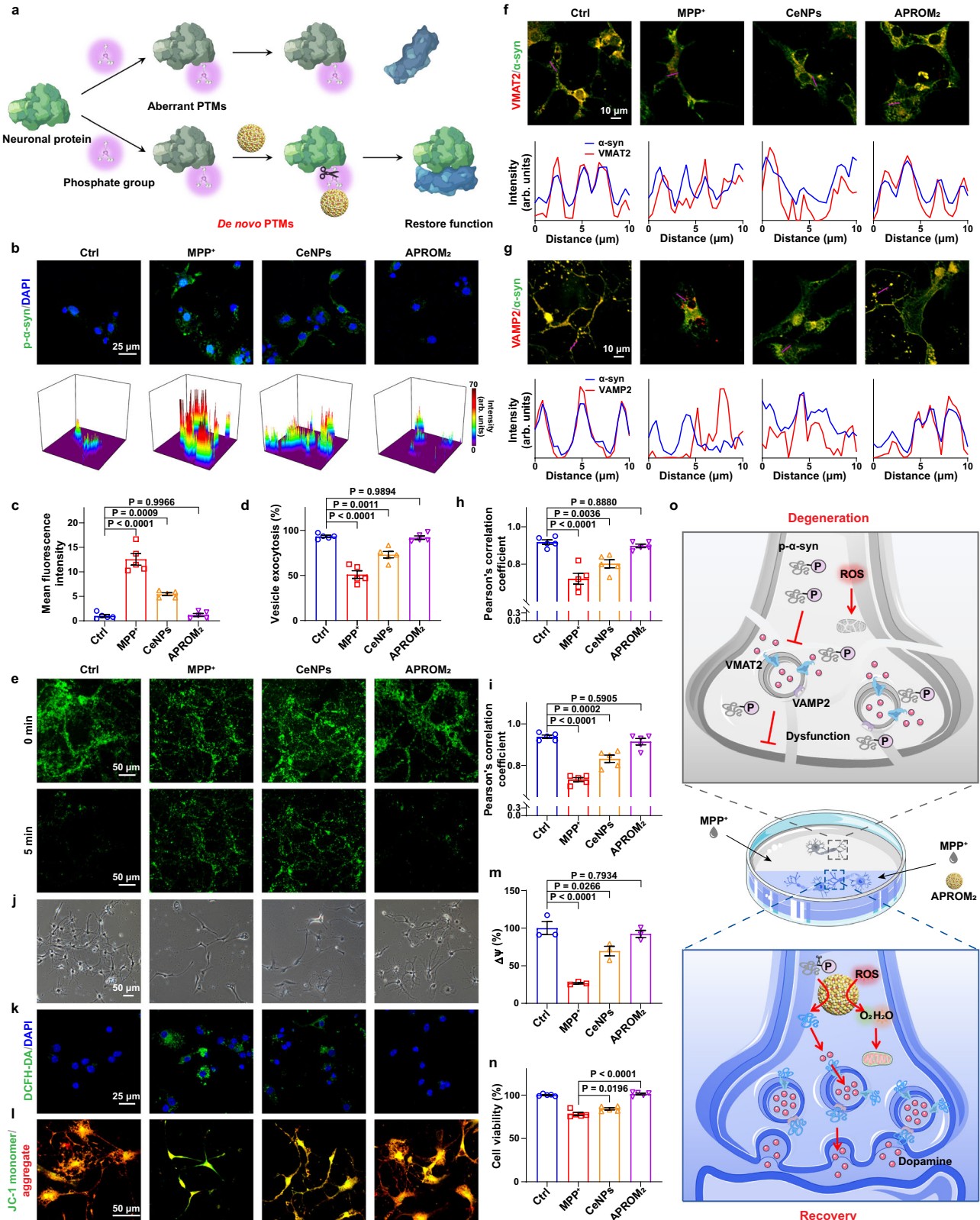

## Instruments

Transmission electron microscopy (TEM) images were taken to observe the morphology of CeNPs and APROMs (Hitachi HT7700, Japan). High-resolution transmission electron microscopy (HRTEM) images were taken to observe the facet of CeNPs and APROMs (FEI Tecnai F20, USA). High-angle annular dark-field scanning transmission electron microscopy (HAADF-STEM) images were obtained using a spherical aberration-corrected Titan ChemiSTEM microscope (FEI, USA). The concentrations of Ce and Mn were quantified by using ICP-MS (PerkinElmer NexION 300X, USA). The X-ray powder diffraction (XRD) patterns were obtained by using a Rigaku D/Max-2550 PC instrument (Rigaku, Japan). The in situ XRD pattern was obtained via using a D8 Advanced instrument (Bruker, Germany). X-ray photoelectron spectroscopy (XPS) spectra were obtained via using a Thermo

**Fig. 5 | APROM$_2$ reprograms neuronal protein functions via de novo PTMs to fuel synaptic function. a** Schematic illustration of the de novo PTM strategy of neuronal proteins using APROM$_2$. APROM$_2$ with protein phosphatase-like characteristics can dephosphorylate phospho-proteins, and thus restore the function of neuronal proteins. Created with BioRender.com. **b** Top, representative confocal laser scanning microscopy (CLSM) images of p-α-syn in primary neurons after different treatments. Bottom, 3D mapping of intracellular fluorescence. **c** Mean fluorescence intensity of p-α-syn in primary neurons after different treatments ($n = 5$ biologically independent cultures). Quantitative analysis (**d**) and representative CLSM images (**e**) of synaptic vesicle function indicated by FM1-43 after different treatments ($n = 5$ biologically independent cultures). Immunofluorescence and quantification analysis of the colocalization of α-syn with VMAT2 (**f**, **h**) or VAMP2 (**g**, **i**) in primary neurons after different treatments ($n = 5$ biologically independent cultures). **j** Microscopy images showing the neurite branches and neuronal connectivity of primary neurons after different treatments. **k** Intracellular ROS levels in primary neurons after different treatments. **l**, **m** Mitochondrial membrane potential (Δψ) is indicated by JC-1 staining (**l**) and quantified by normalized JC-1 aggregates/monomers ratio (**m**) ($n = 3$ biologically independent cultures). JC-1 monomers (green) represent low Δψ, and JC-1 aggregates (red) represent high Δψ. **n** Protective effect of APROM$_2$ and CeNPs on MPP$^+$ treated cells ($n = 5$ biologically independent cultures). **o** Schematic illustration of APROM$_2$ that reprograms α-syn function by the de novo PTM strategy and protects mitochondria for fueling synaptic function. APROM$_2$ directly modulates p-α-syn by cleaving the phosphate monoester bond, thus, α-syn regains biological functions of binding to VMAT2 and VAMP2. In addition, APROM$_2$ protects mitochondria against ROS to maintain presynaptic energy homeostasis. All the data are presented as means ± s.e.m. Statistical significance was analyzed by one-way ANOVA with multiple comparisons test. Source data are provided as a Source Data file.

Scientific ESCALAB 250 Xi XPS system (Thermo, UK). The hydrodynamic size of CeNPs and APROMs was detected by using Zetasizer Nano ZS90 (Malvern Instruments, UK). Raman spectra were obtained via using LabRAM HR evolution (HORIBA, France). The time-of-flight secondary ion mass spectrometry (ToF-SIMS) was obtained via using ToF-SIMS 5-100 (ION-TOF, Germany).

## Synthesis of artificial protein modulators (APROMs)

Firstly, a 0.5 g mixture of cerium acetylacetonate hydrate and manganese acetylacetonate in an appropriate molar ratio (molar ratios are 9:1, 7:3 and 5:5 for APROM$_1$, APROM$_2$ and APROM$_3$, respectively) was dissolved in 15 mL of oleylamine under vigorous sonicating. Then, the mixture was heated to 80 °C with a heating rate of 2 °C/min under the argon atmosphere. After aging at 80 °C for 24 h, the resulting nanoparticles (APROMs) were precipitated by the addition of acetone, collected via centrifuge ($12,000 \times g$, 15 min), washed with acetone three times, and finally dispersed in CHCl$_3$ for further use.

To further study the multi-metal collaborative catalytic center evolution mechanism. 0.371 g of cerium acetylacetonate hydrate and 0.129 g of manganese acetylacetonate were dissolved in 15 mL of oleylamine under vigorous sonicating. Then, the mixture was heated to 80 °C with a heating rate of 2 °C/min under the argon atmosphere. After aging at 80 °C for 0, 0.25, 0.5, 1, 3, 6, 12, and 24 h, the resulting nanoparticles were precipitated by the addition of acetone, and collected via centrifuge ($12,000 \times g$, 15 min) for further study.

## Synthesis of ceria nanoparticles (CeNPs)

CeNPs were synthesized via a modified reverse micelle method. Firstly, 0.43 g of cerium acetate hydrate and 3.25 g of oleylamine were added into 15 mL of xylene. The mixture was vigorously stirred at room temperature for 12 h, and then heated to 90 °C with a heating rate of 2 °C/min under the argon atmosphere. Subsequently, 1 mL of deionized water was added into the heated mixture. After aging at 90 °C for 3 h, the resulting nanoparticles (CeNPs) were precipitated by the addition of acetone, collected via centrifuge ($7000 \times g$ 15 min), and finally dispersed in CHCl$_3$ for further use.

## Surface modification of APROMs and CeNPs with CA and PEG

1 mL of APROMs or CeNPs and 0.3 g of CA were added into 15 mL of the mixture consisting of DMF and CHCl$_3$, followed by stirring at room temperature for 24 h. Subsequently, CA modified APROMs and CeNPs were precipitated by the addition of ethyl ether, collected via centrifuge ($21,000 \times g$, 15 min), and washed with acetone three times. The obtained CA modified APROMs and CeNPs were dispersed in MES solution (pH = 5.6) and activated with EDC/NHS (7.5 mg of EDC and 7.5 mg of NHS) for 30 min. Afterwards, 40 mg of mPEG$_{2000}$-NH$_2$ was added, and the mixture was stirred at room temperature overnight to synthesize CA and PEG modified APROMs and CeNPs.

## Density functional theory (DFT) calculations

The structure of CeO$_2$ cell was first obtained from Materials Project (legacy.materialsproject.org, Materials ID: mp-20194) and optimized. Ce$_{0.75}$Mn$_{0.25}$O$_2$ and Ce$_{0.5}$Mn$_{0.5}$O$_2$ cells were gained and replaced one or two Ce atoms to Mn atoms from the CeO$_2$ cell and optimized. Then the CeO$_2$ and Ce$_{0.75}$Mn$_{0.25}$O$_2$ bulk structures were cleaved as (1 1 1) surface and modeled with the (2 × 2) periodically repeated supercell, consisting of six atomic layers, with a vacuum space of 20 Å.

The calculations were performed using the DFT method in combination with a standard solid-state pseudopotential (SSSP) for efficiency, as implemented in the Quantum-Espresso package (Version 6.8). The kinetic energy cutoff for wave functions was set as 65 Ry, and the charge density cutoff for SSSP efficiency was set as 780 Ry, as SSSP efficiency recommended. For all geometry optimizations, the energy and force convergence criterion were set as $1.0 \times 10^{-4}$ arb. units and $1.0 \times 10^{-3}$ arb. units, respectively. 2 × 2 × 1 Monkhorst–Pack k-point mesh samplings were used. The Hubbard U parameters of Ce and Mn were set as 6 and 2.5 separately. The transition states were located with the nudged elastic band (NEB) algorithm.

## Protein phosphatases-mimetic activity assay

The protein phosphatase-mimetic activity of APROMs and CeNPs (20 μg/mL) was detected by using O-phospho-L-serine (P-Ser) as the phospho-substrate. The produced free phosphate was analyzed by using the malachite green assay and the absorbance at 620 nm was measured by using a UV-Vis spectrophotometer UV-2600 (Shimadzu, Japan). A phosphate standard curve was obtained by using KH$_2$PO$_4$ as the substrate.

Additionally, the protein phosphatase-mimetic activity of APROM$_2$ (20 μg/mL) was also detected by using p-α-syn, O-phospho-L-tyrosine (P-Tyr) and O-phospho-L-threonine (P-Thr) as the phospho-substrates. The kinetic assays of APROM$_2$ (20 μg/mL) were carried out at 37 °C using a series of P-Ser concentrations (6.25, 12.5, 25, 50, 100 μM) or a series of p-α-syn concentrations (6.25, 12.5, 25, 50 μM). The Michaelis–Menten constant was calculated by using GraphPad Prism 8.0 (GraphPad Software).

## Catalase (CAT)-mimetic activity assay

Firstly, 200 μL of APROMs or CeNPs (0.8 mg/mL) and 800 μL of 3% H$_2$O$_2$ solution were added into PBS buffer (pH = 7.4) with a total volume of 8 mL, and the reaction was carried out at 37 °C. The generated O$_2$ (unit: mg/L) at different reaction times was measured by an oxygen electrode on Dissolved Oxygen Meter JPSJ-606L (Leici, China). The kinetic assays of CeNPs, APROM$_2$ and natural CAT were carried out at 37 °C by adding different amounts (100, 50, 25, 12.5, 6.25 μL) of 30% H$_2$O$_2$ solution into PBS buffer (pH = 7.4) with a total volume of 8 mL. The Michaelis–Menten constant was calculated by using GraphPad Prism 8.0 (GraphPad Software).

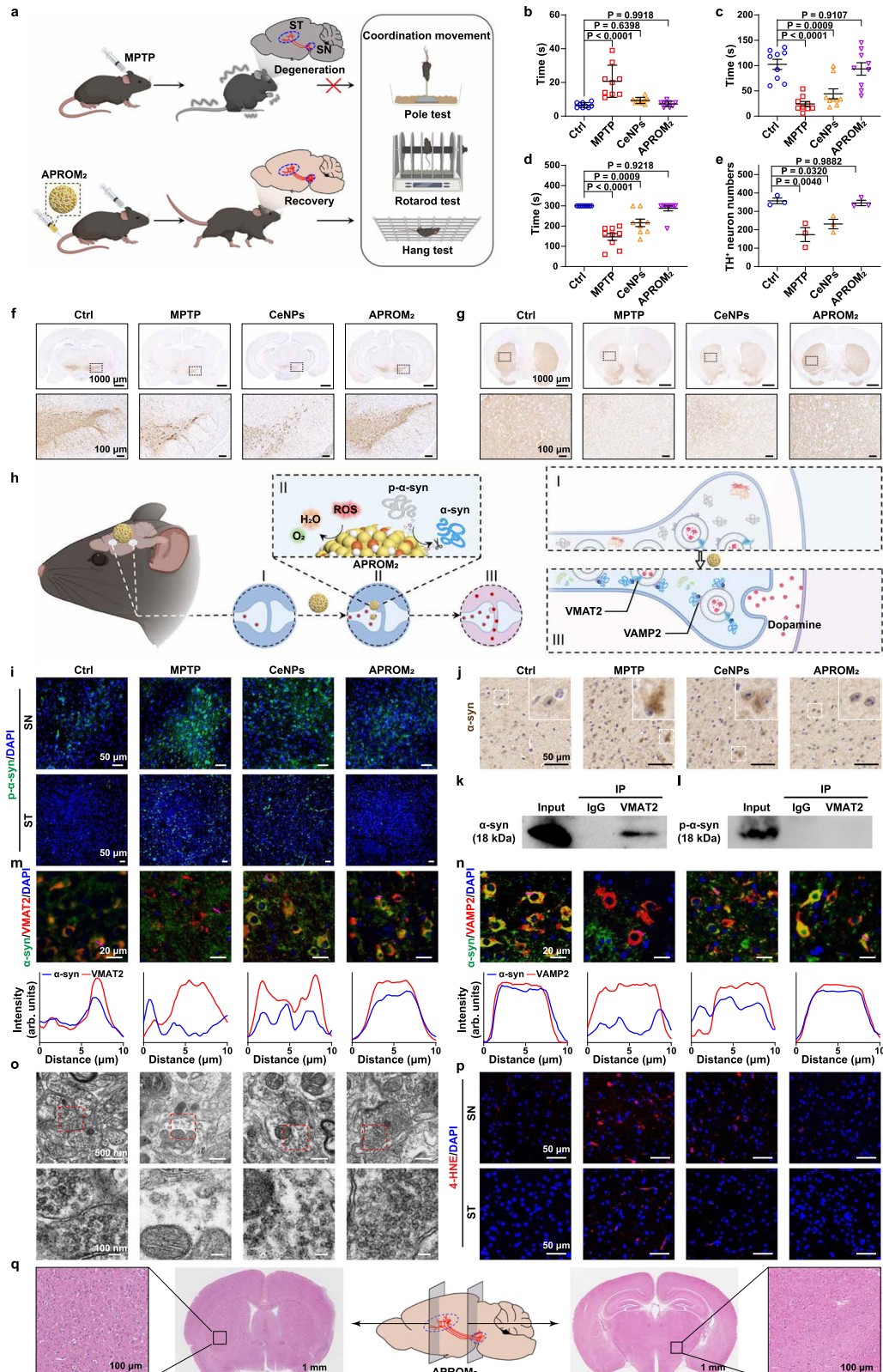

## Superoxide dismutase (SOD)-mimetic activity assay

The SOD-mimetic activity of APROMs and CeNPs (20 µg/mL) was tested by the Total Superoxide Dismutase Assay Kit with WST (Dojindo, Japan). Nitrotetrazolium blue chloride (NBT), an $O_2^-$ sensitive probe, was used to study the kinetic assay of $APROM_2$ and natural SOD (Beyotime, China). $APROM_2$ or natural SOD were mixed with NBT (100 µg/mL), xanthine oxidase (0.3 U/mL), and various concentrations of xanthine (0.01875, 0.0375, 0.075, 0.15, 0.3 mM) in tris-HCl buffer (0.1 M). The mixed solution was continuously monitored the absorbance at 550 nm by using a UV-Vis spectrophotometer UV-2600 (Shimadzu, Japan). The Michaelis–Menten constant was calculated by using GraphPad Prism 8.0 (GraphPad Software).

**Fig. 6 | APROM₂ mediated synaptic plasticity improvement in vivo. a** Schematic illustration of APROM₂ that protects dopaminergic neurons against neurodegeneration for rescuing motor coordination in PD. Created with BioRender.com. Behavioral evaluation of pole test (**b**) rotarod test (**c**) and hang test (**d**) of mice after different treatments ($n$ = 9 biologically independent mice). Quantitative analysis (**e**) and representative immunohistochemical staining images (**f**) of TH positive neurons in the SN ($n$ = 3 biologically independent mice). Regions of interest (white square) in the top panels are shown at higher magnification in the bottom panels. **g** Representative immunohistochemical staining images of dopaminergic fibers in the ST. Regions of interest (white square) in the top panels are shown at higher magnification in the bottom panels. **h** Schematic illustration of APROM₂ that improves synaptic plasticity of dopaminergic neurons in PD. I hyperphosphorylation of α-syn and oxidative stress impair synaptic function of dopaminergic neurons. II, APROM₂ dephosphorylates p-α-syn via protein phosphatase-mimetic activity and scavenges ROS via antioxidant activity. III, APROM₂ reprograms α-syn biological function via the de novo PTM strategy and alleviates oxidative stress for fueling synaptic function. Created with BioRender.com. **i** Representative immunofluorescence staining images of p-α-syn in the SN and ST. **j** Representative immunohistochemical staining images of α-syn inclusion in the SN. Inset, higher magnification of dopaminergic neurons. Co-immunoprecipitation of VMAT2 with α-syn (**k**) or p-α-syn (**l**) in the midbrain of APROM₂ treated PD mice. Representative immunofluorescence staining images and colocalization analysis along the white line of α-syn with VMAT2 (**m**) or VAMP2 (**n**). **o** Bio-TEM images of synaptic vesicles in the SN of mice after different treatments. Regions of interest (red square) in the top panels are shown at a higher magnification in the bottom panels. **p** Representative immunofluorescence staining images of 4-HNE. **q** Schematic of the midbrain section and histological assay with H&E staining for the SN and ST of APROM₂ treated mice. All the data are presented as means ± s.e.m. Statistical significance was analyzed by one-way ANOVA with multiple comparisons test. Source data are provided as a Source Data file.

## Cell culture

SH-SY5Y cells obtained from Procell Life Science & Technology Co., Ltd (CL-0208, Wuhan, China) were cultured in DMEM supplemented with 10% FBS at 37 °C under the humidified 5% $CO_2$ atmosphere.

The midbrains of mice within 24 h of birth were harvested quickly on a cold stage, and then digested by TrypLE™ Express containing 1 mg/mL DNase I at 37 °C for 12 min. Subsequently, the cell suspension was passed through filters and centrifuged at $200 \times g$ for 5 min. The cells were resuspended in DMEM/F12 medium supplemented with 10% FBS, and plated onto PLL coated plates at 37 °C and 5% $CO_2$ atmosphere for 4 h. Then, the culture medium was replaced by Neurobasal-A medium supplemented with 2% B27. The maturation of primary neurons required 7-8 days in vitro (DIV) with medium changes every 2-3 days. The experiments, including the assessment of cellular ROS-scavenging capability, mitochondrial protective effect, and microscopy images of primary neurons, were conducted over a period of 7-8 DIV. Additionally, experiments measuring phosphorylation levels, synaptic activity and colocalization analysis were carried out after an additional week of culture.

## In vitro cell uptake assay

SH-SY5Y cells ($10^6$ cells/mL) were seeded in confocal laser scanning microscopy (CLSM)-specific dishes for CLSM imaging. SH-SY5Y cells were incubated with FITC modified APROM₂ (20 μg/mL) at 37 °C for 1, 2, 4, 6, and 12 h, respectively. Then, SH-SY5Y cells were washed with PBS three times, and fixed in 4% paraformaldehyde for 15 min. The nuclei were stained with DAPI for 8 min. Subsequently, SH-SY5Y cells were observed by using a laser scanning confocal microscope (Olympus FV1200, Japan).

## Cell viability assay

To test the biocompatibility of APROM₂ and CeNPs, SH-SY5Y cells were cultured with different concentrations of APROM₂ (0, 2.5, 5, 10, 20, 30 μg/mL) or CeNPs (0, 2.5, 5, 10, 20, 30 μg/mL) for 24 h at 37 °C and 5% $CO_2$ atmosphere. Then, 100 μL of DMEM medium containing 50 μg of MTT was added to each well. 3 h later, the cultured medium was extracted and 100 μL of dimethyl sulfoxide was added to each well. Afterward, the absorbance at 570 nm was measured on a microplate reader (Bio Tech, USA).

To investigate the protective effect of APROM₂ and CeNPs, DMEM medium containing MPP⁺ (1 mM) and APROM₂ (20 μg/mL) or CeNPs (20 μg/mL) was added into culture dishes, and SH-SY5Y cells were cultured for 24 h at 37 °C and 5% $CO_2$ atmosphere. Then, 100 μL of DMEM medium containing 50 μg of MTT was added to each well. 3 h later, the cultured medium was extracted and 100 μL of dimethyl sulfoxide was added to each well. Afterward, the absorbance at 570 nm was measured on a microplate reader (Bio Tech, USA).

## In vitro immunofluorescence

After different treatments, primary neurons were fixed with 4% paraformaldehyde for 20 min, permeabilized with 0.2% Triton X-100 for another 20 min, and then blocked with 5% bovine serum albumin in PBS for 1 h at room temperature. Subsequently, primary neurons were incubated with primary antibodies at 37 °C for 2 h: anti-phospho-synuclein alpha (Ser129) (cat. no. AF3285, 1:500, Affinity Biosciences), anti-VMAT2 (cat. no. PA5-112713, 1:200, Thermo Fisher Scientific), anti-VAMP2 (cat. no. DF6381,1:200, Affinity Biosciences), and anti-α-syn (cat. no. OM239190, 1:200, Omnimabs). Then, primary neurons were incubated with second antibodies from Boster Biological Technology Co., Ltd. for 1 h at room temperature. The nuclei were stained with DAPI. The fluorescence images were obtained by using a laser scanning confocal microscope (Olympus FV1200, Japan) and the Pearson's correlation coefficient was calculated by using Fiji software (version 1.54 f).

## Analysis of synaptic activity

Fluorescent dye FM1-43 was used to mimic the release of neurotransmitters as it can be incorporated into the synaptic vesicle membrane, and the release of neurotransmitters was measured by the decrease of FM1-43 fluorescence intensity in primary neurons. Firstly, DMEM medium containing MPP⁺ (50 μM) and APROM₂ (20 μg/mL) or CeNPs (20 μg/mL) was added into culture dishes, and primary neurons were cultured for 24 h. Then, primary neurons were incubated with HEPES buffer containing 100 μM FM1-43, 1 mM $Ca^{2+}$ and 30 mM $K^+$ for 2 min, and the fluorescent signal was tested by using a Nikon confocal microscope (Nikon, Japan). Subsequently, neurons were incubated with HEPES buffer containing 1 mM $Ca^{2+}$ and 15 mM $K^+$ for 5 min, during which FM1-43 contained synaptic vesicles were released. The fluorescent signal was detected again by using a Nikon confocal microscope.

## Cellular ROS-scavenging capability

To assay the ROS-scavenging capability of APROM₂ and CeNPs, DMEM medium containing MPP⁺ (50 μM) and APROM₂ (20 μg/mL) or CeNPs (20 μg/mL) was added into culture dishes, and primary neurons were cultured for 24 h. Subsequently, primary neurons were incubated with DCFH-DA at 37 °C for 20 min and fixed in 4% paraformaldehyde for 15 min. The nuclei were stained with DAPI. The fluorescence images were obtained by using a laser scanning confocal microscope.

## Analysis of mitochondrial protective effect

To test the mitochondrial protective effect of APROM₂ and CeNPs, DMEM medium containing MPP⁺ (50 μM) and APROM₂ (20 μg/mL) or CeNPs (20 μg/mL) was added into culture dishes, and primary neurons were cultured for 24 h. Afterward, primary neurons were washed three

times with PBS and treated according to the JC-1 kit. Finally, primary neurons were detected by a laser scanning confocal microscope.

## Acute PD mice model and in vivo therapy

7-8 week-old male C57BL/6 mice obtained from Shanghai SLAC Laboratory Animal Co. Ltd. (Shanghai, China) were trained for the behavioral tests and randomly divided into four groups (Sham group, MPTP group, CeNPs group, and $APROM_2$ group). Mice from each group were administrated with saline, CeNPs (2 mg/kg body weight), or $APROM_2$ (2 mg/kg body weight) by tail vein injection. 24 h later, mice were treated with MPTP (20 mg/kg body weight) via subcutaneous injection four times at 2 h intervals to induce PD. The control group was treated with the equivalent volume of saline. 24 h later, mice were administrated with saline, CeNPs (2 mg/kg body weight), or $APROM_2$ (2 mg/kg body weight) again by tail vein injection. 24 h after the last injection, behavioral assessments were performed, including rotarod test, hang test and pole test. Afterward, mice were sacrificed for further study.

## In vivo immunohistochemistry and immunofluorescence

Mouse brains were fixed in 10% formalin, embedded in paraffin, and sectioned. Brain sections were collected for further analysis of TH positive neurons and dopaminergic fibers (anti-TH antibody, cat. no. 25859-1-AP, 1:5000, Proteintech), p-α-syn levels (anti-phospho-synuclein alpha (Ser129) antibody, cat. no. AF3285, 1:500, Affinity Biosciences), α-syn inclusion (anti-α-syn antibody, cat. no. FNab09891, clone number: 5C6, 1:1000, FineTest), colocalization of α-syn (anti-α-syn antibody, cat. no. OM239190, 1:200, Omnimabs) with VMAT2 (anti-VMAT2 antibody, cat. no. PA5-112713, 1:200, Thermo Fisher Scientific) or VAMP2 (anti-VAMP2 antibody, cat. no. DF6381,1:200, Affinity Biosciences), oxidative stress (anti-4-HNE antibody, cat. no. bs-6313R, 1:200, Bioss) and inflammation (anti-IBA1 antibody, cat. no. BM5765, 1:100, Boster) by immunohistochemical staining or immunofluorescence staining.

## Co-immunoprecipitation

Proteins were extracted by incubating brain tissues with co-immunoprecipitation lysis buffer (20 mM Tris (pH = 7.5), 150 mM NaCl, 1% Triton X-100), supplemented with freshly added protease inhibitor cocktail and phosphatase inhibitor cocktail, for 30 min on ice. The lysates were then centrifuged (13,700 × g) for 10 min at 4 °C. Subsequently, the supernatant was incubated overnight at 4 °C with rocking, along with anti-α-syn (cat. no. OM239190, Omnimabs), anti-VAMT2 (cat. no. ab259970, clone number: EPR24197-51, Abcam), anti-p-α-syn (cat. no. AF3285, Affinity Biosciences), or IgG (cat. no. 30000-0-AP, Proteintech). Then, samples were incubated with magnetic protein A/G beads (MedChemExpress) for 60 min at 4 °C with gentle rocking. The formed complexes were washed with chilled buffers, and then eluted by boiling in 5X loading buffer for western blotting analysis.

## Hematoxylin and eosin (H&E) staining

Mice were treated with saline, CeNPs, and $APROM_2$ (2 mg/kg body weight) once every two days in two doses. 1 d after the last injection, hearts, livers, spleens, lungs, kidneys, and brains were harvested, fixed in 10% formalin, embedded in paraffin, sectioned, and stained by H&E for further analysis.

## Statistics & reproducibility

Data were presented as means ± s.e.m. ($n \geq 3$) and data analysis was performed using GraphPad Prism Software Version 8.0 (GraphPad Prism, USA) and Origin 2018 (OriginLab Corporation, USA). For statistical comparison, we performed one-way ANOVA to determine the significance. Sample size choice was based on previous studies (refs. 57–60), not predetermined by a statistical method. Sample sizes were indicated in the legend of each Figure and Supplementary Figure.

No data were excluded. We confirm all attempts at replication were successful. Replicates were conducted for all experiments quantified as described in the Figure legends. Figures 2b, p, q, 5b, j, k, 6f, g, i–q and Supplementary Figs. 3, 5a, 6, 7, 9, 15, 17, 18, 19a, 21, 24, 26, 28, 29 were repeated at least three times and representative example are shown. All samples were randomly allocated into experimental groups. Investigators were not blinded for nanomaterial synthesis, because determination of nanoparticles concentrations are considered as objective measures, not subject to bias. For in vivo experiments, the investigators were blinded to group allocation during data collection and analysis.

## Reporting summary

Further information on research design is available in the Nature Portfolio Reporting Summary linked to this article.

## Data availability

The data generated in this study are provided in the Figures, Supplementary Information, and Source Data file. Source data are provided with this paper.

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

## Acknowledgements

We gratefully acknowledge the support from the National Key Research and Development Program of China (2022YFB3203801, 2022YFB3203804, and 2022YFB3203800 to D.L. and F.L.), Leading Talent of "Ten Thousand Plan"-National High-Level Talents Special Support Plan, National Natural Science Foundation of China (32071374 to F.L.), Program of Shanghai Academic Research Leader under the Science and Technology Innovation Action Plan (21XD1422100 to D.L.), Explorer Program of Science and Technology Commission of Shanghai Municipality (22TS1400700 to D.L.), start-up funds from Shanghai Jiao Tong University (22×010201631 to D.L.), Zhejiang Provincial Natural Science Foundation of China (LR22C100001 to F.L.), Innovative Research

Team of High-Level Local Universities in Shanghai (SHSMU-ZDCX20210900 to D.L.), the CAS Interdisciplinary Innovation Team (JCTD-2020-08 to D.L.), Shanghai Municipal Science and Technology Commission (21dz2210100 to D.L.). The authors thank the beamline BL14W1 of Shanghai Synchrotron Radiation Facility (SSRF, China) for providing the beamtime.

## Author contributions

D.L. and F.L. conceived and supervised the project. D.L., F.L., P.L. and B.Z. developed the study. P.L. fabricated the catalyst. P.L., H.Y. and (Shiyi Yu) S.Y. characterized the catalyst. J.Z and C.F carried out the XAFS measurements. Y.Z. and L.C. performed the in-situ XRD characterizations. B.Z. performed the DFT calculation. P.L., H.Y. and Y.C. performed the cell and animal experiments and analyses. P.L. drafted the manuscript. B.Z., H.Y, P.X., (Shengfei Yang) S.Y, J.Z, Y.Z. and C.F. provide constructive advice for data analysis and manuscript writing. All the authors discussed the results and approved the final version of the manuscript.

## Competing interests

The authors declare no competing interests.
