## [Peer Review File · Nature Communications]

REVIEWER COMMENTS

Reviewer #1 (Remarks to the Author):

In their manuscript, Lin et al describe an artificial protein modulator (APROM) that exhibits protein phosphatase-like activity (and apparently also catalase-like activity). A major part of the work deals with the development of APROM. The authors end the manuscript by proposing that APROM can be used to mitigate alpha-synuclein toxicity. The idea might be that APROM dephosphorylates phosphorylated Serine 129 alpha-synuclein and could on top of that scavenge free radicals due to its catalase activity. Overall, the development of the APROM seems like a good achievement from a chemist's point of view. In terms of its strategic use conceptually and experimentally, there is less enthusiasm. The results of using the new tool in biological systems, as presented in the manuscript, beg more questions than answering anything significant. The following comments may, however, be helpful for the authors to prepare their study for a specialized journal.

1. Protein phosphatase superfamily comprises several members that efficiently modulate protein functions by closely working with kinases during dynamic reversible phosphorylation. It's somewhat unclear whether APROM mimics any or how many of the phosphatases. A detailed characterization of APROM, along with various protein phosphatases, will provide valuable information.

2. It should be explained better why this protein phosphatase also possesses a catalase-like activity.. Is APROM closer to an enzyme that scavenges free radicals or a phosphatase? While it can theoretically be viewed as an advantage that APROM both dephosphorylates aSyn and scavenges free radicals, the big concern is that APROM is not only a phosphatase of undefined specificity, but may - beyond catalase activity – have additional pleiotropic effects. This raises critical questions about its application and this limitation must be highlighted.

3. Dynamic reversible phosphorylation of proteins coordinated by kinases and phosphatases can be crucial for a given protein's functions. In pathological states, dynamic reversible phosphorylation may have been affected. It would thus be important to bring back the conditions to non-diseased states. If decreasing abnormal phosphorylation is a key therapeutic approach, we would have a meaningful treatment for synucleinopathies, as the authors claim. However, the role of aSyn phosphorylation in PD pathology is not that clear - and detrimental, protective and neutral roles have all been proposed. A recent publication (DOI: 10.1073/pnas.2109617119) suggests that serine-129 phosphorylation is a late epi-phenomenon that is not involved in aggregation (or toxicity of aSyn).

It is to be noted that hyper- or hypo-phosphorylation can equally be bad for a given biological process. APROM phosphatase activity does not seem to be specific. What if APROM dephosphorylated important PTMs in proteins required for normal biological processes? More details below.

4. The rationale for testing APROM's application in synucleinopathies lacks a rationale. Anyway, it will be necessary first to test whether APROM dephosphorylates Serine 129 alpha-synuclein directly in vitro. And how comparable is its activity to PP2A?

5. If the authors wish to study the effects of APROM on alpha-synuclein's synaptic function better, they should consider working with experts in the field. All results described in Figure 5 are sub-standard quality per the current synuclein research. Based on the data, the phrase "APROM reprograms the biological function of neuronal proteins to fuel synaptic function via de nova PTMs" is overstated and premature. Having someone who worked with cultured neurons for several years, I wouldn't dare to say that the maturation of primary neurons requires only 7-8 days. The primary dopaminergic neuron culture the authors described in the methods appears not standard in the field. A proper QC needs to be done before doing any experiments with this culture. Again, I believe working with experts will help the authors.

6. A more thorough job is needed for the in vivo experiments. First, quantification for TH-positive neurons is required (dopamine levels should be measured). Then, a detailed mechanism needs to be worked out. The cartoon in Fig. 6B is too simple. APROM appears to have a pleiotropic effect. Does APROM work only on alpha-synuclein? It is very unlikely. APROM might work on 100s of proteins dephosphorylating if I am not too generous with the numbers.

7. Simply presenting the APROM development may already be sufficient for a specialized journal. Given the limitations it might pose (as it stands), it will be too difficult to attribute any translational application for APROM.

Reviewer #2 (Remarks to the Author):

Reviewer #3 (Remarks to the Author):

Post-translational modifications (PTMs) of proteins are important for cellular functions. Protein phosphatases play a crucial role in the regulation of phosphorylation, a vital PTM. However, this regulation is disrupted in pathological conditions, such as neurodegenerative diseases, which have detrimental effects on neuronal proteins. The development of artificial protein modulators (APROMs) could solve this problem.

This study utilized a newly developed catalyst with a single manganese (Mn) atom, Mn/CeO_{2-x}, which employs a self-weighted surface oxygen vacancy (SOV) as its active center. This catalyst, similar to protein phosphatase, can correct abnormal PTMs in neurons. Furthermore, this catalyst possesses antioxidant properties that may contribute to improved mitochondrial function.

The properties and structure of the catalysts were analyzed in detail using X-ray diffraction (XRD), X-ray photoelectron spectroscopy (XPS), high-resolution transmission electron microscopy (HRTEM), and density functional theory (DFT) calculations, which were performed to gain a deeper understanding of the mechanism of action of the catalyst and SOV formation. The results of these analyses indicate that the bond between Mn and oxygen (O) is shorter than that between Ce and O, which accounts for the high activity of the catalyst.

APROM2 has been demonstrated to have a range of effects, particularly in neurons, including the ability to correct abnormal post-translational modifications (PTMs) and normalize the function of neuronal proteins. Additionally, it has been reported to improve synaptic function, which is critical in progressive neurological diseases, and has neuroprotective effects in animal models of Parkinson's disease (PD) by protecting dopaminergic neurons and enhancing synaptic function. Furthermore, biocompatibility and low toxicity of APROM2 were confirmed.

Therefore, APROM2 can be used to treat neurological disorders. It has the ability to rectify abnormal protein phosphorylation and enhance overall neuronal well-being. These discoveries may profoundly impact the development of novel therapies for neurodegenerative diseases, particularly Parkinson's disease.

The development of artificial catalysts has been validated, and their efficacy has been determined through a range of methods, including cell tests and immunohistochemistry. However, a more specialized evaluation is required for the first half of Figures 1-4. The peer review results of the second half of Figures 5-6, which encompasses the quantitative analysis of biochemistry and immunohistochemistry, are provided. Further verification is necessary for the therapeutic effects, including the quantitative analysis of biochemistry and quantitative immunohistochemical analysis and the verification of α -synuclein phosphorylation and inclusion body formation through biochemical analysis.

(1) Regarding the cell dosing experiment presented in Figure 5, I believe that APROM may have been administered earlier in the day. In such circumstances, whether the cells exhibit dephosphorylation or a reduction in MPTP toxicity is unclear. In order to address this issue, the authors should consider the timing of administration and explore experimental methods that can verify this. Some potential

approaches include adjusting the timing of administration or utilizing techniques such as pulse-chase analysis.

(2) While Figure 5b, c shows dephosphorylation without affecting the amount of synuclein, additional experiments, such as Western blotting, are necessary to confirm this.

(3) Sup16 is needed to prove the hypothesis, but it should be provided with appropriate contrast to clarify it.

(4) While Figure 5d-g shows confocal localization to synaptic vesicles, additional methods, such as isolating vesicles and verifying the proteins present, are necessary to prove that a-syn binds directly to VMAT2 and that this is dependent on phosphorylation.

(5) Figure 5j has not been analyzed quantitatively.

(6) Figure 6 lends credence to the notion that APROM mitigates the acute toxicity of MPTP, but it is uncertain whether APROM is dephosphorylated, as they propose. Demonstrating the drug's effect following phosphorylation would provide a more rigorous examination of the hypothesis. Alternatively, other animal models can be utilized to test this hypothesis.

(7) The immunohistochemical analysis of Figure 6 has not been biochemically or quantitatively analyzed, biochemically insoluble fractions, and the degree of phosphorylation needs to be shown for 6i.

(8) I cannot confirm the expression of α -syn in Figures 6k and l.

(9) While the paper suggests improved synaptic function in Parkinson's disease, this is an oversimplification as the study is conducted in a model system of cultured cells.

(10) Although the results suggest that neuronal cell death may be inhibited, it is impossible to describe morphology or synaptogenesis from these data.

In line 37, you mention improved synaptic function in Parkinson's disease, but I think this is an oversimplification, as the paper only shows results in a model system of cultured cells.

Lines 425 and 492, but from these results, it is not possible to describe morphology or synaptogenesis, although it is possible that neuronal cell death is inhibited.

Reviewer #4 (Remarks to the Author):

Comments to Manuscript NCOMMS-23-36937-T

This manuscript describes interesting and noteworthy results, but the work requires substantial improvement and the manuscript requires revision addressing the specific comments below.

1) Page 3, Line 65: This formulation needs to be improved. What is meant by the authors with the term “asymmetric molecular catalysts”?

2) Page 4, Line 66: With the unclear definition of “asymmetric molecular catalysts” the challenges of the low solubility and off-target side effects should be explained in more details.

3) Page 4, Line 68: It should be explained in more detail why nanomaterials based heterogeneous catalysts show promise in overcoming the inherent limitations of molecular catalysts.

4) Page 13, Line 241: Which classes of phospho-substrates?

5) Page 13, Line 258: The general statement that “ ... the APROM exhibits strong catalytic activity in phospho-substrate hydrolysis, resembling the function of protein phosphatases.”

needs to be supported by a range of representative phospho-substrates.

6) Page 13, Line 263: Have any other phospho-substrates than O-phospho-L-serine (P-Ser) been tested? If only O-phospho-L-serine (P-Ser) was used, then the term “ phospho-substrates” should be replaced by the term “phospho-substrate”.

7) Page 14 Line 265: The statement that “Comparative analysis reveals that APROMs exhibit superior protein phosphatase-mimetic activity” is not sufficiently supported by figures 3j and the supplementary figure 10. The kinetic characterization, statements on the kinetics and catalytic efficiency of the APROMS with regard to their protein phosphatase-mimetic activity, and a figure on the dependence of initial rates on the concentration of O-phospho-L-serine (P-Ser) are missing.

8) Page 14, Line 267: It is not entirely clear what the authors mean by “optimal protein phosphatase-mimetic activity”.

9) Page 14, Line 268: Have any experimental tests been done to check whether APROM2 can efficiently dephosphorylate p- α -syn, converting it into α -syn?

10) Page 14, Line 269: What was the phospho-substrate of the dephosphorylation reaction shown in figure 3k?

11) Page 14, Line 274: This should be experimentally demonstrated.

12) Page 16, Line 296: Figure 4a is giving inhibition rate in per cent and not the superoxide dismutase (SOD)-mimetic activity of APROMs mentioned in the text. A comparison of the kcat of the APROMS with the ones of the very efficient superoxide dismutases should be provided.

13) Page 16, Line 300: A definition of CAT-mimetic activity is missing and for the CAT-mimetic activity of APROM2 kcat should be provided, in addition to v_{max} (also for CeNP and natural CAT)

14) Page 19, Line 346: This statement should be improved and be made more specific.

15) Page 19, Line 348: A figure for the APROM2-catalyzed dephosphorylation kinetics of p- α -syn should be added.

16) Page 24, Line 435: What about the effect of APROM2 on the dephosphorylation of α -syn phosphorylated at other phosphorylation sites, such as S87, Y125, S129, Y133, and Y136?

17) Page 27, Line 486: Which of these multiple protein phosphatase functions can be performed by APROMs?

18) Page 28, Line 491: The statement on the “efficient rebalancing of aberrant PTMs of phospho-proteins” is too general and should be made more specific.

19) Page 28, Line 495: This “avenue for reprogramming protein function using the de novo PTM strategy” should be discussed in more detail.

20) Page 28, Line 498: The “precise rebalancing of aberrant PTMs” should be specified.

REVIEWER COMMENTS

Reviewer #1 (Remarks to the Author):

In their manuscript, Lin et al describe an artificial protein modulator (APROM) that exhibits protein phosphatase-like activity (and apparently also catalase-like activity). A major part of the work deals with the development of APROM. The authors end the manuscript by proposing that APROM can be used to mitigate alpha-synuclein toxicity. The idea might be that APROM dephosphorylates phosphorylated Serine 129 alpha-synuclein and could on top of that scavenge free radicals due to its catalase activity. Overall, the development of the APROM seems like a good achievement from a chemist's point of view. In terms of its strategic use conceptually and experimentally, there is less enthusiasm. The results of using the new tool in biological systems, as presented in the manuscript, beg more questions than answering anything significant. The following comments may, however, be helpful for the authors to prepare their study for a specialized journal.

Response: *Thank you very much for your valuable comments. Based on your constructive suggestions, we have meticulously made point-to-point responses and heavily revised the manuscript with additional experiments. We believe that your comments have significantly improved the quality of our manuscript.*

1. Protein phosphatase superfamily comprises several members that efficiently modulate protein functions by closely working with kinases during dynamic reversible phosphorylation. It's somewhat unclear whether APROM mimics any or how many of the phosphatases. A detailed characterization of APROM, along with various protein phosphatases, will provide valuable information.

Response: *Thank you for your insightful comments. We appreciate your attention to the nuances of the protein phosphatase superfamily. As you rightly pointed out, this superfamily, encompassing various members that intricately regulate protein functions through reversible phosphorylation, has been traditionally classified into protein serine/threonine (Ser/Thr) phosphatases (PSPs) and protein tyrosine phosphatases (PTPs) based on the amino acidic residues they catalyze (references: Moll. Cell 65, 347-360 (2017); Biochim. Biophys. Acta Rev. Cancer 1876, 188562 (2021); Cancer Lett. 335, 9-18 (2013)). PSPs, further divided into phosphoprotein phosphatases (PPPs), metal-dependent protein phosphatases (PPMs), and aspartate-based phosphatases, all share the common mechanism of dephosphorylating phospho-Ser/Thr through cleaving phosphate monoester bonds (reference: Cell 139, 468-484 (2009)). Similarly, PTPs, grouped into four classes, remove phosphate groups from tyrosine residues through phosphate monoester bond cleavage (reference: Cell 117, 699-711 (2004)).*

Considering the catalytic activity of cerium ions in binding with phosphate groups and hydrolyzing phosphate monoester bonds of phospho-substrates (reference: *Coord. Chem. Rev.* 382, 145-159 (2019)), and the demonstrated ability of ceria-based catalysts to catalytically cleave phosphate monoester bonds (references: *Adv. Sci.* 8, 2004115 (2021); *ACS Catal.* 13, 504-514 (2023)), we undertook a comprehensive evaluation of the protein phosphatase-mimetic activity of the artificial protein modulator (APROM). Using *O*-phospho-*L*-serine (*P*-Ser), *O*-phospho-*L*-tyrosine (*P*-Tyr), and *O*-phospho-*L*-threonine (*P*-Thr) as phospho-substrates (Fig. 3j,q), we observed that the APROM indeed catalytically cleaves the phosphate monoester bonds of *P*-Ser, *P*-Tyr, and *P*-Thr, demonstrating robust dephosphorylation activity. These results collectively affirm that the APROM effectively emulates the functions of protein phosphatases, aligning with the activities of both PSPs and PTPs. We hope this clarification addresses your concerns and enhances the understanding of the APROM's catalytic capabilities.

Our modification to the manuscript: *The following sentences were added on page 11, and the protein phosphatase-mimetic activity of APROM₂ by using P-Tyr and P-Thr as phospho-substrates was added as Fig. 3q in the revised manuscript.*

- Page 11

“.....Moreover, APROM₂ exhibits pronounced dephosphorylation activity towards *O*-phospho-*L*-tyrosine (*P*-Tyr) and *O*-phospho-*L*-threonine (*P*-Thr), effectively cleaving the phosphate monoester bonds (Fig. 3q). This collective evidence solidifies the notion that APROM₂ faithfully emulates the functionality of protein phosphatases, encompassing both protein serine/threonine phosphatases and protein tyrosine phosphatases.”

• Figure 3

Figure 3. Asymmetric catalytic centers confer protein phosphatase-like characteristics to APROMs. **a,b**, Local coordination environments and corresponding differential charge density maps of OH absorbed CeNPs (**a**) and APROM (**b**). Red and blue represent accumulation and

depletion charge areas, respectively. **c**, Free-energy diagram of phospho-substrate hydrolysis on the APROM and CeNPs. **d,e**, Differential charge density maps of OH and phospho-substrate absorbed CeNPs (**d**) and APROM (**e**). **f**, Bader charge of the O atom of OH before and after phospho-substrate absorption. A positive value indicates that the atom gains electrons. **g**, DOS of CeNPs and the APROM. E = energy level, E_{Fermi} = Fermi level. **h,i**, PDOS of CeNPs (**h**) and the APROM (**i**). **j**, Protein phosphatases-mimetic activity of APROMs and CeNPs by using P-Ser as the phospho-substrate (n = 3 independent experiments). **k**, Raman spectra of APROM₂ with and without incubation of P-Ser. The P-Ser is used as the phospho-substrate for the dephosphorylation reaction. F_{2g} , the Raman-active vibrational mode of the cubic fluorite structure. **l**, The Ce-to-Mn ratio of APROM₂ before and after catalyzing phospho-substrate dephosphorylation. **m**, *In situ* XRD of APROM₂ under dephosphorylation reaction conditions. **n**, Protein phosphatases-mimetic activity of APROM₂ by using p- α -syn as the phospho-substrates (n = 3 independent experiments). **o,p**, Michaelis-Menten kinetics (**o**) and Lineweaver-Burk plotting (**p**) of APROM₂ obtained by adding different concentrations of p- α -syn (n = 3 independent experiments). **q**, Protein phosphatases-mimetic activity of APROM₂ by using P-Tyr and P-Thr as the phospho-substrates (n = 3 independent experiments). All the data are presented as means \pm s.e.m. Statistical significance was analyzed by one-way ANOVA with multiple comparisons test.

2. It should be explained better why this protein phosphatase also possesses a catalase-like activity. Is APROM closer to an enzyme that scavenges free radicals or a phosphatase? While it can theoretically be viewed as an advantage that APROM both dephosphorylates aSyn and scavenges free radicals, the big concern is that APROM is not only a phosphatase of undefined specificity, but may - beyond catalase activity – have additional pleiotropic effects. This raises critical questions about its application and this limitation must be highlighted.

Response: Thank you for your insightful comments. APROM, an artificial protein phosphatase, is intricately designed by integrating site-specific single manganese atoms into heterogeneous ceria nanoparticulate catalysts (Mn/CeO_{2-x}) through a self-aggravating surface oxygen vacancy (SOV)-driven cation exchange reaction. The resulting asymmetric catalytic centers of APROM comprise SOVs, Mn ions, and the reversible Ce³⁺/Ce⁴⁺ redox pair (Fig. 2c-e).

As highlighted in your comments, the reversible Ce³⁺/Ce⁴⁺ redox pair in APROM grants it catalase (CAT)-mimetic activity (Fig. 4d-f). This activity involves a series of reactions (Eq. 1 and Eq. 2) where H₂O₂ is decomposed into O₂ and H₂O (reference: *Adv. Mater.* 2210819 (2023)). Notably, due to the presence of Mn in the asymmetric catalytic centers, APROM outperforms CeNPs in CAT-mimetic activity. This phenomenon has been further elucidated through density functional theory (DFT) calculations (Fig. 4g), revealing a distinctive O₂ desorption mechanism in APROM, which contributes to its superior CAT-mimetic activity

compared to CeNPs.

Regarding the comparison with natural phosphatases, our study draws inspiration from the asymmetric unit concept at catalytic centers (references: *Nature* 376, 745-753 (1995); *Catalysts* 6, 60 (2016); *J. Mol. Biol.* 415, 102-117 (2012)). By precisely incorporating site-specific single manganese atoms, we emulate this asymmetric structure, leading to enhanced protein phosphatase-mimetic activity, as evidenced in Fig. 3. It is important to note that APROM, while closely resembling natural phosphatases, also exhibits reactive oxygen species (ROS) scavenging capability, a property intricately tied to its atomic structure and electronic coordination environments.

As you rightly point out, the multifunctional nature of artificial catalysts is not uncommon, and APROM is no exception. It demonstrates pleiotropic effects, combining the dephosphorylation of hyperphosphorylated proteins with ROS scavenging. This dual action is particularly advantageous for diseases like Parkinson's disease (PD), where both hyperphosphorylated α -synuclein (p - α -syn) and oxidative stress contribute to pathology. Beyond the protein phosphatase-mimetic and CAT-mimetic activity, APROM also exerts superoxide dismutase (SOD)-mimetic activity because of its reversible $\text{Ce}^{3+}/\text{Ce}^{4+}$ redox pair, catalyzing the dismutation of superoxide radicals (O_2^-) into O_2 and H_2O_2 (Fig. 4a). Through the synergistic interplay of SOD-mimetic and CAT-mimetic activity, APROM efficiently eliminates ROS, alleviating oxidative stress. The ROS scavenging capability of APROM is concentration-dependent, allowing it to modulate its antioxidant activity based on the local ROS levels in the pathological microenvironment. Importantly, APROM does not elicit any additional effects. Notably, in the presence of H_2O_2 , APROM does not generate hydroxyl radicals through Fenton-like reactions, as illustrated in Supplementary Fig. 14. This is of significance, as hydroxyl radicals are recognized for their potential to induce damage to diverse biomolecules, encompassing DNA, proteins, and lipids.

In summary, the pleiotropic effects of APROM, encompassing both dephosphorylation and ROS scavenging, strategically position it as a promising candidate for diseases marked by abnormal protein phosphorylation and oxidative stress. This versatility extends its potential utility across diverse pathological scenarios. However, it is crucial to acknowledge the limitations of APROM. Notably, its suitability may be restricted for diseases lacking abnormal protein phosphorylation. Moreover, conditions linked to abnormal protein phosphorylation that rely on ROS production in the pathological microenvironment, such as certain tumor contexts, may not align optimally with APROM's mechanisms. We appreciate your thoughtful consideration and suggestions.

Our modification to the manuscript: *The following sentences were modified on pages 11-12 and added on the discussion, and the reference was added in the revised manuscript.*

- Pages 11-12

“Ascribed from the reversible $\text{Ce}^{3+}/\text{Ce}^{4+}$ redox pair in asymmetric catalytic centers⁴³ (Fig. 2d), APROMs simultaneously possess antioxidant properties, including catalase (CAT)-mimetic and superoxide dismutase (SOD)-mimetic activities, corresponding to the multi-enzymatic activities achieved by artificial catalysts instead of natural enzymes so as to exert synergistic effect²¹. Remarkably, the SOD-mimetic activity of APROMs is enhanced, with APROM₂ performing best in scavenging superoxide radicals (O_2^-), consistent with the high Ce^{3+} -to- Ce^{4+} ratio induced by asymmetric catalytic centers (Fig. 4a). More importantly, the SOD-mimetic activity of APROM₂ is superior to that of natural SOD (Fig. 4b,c, Supplementary Table 4). Notably, different from the downregulated CAT-mimetic activity affected by decreasing surface Ce^{4+} levels⁴⁴, APROMs exhibit superior CAT-mimetic activity in decomposing H_2O_2 into H_2O and O_2 compared to CeNPs, despite a lower Ce^{4+} -to- Ce^{3+} ratio, which possibly due to the presence of asymmetric catalytic centers (Fig. 4d).”

- Discussion

“Our findings offer a promising avenue for reprogramming protein function using the *de novo* PTM strategy, which involves the dephosphorylation of hyperphosphorylated proteins by APROMs, with the potential to revolutionize the treatment of diseases, particularly neurological disorders. Indeed, while APROM demonstrates a broad spectrum of biological applications, it is crucial to acknowledge its limitations. Specifically, APROM may not be suitable for the treatment of diseases unrelated to abnormal protein phosphorylation. Additionally, it may not be optimally suited for conditions associated with abnormal protein phosphorylation that specifically require ROS production in the pathological microenvironment, such as certain tumor scenarios. Future advancements focusing on specific protein-targeted APROMs enable precise rebalancing of aberrant PTMs of the proteins that contain phosphate monoester bonds at amino acid residues like serine, tyrosine and threonine within affected areas, thereby reprogramming protein function and maintaining proteostasis for effective disease therapy. Additionally, the utilization of probe-modified APROMs will facilitate the study of protein-to-protein interaction patterns and the investigation of biological functions associated with reverse-modified proteins in cellular homeostasis.”

- Reference

43. Lee, J. et al. Exploration of nanozymes in viral diagnosis and therapy. *Exploration* **2**, 20210086 (2022).

3. Dynamic reversible phosphorylation of proteins coordinated by kinases and phosphatases can be crucial for a given protein's functions. In pathological states, dynamic reversible phosphorylation may have been affected. It would thus be important to bring back the conditions to non-diseased states. If decreasing abnormal phosphorylation is a key therapeutic

approach, we would have a meaningful treatment for synucleinopathies, as the authors claim. However, the role of α Syn phosphorylation in PD pathology is not that clear - and detrimental, protective and neutral roles have all been proposed. A recent publication (DOI: 10.1073/pnas.2109617119) suggests that serine-129 phosphorylation is a late epi-phenomenon that is not involved in aggregation (or toxicity of α Syn).

It is to be noted that hyper- or hypo-phosphorylation can equally be bad for a given biological process. APROM phosphatase activity does not seem to be specific. What if APROM dephosphorylated important PTMs in proteins required for normal biological processes? More details below.

Response: *In the recent publication (DOI: 10.1073/pnas.2109617119), the authors employed α -syn preformed fibrils (PFFs) to induce Parkinson's disease (PD) in mice. Their findings have revealed that α -syn phosphorylation at serine 129 (pS129- α -syn) occurs as a secondary event to the initial aggregation of nonphosphorylated S129- α -syn (WT- α -syn). These experimental outcomes, however, are specifically applied in the model constructed by PFFs which is different from the PD mouse model induced by compounds as acknowledged by the authors (J. Biol. Chem. 283, 23179-23188 (2008)). Since 1-methyl-4-phenyl-1,2,3,6-tetrahydropyridine (MPTP) is the only known dopaminergic neurotoxin capable of causing a clinical picture in both humans and monkeys indistinguishable from PD (reference: Nat. Protoc. 2, 141-151 (2007)), we opted for compound 1-methyl-4-phenylpyridinium (MPP⁺, metabolite of MPTP) and MPTP to establish the PD phenotype cell and PD mice models, respectively, deviating from the approach employed in the aforementioned publication (DOI: 10.1073/pnas.2109617119). Notably, in our investigation, we observed a significant increase in the p- α -syn level in the immunofluorescence (Fig. 5b,c, Fig. 6i, Supplementary Fig. 22) and Western blot analysis (Supplementary Fig. 17). Crucially, we confirmed that a direct binding between α -syn and VMAT2 depends on the dephosphorylation (Fig. 6k,l, Supplementary Fig. 23). Hence, dephosphorylating p- α -syn is pivotal for restoring its biological functions, and we present a novel strategy for this dephosphorylation.*

In the physiological microenvironment, reversible protein phosphorylation is a dynamic process occurring on timescales ranging from seconds to minutes. Remarkably, the kinetics of protein-protein interactions surpass those of phosphorylation/dephosphorylation reactions, facilitating rapid cellular responses to external stimuli, regulation of signal transduction pathways, and maintenance of cellular homeostasis. Additionally, the dephosphorylation rates of APROM are contingent upon the specific types of proteins involved, each displaying distinct affinities for APROM. In PD, the compromised catalytic activity of protein phosphatases responsible for dephosphorylating p- α -syn leads to the extensive accumulation of p- α -syn. This sustained phosphorylation state enhances interactions with APROM. Moreover, the C-terminal region (residues 96-140) of α -syn, containing motifs of charged amino acids, can bind to various metal ions, including Mn, Fe, and Cu (reference: J. Am. Chem. Soc. 128, 9893-9901

(2006)). Phosphorylation at Ser 129 induces alterations in the charge, hydrogen-binding patterns, and backbone angles of the C-terminal region, intensifying the affinity between metal ions and the C-terminal region of α -syn (reference: J. Parkinson. Dis. 6, 39-51 (2016)). Consequently, APROM is more likely to selectively bind with p- α -syn for dephosphorylation without significantly impacting the functionality of other proteins essential for normal biological processes. Phosphoproteomic analyses indicate that, although APROM₂ demonstrates protein-phosphatase-mimetic activity, it does not affect the phosphorylation level of normal biological process-related proteins in the midbrain (Supplementary Figs. 29-33), including amino acid metabolism, carbohydrate metabolism, energy metabolism, lipid metabolism, and nucleotide metabolism.

Our modification to the manuscript: The following sentences were added on pages 13, 16 and 17 in the revised manuscript. The results of co-immunoprecipitation were added as Fig. 6k,l in the revised manuscript, and added as Supplementary Fig. 24 in the revised supporting information. The Western blot analysis of p- α -syn was added as Supplementary Fig. 17, the quantitative analysis of phosphorylation degree was added as Supplementary Fig. 22, and the phosphoproteomic analyses were added as Supplementary Figs. 30-34 in the revised supporting information.

- Page 13

“.....In subsequent experiments, we established a primary neuronal model with abnormal PTMs by treating neurons with the neurotoxin 1-methyl-4-phenylpyridinium (MPP⁺). Upon MPP⁺ exposure, the intracellular p- α -syn level significantly increases (Fig. 5b,c, Supplementary Fig. 17). The C-terminal region of α -syn, which contains motifs of charged amino acids, is capable of binding to metal ions⁴⁶. Notably, phosphorylation of Ser 129 within the C-terminal region significantly enhances this affinity⁴⁷. In line with these observations, APROM₂ remarkably dephosphorylates p- α -syn with decreasing the p- α -syn level approximate to the physiological level via effective interaction with p- α -syn, thereby rebalancing the aberrant PTMs. In contrast, although CeNPs demonstrate some reduction in the p- α -syn level in primary neurons, aberrant PTMs of α -syn still persist.”

- Page 16

“.....APROM₂ effectively reverses the hyperphosphorylation state of α -syn at Ser-129 in both the SN and striatum (ST) (Fig. 6i, Supplementary Figs. 22 and 23). This ability highlights the capacity of APROM₂ to rebalance the abnormal PTMs of α -syn and consequently inhibit the subsequent formation of toxic α -syn inclusions (Fig. 6j, Supplementary Figs. 22). More importantly, through APROM₂'s *de novo* PTM strategy, the biological function of α -syn is regained, as verified by the promoted colocalization of α -syn with VMAT2 in dopaminergic neurons that is dependent on dephosphorylation (Fig. 6k-m, Supplementary Fig. 24), as well as colocalization of α -syn with VAMP2 (Fig. 6n).”

- Page 17

“.....Notably, while APROM₂ demonstrates protein-phosphatase-mimetic activity, it does not affect the phosphorylation levels of metabolism-related proteins in the midbrain as verified by phosphoproteomic analysis (Supplementary Figs. 30-34), including amino acid metabolism, carbohydrate metabolism, energy metabolism, lipid metabolism, and nucleotide metabolism. These results demonstrate that APROM₂ exhibits excellent biocompatibility and does not induce local or systemic toxicity.”

• Figure 6

Figure 6. APROM₂ mediated synaptic plasticity improvement *in vivo*. **a**, Schematic illustration of APROM₂ that protects dopaminergic neurons against neurodegeneration for rescuing motor coordination in PD. **b-d**, Behavioral evaluation of pole test (**b**) rotarod test (**c**) and hang test (**d**) of mice after different treatments (n = 9 biologically independent mice). **e,f**, Quantitative analysis (**e**) and representative immunohistochemical staining images (**f**) of TH positive neurons in the SN (n = 3 biologically independent mice). Regions of interest (white square) in the top panels are shown at higher magnification in the bottom panels. **g**, Representative immunohistochemical staining images of dopaminergic fibers in the ST. Regions of interest (white square) in the top panels are shown at higher magnification in the bottom panels. **h**, Schematic illustration of APROM₂ that improves synaptic plasticity of dopaminergic neurons in PD. **I**, hyperphosphorylation of α -syn and oxidative stress impair synaptic function of dopaminergic neurons. **II**, APROM₂ dephosphorylates p- α -syn via protein phosphatase-mimetic activity and scavenges ROS via antioxidant activity. **III**, APROM₂ reprograms α -syn biological function via the *de novo* PTM strategy and alleviates oxidative stress for fueling synaptic function. **i**, Representative immunofluorescence staining images of p- α -syn in the SN and ST. **j**, Representative immunohistochemical staining images of α -syn inclusion in the SN. Inset, higher magnification of dopaminergic neurons. **k,l**, Co-immunoprecipitation of VMAT2 with α -syn (**k**) or p- α -syn (**l**) in the midbrain of APROM₂ treated PD mice. **m,n**, Representative immunofluorescence staining images and colocalization analysis along the white line of α -syn with VMAT2 (**m**) or VAMP2 (**n**). **o**, Bio-TEM images of synaptic vesicles in the SN of mice after different treatments. Regions of interest (red square) in the top panels are shown at a higher magnification in the bottom panels. **p**, Representative immunofluorescence staining images of 4-HNE. **q**, Schematic of the midbrain section and histological assay with H&E staining for the SN and ST of APROM₂ treated mice. All the data are presented as means \pm s.e.m. Statistical significance was analyzed by one-way ANOVA with multiple comparisons test.

- Supplementary Figure 17

Supplementary Figure 17. Western blot analysis of p- α -syn in primary neurons with different treatments.

- Supplementary Figure 24

Supplementary Figure 24. Co-immunoprecipitation of α -syn with VMAT2 in the midbrain of APROM₂ treated PD mice.

- Supplementary Figure 30

Supplementary Figure 30. The heatmap of amino acid metabolism related phospho-proteins.

- Supplementary Figure 31

Supplementary Figure 31. The heatmap of carbohydrate metabolism related phosphoproteins.

- Supplementary Figure 32

Supplementary Figure 32. The heatmap of energy metabolism related phospho-proteins.

- Supplementary Figure 33

Supplementary Figure 33. The heatmap of lipid metabolism related phospho-proteins.

- Supplementary Figure 34

Supplementary Figure 34. The heatmap of nucleotide metabolism related phospho-proteins.

4. The rationale for testing APROM's application in synucleinopathies lacks a rationale. Anyway, it will be necessary first to test whether APROM dephosphorylates Serine 129 alpha-synuclein directly in vitro. And how comparable is its activity to PP2A?

Response: *Thank you for your insightful comments. We appreciate the thorough evaluation of our manuscript. In response to your suggestion, we conducted experiments to specifically assess the protein phosphatase-mimetic activity of APROM₂ using p-α-syn phosphorylated at Serine 129 as the phospho-substrate. As anticipated, APROM₂ exhibited a concentration-dependent dephosphorylation of p-α-syn.*

*The kinetic parameters for APROM₂ in this dephosphorylation process were determined as follows: the Michaelis-Menten constant (K_m) is 0.175 mM, the maximum velocity (V_{max}) is 0.0015 $\mu\text{M s}^{-1}$, and the turnover number (k_{cat}) is 0.764 s^{-1} . To provide context, studies have reported k_{cat} values for the dephosphorylation of various phospho-substrates by PP2A ranging from 0.005 to 594 s^{-1} (references: *eLife* 3, e01695 (2014); *Moll. Cell* 23, 757-764 (2006); *Curr. Med. Chem.* 26, 2634-2660 (2019); *Bull. Korean Chem. Soc.* 35, 3385-3388 (2014); *Biochem. J.* 357, 225-232 (2001)). Remarkably, APROM₂'s k_{cat} for p-α-syn dephosphorylation is within this range and comparable to that of natural PP2A, underscoring its effectiveness in catalyzing the dephosphorylation of p-α-syn.*

We believe that these findings strengthen the manuscript and further support the proposition that APROM₂ exhibits protein phosphatase-mimetic activity comparable to natural phosphatases.

Our modification to the manuscript: *The following sentences were added on page 11, and the steady-state kinetics results of APROM₂ were added as Fig. 3o,p in the revised manuscript. The steady-state kinetics results of APROM₂ were added as Supplementary Table 3 in the revised supporting information.*

- Page 11

*“.....Inspired by the remarkable dephosphorylation capability of APROM₂, we further investigated its protein phosphatase-mimetic activity using p-α-syn as the phospho-substrate. As expected, APROM₂ readily dephosphorylates p-α-syn in a concentration-dependent manner (Fig. 3n), and the K_m , V_{max} , k_{cat} of APROM₂ are 0.175 mM, 0.0015 $\mu\text{M s}^{-1}$, and 0.764 s^{-1} (Fig. 3o,p, Supplementary Table 3), respectively. This suggests that APROM₂ can efficiently dephosphorylate p-α-syn, converting it into α-syn through *de novo* PTMs.”*

• Figure 3

Figure 3. Asymmetric catalytic centers confer protein phosphatase-like characteristics to APROMs. **a,b**, Local coordination environments and corresponding differential charge density maps of OH absorbed CeNPs (**a**) and APROM (**b**). Red and blue represent accumulation and

depletion charge areas, respectively. **c**, Free-energy diagram of phospho-substrate hydrolysis on the APROM and CeNPs. **d,e**, Differential charge density maps of OH and phospho-substrate absorbed CeNPs (**d**) and APROM (**e**). **f**, Bader charge of the O atom of OH before and after phospho-substrate absorption. A positive value indicates that the atom gains electrons. **g**, DOS of CeNPs and the APROM. E = energy level, E_{Fermi} = Fermi level. **h,i**, PDOS of CeNPs (**h**) and the APROM (**i**). **j**, Protein phosphatases-mimetic activity of APROMs and CeNPs by using P-Ser as the phospho-substrate (n = 3 independent experiments). **k**, Raman spectra of APROM₂ with and without incubation of P-Ser. The P-Ser is used as the phospho-substrate for the dephosphorylation reaction. F_{2g} , the Raman-active vibrational mode of the cubic fluorite structure. **l**, The Ce-to-Mn ratio of APROM₂ before and after catalyzing phospho-substrate dephosphorylation. **m**, *In situ* XRD of APROM₂ under dephosphorylation reaction conditions. **n**, Protein phosphatases-mimetic activity of APROM₂ by using p- α -syn as the phospho-substrates (n = 3 independent experiments). **o,p**, Michaelis-Menten kinetics (**o**) and Lineweaver-Burk plotting (**p**) of APROM₂ obtained by adding different concentrations of p- α -syn (n = 3 independent experiments). **q**, Protein phosphatases-mimetic activity of APROM₂ by using P-Tyr and P-Thr as the phospho-substrates (n = 3 independent experiments). All the data are presented as means \pm s.e.m. Statistical significance was analyzed by one-way ANOVA with multiple comparisons test.

- Supplementary Table 3

Supplementary Table 3. Kinetics parameters of APROM₂ for protein phosphatase-mimetic activity.

Catalyst	Substrate	K_m (mM)	V_{max} ($\mu\text{M s}^{-1}$)	k_{cat} (s^{-1})
APROM ₂	P-Ser	39.17	0.49	2.47×10^2
APROM ₂	p- α -syn	0.175	1.5×10^{-3}	0.764

K_m is the Michaelis-Menten constant, V_{max} is the maximal reaction velocity, and k_{cat} is the catalytic constant.

5. If the authors wish to study the effects of APROM on alpha-synuclein's synaptic function better, they should consider working with experts in the field. All results described in Figure 5 are sub-standard quality per the current synuclein research. Based on the data, the phrase "APROM reprograms the biological function of neuronal proteins to fuel synaptic function via de nova PTMs" is overstated and premature. Having someone who worked with cultured neurons for several years, I wouldn't dare to say that the maturation of primary neurons requires only 7-8 days. The primary dopaminergic neuron culture the authors described in the methods appears not standard in the field. A proper QC needs to be done before doing any experiments with this culture. Again, I believe working with experts will help the authors.

Response: Thank you for bringing up this important point. We appreciate your attention to detail and the importance of standardized procedures in neuronal cell culture. As mentioned

in our Methods section, we adopted a modified version of the primary neuron preparation method, as previously reported (references: Mol Cell 33, 627-638 (2009); Brit. J. Pharmacol. 175, 631-643 (2018)), to suit the specific requirements of our experimental design. We acknowledge that the maturation period for primary neurons is a critical factor in experimental design. To ensure the robustness and reliability of our results, we followed established protocols for primary neuron culture duration. The maturation of primary neurons was required a period of 7-8 days in vitro (DIV), and then primary neurons were used for treatments (references: Brit. J. Pharmacol. 175, 631-643 (2018); Mol. Psychiatry 11, 1116-1125 (2006)). In accordance with established protocols and the prevailing consensus in the field, the experiments, including the assessment of cellular ROS-scavenging capability, mitochondrial protective effect, and microscopy images of primary neurons, were conducted over a period of 7-8 DIV. Additionally, experiments measuring phosphorylation levels, synaptic activity and colocalization analysis were carried out after an additional week of culture to facilitate further synapse maturation. We have modified the Methods section for more clarity. This timeframe aligns with widely accepted practices and has been substantiated by various studies (references: Brit. J. Pharmacol. 175, 631-643 (2018); ACS Nano 14, 1533-1549 (2020); Nano Today 36, 101027 (2021); Mol. Psychiatry 11, 1116-1125 (2006); J. Neurosci. Meth. 161, 75-87 (2007)). This adaptation has been validated with the expertise and input of Professor Tianle Xu, our colleague at Shanghai Jiao Tong University, who specializes in this field.

To address concerns about the phrase “APROM reprograms the biological function of neuronal proteins to fuel synaptic function via de novo PTMs”, we have conducted numerous additional experiments. Co-immunoprecipitation of α -syn or p- α -syn with VMAT2 in the midbrain of APROM₂ treated PD mice reveals that α -syn is co-immunoprecipitated with VMAT2 rather than p- α -syn, proving that α -syn can bind directly to VMAT2 and this interaction is dependent on dephosphorylation (Fig. 6k,l, Supplementary Fig. 24). Moreover, APROM₂ efficiently dephosphorylates p- α -syn in vitro and in vivo (Fig. 3n-p, Fig. 5b,c, Fig. 6i, and Supplementary Fig. 22), converting it into α -syn. Notably, the colocalization of α -syn with VMAT2 and VAMP2 is promoted (Fig. 5d-g, Fig. 6m,n), the synaptic transmission is enhanced (Fig. 5h,i), the synaptic vesicles are significantly recovered (Fig. 6o), and the dopamine level of PD mice return to normal physiological levels (Supplementary Fig. 25).

We believe that these findings strengthen the manuscript, providing robust evidence to support the phrase that APROM reprograms the biological function of neuronal proteins to fuel synaptic function via de novo PTMs.

Our modification to the manuscript: *The following sentences were modified on pages 11 and 16, and the method was modified in the revised manuscript. The results of APROM₂'s protein phosphatase-mimetic activity by using p- α -syn as the phospho-substrate were added as Fig. 3n-p in the revised manuscript. The quantitative analysis of the phosphorylation degree was added as Supplementary Fig. 22 in the revised supporting information. The results of co-*

immunoprecipitation were added as Fig. 6k,l in the revised manuscript, and added as Supplementary Fig. 24 in the revised supporting information, respectively. The bio-TEM images were added as Fig. 6o in the revised manuscript. The dopamine level measurement was added as Supplementary Fig. 25 in the revised supporting information.

- Page 11

“.....Inspired by the remarkable dephosphorylation capability of APROM₂, we further investigated its protein phosphatase-mimetic activity using p- α -syn as the phospho-substrate. As expected, APROM₂ readily dephosphorylates p- α -syn in a concentration-dependent manner (Fig. 3n), and the K_m , V_{max} , k_{cat} of APROM₂ are 0.175 mM, 0.0015 μ M s⁻¹, and 0.764 s⁻¹ (Fig. 3o,p, Supplementary Table 3), respectively. This suggests that APROM₂ can efficiently dephosphorylate p- α -syn, converting it into α -syn through *de novo* PTMs.”

- Page 16

“.....APROM₂ effectively reverses the hyperphosphorylation state of α -syn at Ser-129 in both the SN and striatum (ST) (Fig. 6i, Supplementary Figs. 22 and 23). This ability highlights the capacity of APROM₂ to rebalance the abnormal PTMs of α -syn and consequently inhibit the subsequent formation of toxic α -syn inclusions (Fig. 6j, Supplementary Fig. 22). More importantly, through APROM₂'s *de novo* PTM strategy, the biological function of α -syn is regained, as verified by the promoted colocalization of α -syn with VMAT2 in dopaminergic neurons that is dependent on dephosphorylation (Fig. 6k-m, Supplementary Fig. 24), as well as colocalization of α -syn with VAMP2 (Fig. 6n). Moreover, synaptic vesicles are significantly recovered with the treatment of APROM₂ (Fig. 6o), and the dopamine level of PD mice return to normal physiological levels (Supplementary Fig. 25). This finding indicates the reestablishment of synaptic function.”

- Methods

“**Cell culture.**The midbrains of mice within 24 h of birth were harvested quickly on a cold stage, and then digested by TrypLE™ Express containing 1 mg/mL DNase I at 37 °C for 12 min. Subsequently, the cell suspension was passed through filters and centrifuged at 200 × g for 5 min. The cells were resuspended in DMEM/F12 medium supplemented with 10% FBS, and plated onto PLL coated plates at 37 °C and 5% CO₂ atmosphere for 4 h. Then, the culture medium was replaced by Neurobasal-A medium supplemented with 2% B27. The maturation of primary neurons required 7-8 days *in vitro* (DIV) with medium changes every 2-3 days. The experiments, including the assessment of cellular ROS-scavenging capability, mitochondrial protective effect, and microscopy images of primary neurons, were conducted over a period of 7-8 DIV. Additionally, experiments measuring phosphorylation levels, synaptic activity and colocalization analysis were carried out after an additional week of culture.”

• Figure 3

Figure 3. Asymmetric catalytic centers confer protein phosphatase-like characteristics to APROMs. **a,b**, Local coordination environments and corresponding differential charge density maps of OH absorbed CeNPs (**a**) and APROM (**b**). Red and blue represent accumulation and

depletion charge areas, respectively. **c**, Free-energy diagram of phospho-substrate hydrolysis on the APROM and CeNPs. **d,e**, Differential charge density maps of OH and phospho-substrate absorbed CeNPs (**d**) and APROM (**e**). **f**, Bader charge of the O atom of OH before and after phospho-substrate absorption. A positive value indicates that the atom gains electrons. **g**, DOS of CeNPs and the APROM. E = energy level, E_{Fermi} = Fermi level. **h,i**, PDOS of CeNPs (**h**) and the APROM (**i**). **j**, Protein phosphatases-mimetic activity of APROMs and CeNPs by using P-Ser as the phospho-substrate (n = 3 independent experiments). **k**, Raman spectra of APROM₂ with and without incubation of P-Ser. The P-Ser is used as the phospho-substrate for the dephosphorylation reaction. F_{2g} , the Raman-active vibrational mode of the cubic fluorite structure. **l**, The Ce-to-Mn ratio of APROM₂ before and after catalyzing phospho-substrate dephosphorylation. **m**, *In situ* XRD of APROM₂ under dephosphorylation reaction conditions. **n**, Protein phosphatases-mimetic activity of APROM₂ by using p- α -syn as the phospho-substrates (n = 3 independent experiments). **o,p**, Michaelis-Menten kinetics (**o**) and Lineweaver-Burk plotting (**p**) of APROM₂ obtained by adding different concentrations of p- α -syn (n = 3 independent experiments). **q**, Protein phosphatases-mimetic activity of APROM₂ by using P-Tyr and P-Thr as the phospho-substrates (n = 3 independent experiments). All the data are presented as means \pm s.e.m. Statistical significance was analyzed by one-way ANOVA with multiple comparisons test.

• Figure 6

Figure 6. APROM₂ mediated synaptic plasticity improvement *in vivo*. **a**, Schematic illustration of APROM₂ that protects dopaminergic neurons against neurodegeneration for rescuing motor coordination in PD. **b-d**, Behavioral evaluation of pole test (**b**) rotarod test (**c**) and hang test (**d**) of mice after different treatments (n = 9 biologically independent mice). **e,f**, Quantitative analysis (**e**) and representative immunohistochemical staining images (**f**) of TH positive neurons in the SN (n = 3 biologically independent mice). Regions of interest (white square) in the top panels are shown at higher magnification in the bottom panels. **g**, Representative immunohistochemical staining images of dopaminergic fibers in the ST. Regions of interest (white square) in the top panels are shown at higher magnification in the bottom panels. **h**, Schematic illustration of APROM₂ that improves synaptic plasticity of dopaminergic neurons in PD. I, hyperphosphorylation of α -syn and oxidative stress impair synaptic function of dopaminergic neurons. II, APROM₂ dephosphorylates p- α -syn via protein phosphatase-mimetic activity and scavenges ROS via antioxidant activity. III, APROM₂ reprograms α -syn biological function via the *de novo* PTM strategy and alleviates oxidative stress for fueling synaptic function. **i**, Representative immunofluorescence staining images of p- α -syn in the SN and ST. **j**, Representative immunohistochemical staining images of α -syn inclusion in the SN. Inset, higher magnification of dopaminergic neurons. **k,l**, Co-immunoprecipitation of VMAT2 with α -syn (**k**) or p- α -syn (**l**) in the midbrain of APROM₂ treated PD mice. **m,n**, Representative immunofluorescence staining images and colocalization analysis along the white line of α -syn with VMAT2 (**m**) or VAMP2 (**n**). **o**, Bio-TEM images of synaptic vesicles in the SN of mice after different treatments. Regions of interest (red square) in the top panels are shown at a higher magnification in the bottom panels. **p**, Representative immunofluorescence staining images of 4-HNE. **q**, Schematic of the midbrain section and histological assay with H&E staining for the SN and ST of APROM₂ treated mice. All the data are presented as means \pm s.e.m. Statistical significance was analyzed by one-way ANOVA with multiple comparisons test.

- Supplementary Figure 22

Supplementary Figure 22. The quantitative analysis of the p-α-syn level in the SN (a) and ST (b) after different treatments (n = 5 independent experiment). The insoluble fraction levels in the SN (c) and ST (d) after different treatments (n = 5 independent experiment). Data are presented as means ± s.e.m.

- Supplementary Figure 24

Supplementary Figure 24. Co-immunoprecipitation of α-syn with VMAT2 in the midbrain of APROM₂ treated PD mice.

- Supplementary Figure 25

Supplementary Figure 25. The dopamine level in the SN of mice after different treatments (n = 5 biologically independent mice). Data are presented as means ± s.e.m.

6. A more thorough job is needed for the in vivo experiments. First, quantification for TH-positive neurons is required (dopamine levels should be measured). Then, a detailed mechanism needs to be worked out. The cartoon in Fig. 6B is too simple. APROM appears to have a pleiotropic effect. Does APROM work only on alpha-synuclein? It is very unlikely. APROM might work on 100s of proteins dephosphorylating if I am not too generous with the numbers.

Response: *Thank you for your valuable insights. We have conducted a quantitative analysis of TH-positive neurons, revealing a significant preservation of these neurons in the substantia nigra (SN) of PD mice upon APROM₂ treatment, as depicted in Fig. 6e. Moreover, the dopamine level measurement, illustrated in Supplementary Fig. 25, indicate a restoration to normal physiological levels following APROM₂ treatment.*

*Regarding your observation on the mechanism cartoon, we acknowledge the potential confusion and have taken steps to enhance clarity. Specifically, we have modified Fig. 6h and the corresponding figure legend to provide a clearer representation of the detailed mechanism. The presented schematic primarily emphasizes the influence on p- α -syn, as supported by immunofluorescence (Fig. 5b,c, Fig. 6i, Supplementary Figs. 22) and Western blot analysis (Supplementary Fig. 17). It is noteworthy that while APROM₂ exhibits interactions with various proteins, our phosphoproteomic analysis suggests none significant negative impacts on normal biological process-related proteins in the midbrain (Supplementary Figs. 30-34), including amino acid metabolism, carbohydrate metabolism, energy metabolism, lipid metabolism, and nucleotide metabolism. In the physiological microenvironment, reversible protein phosphorylation is a dynamic process, typically unfolding on timescales ranging from seconds to minutes. Notably, the kinetics of protein-protein interactions surpass those of phosphorylation/dephosphorylation reactions, enabling swift cellular responses to external stimuli, the regulation of signal transduction pathways, and the maintenance of cellular homeostasis. Additionally, the dephosphorylation rates of APROM are contingent upon the specific types of proteins involved, each displaying distinct affinities for APROM. Importantly, in Parkinson's disease (PD), the catalytic activity of protein phosphatases that govern the dephosphorylation of p- α -syn becomes compromised, leading to the extensive accumulation of p- α -syn. This sustained phosphorylation state enhances the interactions with APROM. Furthermore, the C-terminal region (residues 96-140) of α -syn, housing motifs of charged amino acids, has the capability to bind to various metal ions, including Mn, Fe, and Cu (reference: *J. Am. Chem. Soc.* 128, 9893-9901 (2006)). Phosphorylation at Ser 129 induces alterations in the charge, hydrogen-binding patterns, and backbone angles of the C-terminal region, intensifying the affinity between metal ions and the C-terminal region of α -syn (reference: *J. Parkinson. Dis.* 6, 39-51 (2016)). Consequently, APROM is more likely to selectively bind with p- α -syn for dephosphorylation without significantly impacting the*

functionality of other proteins essential for normal biological processes. We appreciate your thoughtful consideration and suggestions.

Our modification to the manuscript: *The following sentence was added on pages 13, 16 and 17, and Fig. 6h was modified in the revised manuscript, respectively. The Western blot analysis of p- α -syn was added as Supplementary Fig. 17, the quantitative analysis of p- α -syn levels in the SN and ST was added as Supplementary Fig. 22, and the dopamine level measurement was added as Supplementary Fig. 25, phosphoproteomic analysis was added as Supplementary Figs. 30-34 in the revised supporting information, respectively.*

- Page 13

“.....In subsequent experiments, we established a primary neuronal model with abnormal PTMs by treating neurons with the neurotoxin 1-methyl-4-phenylpyridinium (MPP⁺). Upon MPP⁺ exposure, the intracellular p- α -syn level significantly increases (Fig. 5b,c, Supplementary Fig. 17). The C-terminal region of α -syn, which contains motifs of charged amino acids, is capable of binding to metal ions⁴⁶. Notably, phosphorylation of Ser 129 within the C-terminal region significantly enhances this affinity⁴⁷. In line with these observations, APROM₂ remarkably dephosphorylates p- α -syn with decreasing the p- α -syn level approximate to the physiological level via effective interaction with p- α -syn, thereby rebalancing the aberrant PTMs. In contrast, although CeNPs demonstrate some reduction in the p- α -syn level in primary neurons, aberrant PTMs of α -syn still persist.”

- Page 16

“.....APROM₂ effectively reverses the hyperphosphorylation state of α -syn at Ser-129 in both the SN and striatum (ST) (Fig. 6i, Supplementary Figs. 22 and 23). This ability highlights the capacity of APROM₂ to rebalance the abnormal PTMs of α -syn and consequently inhibit the subsequent formation of toxic α -syn inclusions (Fig. 6j, Supplementary Fig. 22).”

“.....Moreover, synaptic vesicles are significantly recovered with the treatment of APROM₂ (Fig. 6o), and the dopamine level of PD mice return to normal physiological levels (Supplementary Fig. 25). This finding indicates the reestablishment of synaptic function.”

- Page 17

“.....Notably, while APROM₂ demonstrates protein-phosphatase-mimetic activity, it does not affect the phosphorylation levels of metabolism-related proteins in the midbrain as verified by phosphoproteomic analysis (Supplementary Figs. 30-34), including amino acid metabolism, carbohydrate metabolism, energy metabolism, lipid metabolism, and nucleotide metabolism. These results demonstrate that APROM₂ exhibits excellent biocompatibility and does not induce local or systemic toxicity.”

• Figure 6

Figure 6. APROM₂ mediated synaptic plasticity improvement *in vivo*. **a**, Schematic illustration of APROM₂ that protects dopaminergic neurons against neurodegeneration for rescuing motor coordination in PD. **b-d**, Behavioral evaluation of pole test (**b**) rotarod test (**c**) and hang test (**d**) of mice after different treatments (n = 9 biologically independent mice). **e,f**, Quantitative analysis (**e**) and representative immunohistochemical staining images (**f**) of TH positive neurons in the SN (n = 3 biologically independent mice). Regions of interest (white square) in the top panels are shown at higher magnification in the bottom panels. **g**, Representative immunohistochemical staining images of dopaminergic fibers in the ST. Regions of interest (white square) in the top panels are shown at higher magnification in the bottom panels. **h**, Schematic illustration of APROM₂ that improves synaptic plasticity of dopaminergic neurons in PD. **i**, hyperphosphorylation of α -syn and oxidative stress impair synaptic function of dopaminergic neurons. **ii**, APROM₂ dephosphorylates p- α -syn via protein phosphatase-mimetic activity and scavenges ROS via antioxidant activity. **iii**, APROM₂ reprograms α -syn biological function via the *de novo* PTM strategy and alleviates oxidative stress for fueling synaptic function. **iv**, Representative immunofluorescence staining images of p- α -syn in the SN and ST. **v**, Representative immunohistochemical staining images of α -syn inclusion in the SN. Inset, higher magnification of dopaminergic neurons. **vi**, Co-immunoprecipitation of VMAT2 with α -syn (**vii**) or p- α -syn (**viii**) in the midbrain of APROM₂ treated PD mice. **ix**, Representative immunofluorescence staining images and colocalization analysis along the white line of α -syn with VMAT2 (**x**) or VAMP2 (**xi**). **xii**, Bio-TEM images of synaptic vesicles in the SN of mice after different treatments. Regions of interest (red square) in the top panels are shown at a higher magnification in the bottom panels. **xiii**, Representative immunofluorescence staining images of 4-HNE. **xiv**, Schematic of the midbrain section and histological assay with H&E staining for the SN and ST of APROM₂ treated mice. All the data are presented as means \pm s.e.m. Statistical significance was analyzed by one-way ANOVA with multiple comparisons test.

- Supplementary Figure 17

Supplementary Figure 17. Western blot analysis of p- α -syn in primary neurons with different treatments.

- Supplementary Figure 22

Supplementary Figure 22. The quantitative analysis of the p-α-syn level in the SN (a) and ST (b) after different treatments (n = 5 independent experiment). The insoluble fraction levels in the SN (c) and ST (d) after different treatments (n = 5 independent experiment). Data are presented as means ± s.e.m.

- Supplementary Figure 25

Supplementary Figure 25. The dopamine level in the SN of mice after different treatments (n = 5 biologically independent mice). Data are presented as means ± s.e.m.

- Supplementary Figure 30

Supplementary Figure 30. The heatmap of amino acid metabolism related phospho-proteins.

- Supplementary Figure 31

Supplementary Figure 31. The heatmap of carbohydrate metabolism related phosphoproteins.

- Supplementary Figure 32

Supplementary Figure 32. The heatmap of energy metabolism related phospho-proteins.

- Supplementary Figure 33

Supplementary Figure 33. The heatmap of lipid metabolism related phospho-proteins.

- Supplementary Figure 34

Supplementary Figure 34. The heatmap of nucleotide metabolism related phospho-proteins.

7. Simply presenting the APROM development may already be sufficient for a specialized journal. Given the limitations it might pose (as it stands), it will be too difficult to attribute any translational application for APROM.

Response: *Thank you for your feedback and recognition. We believe the revisions incorporated into the manuscript have elevated its overall quality. Your insightful guidance has been instrumental in refining the content.*

Following these enhancements, we are confident that the data presented in the revised manuscript robustly supports the translational application of APROM. We sincerely appreciate your thorough review and constructive comments, which have played a pivotal role in augmenting the manuscript's overall merit. Your commitment to the peer-review process is truly valued, and we are grateful for your time and dedication.

Reviewer #2 (Remarks to the Author):

Response: *We are grateful for your participation as a co-reviewer and the invaluable comments you provided. Based on your kind suggestions, we have meticulously made point-to-point responses to the listed reports from your collaborating reviewer and revised the manuscript. We believe that your generous guidance to this manuscript, alongside the reviewer you collaborated with, has significantly improved the overall quality of our manuscript.*

Reviewer #3 (Remarks to the Author):

Post-translational modifications (PTMs) of proteins are important for cellular functions. Protein phosphatases play a crucial role in the regulation of phosphorylation, a vital PTM. However, this regulation is disrupted in pathological conditions, such as neurodegenerative diseases, which have detrimental effects on neuronal proteins. The development of artificial protein modulators (APROMs) could solve this problem.

This study utilized a newly developed catalyst with a single manganese (Mn) atom, Mn/CeO_{2-x}, which employs a self-weighted surface oxygen vacancy (SOV) as its active center. This catalyst, similar to protein phosphatase, can correct abnormal PTMs in neurons. Furthermore, this catalyst possesses antioxidant properties that may contribute to improved mitochondrial function.

The properties and structure of the catalysts were analyzed in detail using X-ray diffraction (XRD), X-ray photoelectron spectroscopy (XPS), high-resolution transmission electron microscopy (HRTEM), and density functional theory (DFT) calculations, which were performed to gain a deeper understanding of the mechanism of action of the catalyst and SOV formation. The results of these analyses indicate that the bond between Mn and oxygen (O) is shorter than that between Ce and O, which accounts for the high activity of the catalyst.

APROM2 has been demonstrated to have a range of effects, particularly in neurons, including the ability to correct abnormal post-translational modifications (PTMs) and normalize the function of neuronal proteins. Additionally, it has been reported to improve synaptic function, which is critical in progressive neurological diseases, and has neuroprotective effects in animal models of Parkinson's disease (PD) by protecting dopaminergic neurons and enhancing synaptic function. Furthermore, biocompatibility and low toxicity of APROM2 were confirmed. Therefore, APROM2 can be used to treat neurological disorders. It has the ability to rectify abnormal protein phosphorylation and enhance overall neuronal well-being. **These discoveries may profoundly impact the development of novel therapies for neurodegenerative diseases, particularly Parkinson's disease.**

The development of artificial catalysts has been validated, and their efficacy has been determined through a range of methods, including cell tests and immunohistochemistry. However, a more specialized evaluation is required for the first half of Figures 1-4. The peer review results of the second half of Figures 5-6, which encompasses the quantitative analysis of biochemistry and immunohistochemistry, are provided. Further verification is necessary for the therapeutic effects, including the quantitative analysis of biochemistry and quantitative immunohistochemical analysis and the verification of α -synuclein phosphorylation and inclusion body formation through biochemical analysis.

Response: *Thank you for your encouraging comments. Based on your invaluable suggestions, we have meticulously made point-to-point responses and modified the manuscript with*

additional experiments. We believe that your comments have significantly improved the quality of our manuscript.

(1) Regarding the cell dosing experiment presented in Figure 5, I believe that APROM may have been administered earlier in the day. In such circumstances, whether the cells exhibit dephosphorylation or a reduction in MPTP toxicity is unclear. In order to address this issue, the authors should consider the timing of administration and explore experimental methods that can verify this. Some potential approaches include adjusting the timing of administration or utilizing techniques such as pulse-chase analysis.

Response: *Thank you for bringing up this crucial point. Indeed, in Fig. 5, neurotoxin 1-methyl-4-phenylpyridinium (MPP⁺) and APROM₂ or CeNPs were added into culture dishes simultaneously rather than administration of APROM or CeNPs earlier in the day. We have modified methods for more clarity.*

1-Methyl-4-phenyl-1,2,3,6-tetrahydropyridine (MPTP), which is highly lipophilic, rapidly crosses the blood-brain barrier. Once in the brain, the protoxin MPTP is metabolized to 1-methyl-4-phenyl-2,3-dihydropyridinium by the enzyme monoamine oxidase B (MAO-B) within non-dopaminergic cells, and then (probably by spontaneous oxidation) to 1-methyl-4-phenylpyridinium (MPP⁺), which is the active toxic compound. Thereafter, MPP⁺ is released into the extracellular space and gains access to dopaminergic neurons. Subsequently, MPP⁺ cause damage on dopaminergic neurons (references: Nat. Protoc. 2, 141-151 (2007); J. Neurochem. 76, 1265-1274 (2001)). In animal experiments, APROM₂ has been administered earlier in the day, aligning with widely accepted practices and supported by various studies (references: Adv. Mater. 2106723 (2022); ACS Nano 14, 1533-1549 (2020)).

To address concerns taken by timing of administration, we have employed alternative animal models. Mice were initially treated with MPTP to elevate p- α -syn levels in the brain, followed by the administration of APROM₂ to assess its ability to dephosphorylate p- α -syn. The detailed methods are as follows: Mice were treated with MPTP (20 mg/kg body weight) via subcutaneous injection four times at 2 h intervals to induce PD. The control group was treated with the equivalent volume of saline. 24 h later, mice were administrated with saline, CeNPs (2 mg/kg body weight), or APROM₂ (2 mg/kg body weight) by tail vein injection. 24 h after the last injection, mice were sacrificed for further study. The timing of administration is similar to other studies (references: Angew. Chem. Int. Ed. 57, 1-6 (2018); ACS Nano 17, 7511-7529 (2023); J Control. Release 338 742-753 (2021)). The quantitative analysis of p- α -syn levels in the SN and ST demonstrates that APROM₂ can significantly dephosphorylate p- α -syn in MPTP-induced PD models. We appreciate your diligence in scrutinizing our experimental procedures and providing thoughtful suggestions, which have contributed to the overall robustness of our study.

Our modification to the manuscript: *The methods were modified in the revised manuscript. The quantitative analysis of p- α -syn levels in the SN and ST was added as Supplementary Fig. 23 in the revised supporting information.*

- Methods

“**Cell viability assay.** To investigate the protective effect of APROM₂ and CeNPs, DMEM medium containing MPP⁺ (1 mM) and APROM₂ (20 μ g/mL) or CeNPs (20 μ g/mL) was added into culture dishes, and SH-SY5Y cells were cultured for 24 h at 37 °C and 5% CO₂ atmosphere. The cell viability was measured by MTT assay as described previously.”

“**Analysis of synaptic activity.** Fluorescent dye FM1-43 was used to mimic the release of neurotransmitters as it can be incorporated into the synaptic vesicle membrane, and the release of neurotransmitters was measured by the decrease of FM1-43 fluorescence intensity in primary neurons. Firstly, DMEM medium containing MPP⁺ (50 μ M) and APROM₂ (20 μ g/mL) or CeNPs (20 μ g/mL) was added into culture dishes, and primary neurons were cultured for 24 h. Then, primary neurons were incubated with HEPES buffer containing 100 μ M FM1-43, 1 mM Ca²⁺ and 30 mM K⁺ for 2 min, and the fluorescent signal was tested by using a Nikon confocal microscope (Nikon, Japan). Subsequently, neurons were incubated with HEPES buffer containing 1 mM Ca²⁺ and 15 mM K⁺ for 5 min, during which FM1-43 contained synaptic vesicles were released. The fluorescent signal was detected again by using a Nikon confocal microscope.”

“**Cellular ROS-scavenging capability.** To assay the ROS-scavenging capability of APROM₂ and CeNPs, DMEM medium containing MPP⁺ (50 μ M) and APROM₂ (20 μ g/mL) or CeNPs (20 μ g/mL) was added into culture dishes, and primary neurons were cultured for 24 h. Subsequently, primary neurons were incubated with DCFH-DA at 37 °C for 20 min and fixed in 4% paraformaldehyde for 15 min. The nuclei were stained with DAPI. The fluorescence images were obtained by using a laser scanning confocal microscope.”

“**Analysis of mitochondrial protective effect.** To test the mitochondrial protective effect of APROM₂ and CeNPs, DMEM medium containing MPP⁺ (50 μ M) and APROM₂ (20 μ g/mL) or CeNPs (20 μ g/mL) was added into culture dishes, and primary neurons were cultured for 24 h. Afterward, primary neurons were washed three times with PBS and treated according to the JC-1 kit. Finally, primary neurons were detected by a laser scanning confocal microscope.”

- Supplementary Figure 23

Supplementary Figure 23. The quantitative analysis of the p-α-syn level in the SN (a) and ST (b) after different treatments (n = 3 independent experiment). Data are presented as means ± s.e.m. Mice were treated with MPTP (20 mg/kg body weight) via subcutaneous injection four times at 2 h intervals to induce PD. The control group was treated with the equivalent volume of saline. 24 h later, mice were administrated with saline, CeNPs (2 mg/kg body weight), or APROM₂ (2 mg/kg body weight) by tail vein injection. 24 h after the last injection, mice were sacrificed for further quantitative analysis of the p-α-syn level in the SN and ST.

(2) While Figure 5b, c shows dephosphorylation without affecting the amount of synuclein, additional experiments, such as Western blotting, are necessary to confirm this.

Response: Thank you for your comments. Fig. 5b and c are immunofluorescence and mean fluorescence intensity of p-α-syn in primary neurons after different treatments, respectively. It is important to clarify that these results do not indicate dephosphorylation without affecting the amount of α-syn. In fact, these results convey that APROM₂ dephosphorylate hyperphosphorylated p-α-syn that are caused by MPP⁺, with decreasing the p-α-syn level approximate to the physiological level. Per your suggestion, we have added the Western blot analysis of p-α-syn to clearly identify that APROM₂ can dephosphorylate p-α-syn to rebalance the aberrant PTMs of α-syn.

Our modification to the manuscript: The following sentence was modified on page 13, and the western blot analysis of p-α-syn was added as Supplementary Fig. 17 in the revised manuscript.

- Page 13

“.....Upon MPP⁺ exposure, the intracellular p-α-syn level significantly increases (Fig. 5b,c, Supplementary Fig. 17). The C-terminal region of α-syn, which contains motifs of charged amino acids, is capable of binding to metal ions⁴⁶. Notably, phosphorylation of Ser 129 within the C-terminal region significantly enhances this affinity⁴⁷. In line with these observations, APROM₂ remarkably dephosphorylates p-α-syn with decreasing the p-α-syn level approximate to the physiological level via effective interaction with p-α-syn, thereby rebalancing the aberrant PTMs.”

- Supplementary Figure 17

Supplementary Figure 17. Western blot analysis of p-α-syn in primary neurons with different treatments.

(3) Sup16 is needed to prove the hypothesis, but it should be provided with appropriate contrast to clarify it.

Response: *Thank you for your thoughtful suggestion. The appropriate contrast has been done and shown in Fig. 5b,c in the revised manuscript. Specifically, compared with the control group, MPP⁺ treatment significantly elevates the level of p-α-syn in primary neurons (Fig. 5b,c). Notably, despite the effective intracellular ROS scavenging observed with NAC treatment (Fig. 5k, Supplementary Fig. 18), the p-α-syn level in NAC-treated primary neurons is significantly higher than it in control primary neurons ($p < 0.0001$) (Fig. 5b,c, Supplementary Fig. 19). This finding suggests that the elimination of ROS has limited impact on reducing p-α-syn levels. In contrast, APROM₂ demonstrates a notable reduction in p-α-syn levels in primary neurons. Therefore, our results verify that it is the protein phosphatase-mimetic activity of APROM₂, rather than its antioxidant properties, plays a crucial role in reducing p-α-syn levels. Accordingly, we have modified figure legends for more clarity.*

Our modification to the manuscript: *The following sentences were modified on page 14 in the revised manuscript, and the legends of Supplementary Fig. 18 and 19 were modified in the revised supporting information, respectively.*

- Page 14

“Notably, while the antioxidant N-acetylcysteine (NAC) efficiently scavenges reactive oxygen species (ROS) (Supplementary Fig. 18), it only minimally reduces p-α-syn levels (Fig. 5b,c, Supplementary Fig. 19), suggesting that the protein phosphatase-mimetic activity of APROM₂, rather than its antioxidant properties, plays a crucial role in reducing p-α-syn levels.”

- Supplementary Figure 18

Supplementary Figure 18. Intracellular ROS levels in MPP⁺ and NAC treated primary neurons. For comparison, the control and MPP⁺ groups are presented in **Figure 5k**.

- Supplementary Figure 19

Supplementary Figure 19. Immunofluorescence (a) and mean fluorescence intensity (b) of p- α -syn in MPP⁺ and NAC treated primary neurons (n = 3 biologically independent cultures). For comparison, the control and MPP⁺ groups are presented in **Figure 5b,c**, and the mean fluorescence intensity of the control group is 1. Data are presented as means \pm s.e.m.

(4) While Figure 5d-g shows confocal localization to synaptic vesicles, additional methods, such as isolating vesicles and verifying the proteins present, are necessary to prove that α -syn binds directly to VMAT2 and that this is dependent on phosphorylation.

Response: Thank you for your insightful suggestion. There are many studies have confirmed that α -syn can directly interact with vesicular monoamine transporter 2 (VMAT2) and vesicle-associated membrane protein 2 (VAMP2) in synaptic vesicles to facilitate the incorporation of dopamine into synaptic vesicles and promote vesicle-membrane fusion for dopamine release, respectively (references: *Science* 329, 1663-1667 (2010); *Cell Mol. Neurobiol.* 28, 35-47 (2008); *Aging Cell* 18, e13031 (2019); *Mol. Neurodegener.* 12, 45 (2017)). For example, Burré et al. confirmed that α -syn directly binds to the SNARE-protein synaptobrevin-2/VAMP2 in synaptic vesicles, promoting SNARE-complex assembly (reference: *Science* 329, 1663-1667

(2010)). Guo et al. identified the direct interaction between α -syn and VMAT2, forming the α -syn-VMAT2 complex in presynaptic vesicles (reference: *Cell Mol. Neurobiol.* 28, 35-47 (2008)). Per your suggestion, we performed co-immunoprecipitation to investigate whether the interaction between α -syn and VMAT2 is dependent on phosphorylation. As anticipated, α -syn is co-immunoprecipitated with VMAT2 rather than p- α -syn, proving that α -syn can bind directly to VMAT2 and this interaction is dependent on dephosphorylation. We appreciate your thoughtful consideration and suggestions.

Our modification to the manuscript: The following sentences were modified on page 16 in the revised manuscript. The results of co-immunoprecipitation were added as Fig. 6k,l in the revised manuscript, and added as Supplementary Fig. 24 in the revised supporting information, respectively.

- Page 16

“.....More importantly, through APROM₂'s *de novo* PTM strategy, the biological function of α -syn is regained, as verified by the promoted colocalization of α -syn with VMAT2 in dopaminergic neurons that is dependent on dephosphorylation (Fig. 6k-m, Supplementary Fig. 24), as well as colocalization of α -syn with VAMP2 (Fig. 6n).”

• Figure 6

Figure 6. APROM₂ mediated synaptic plasticity improvement *in vivo*. **a**, Schematic illustration of APROM₂ that protects dopaminergic neurons against neurodegeneration for rescuing motor coordination in PD. **b-d**, Behavioral evaluation of pole test (**b**) rotarod test (**c**) and hang test (**d**) of mice after different treatments (n = 9 biologically independent mice). **e,f**, Quantitative analysis (**e**) and representative immunohistochemical staining images (**f**) of TH positive neurons in the SN (n = 3 biologically independent mice). Regions of interest (white square) in the top panels are shown at higher magnification in the bottom panels. **g**, Representative immunohistochemical staining images of dopaminergic fibers in the ST. Regions of interest (white square) in the top panels are shown at higher magnification in the bottom panels. **h**, Schematic illustration of APROM₂ that improves synaptic plasticity of dopaminergic neurons in PD. **I**, hyperphosphorylation of α -syn and oxidative stress impair synaptic function of dopaminergic neurons. **II**, APROM₂ dephosphorylates p- α -syn via protein phosphatase-mimetic activity and scavenges ROS via antioxidant activity. **III**, APROM₂ reprograms α -syn biological function via the *de novo* PTM strategy and alleviates oxidative stress for fueling synaptic function. **i**, Representative immunofluorescence staining images of p- α -syn in the SN and ST. **j**, Representative immunohistochemical staining images of α -syn inclusion in the SN. Inset, higher magnification of dopaminergic neurons. **k,l**, Co-immunoprecipitation of VMAT2 with α -syn (**k**) or p- α -syn (**l**) in the midbrain of APROM₂ treated PD mice. **m,n**, Representative immunofluorescence staining images and colocalization analysis along the white line of α -syn with VMAT2 (**m**) or VAMP2 (**n**). **o**, Bio-TEM images of synaptic vesicles in the SN of mice after different treatments. Regions of interest (red square) in the top panels are shown at a higher magnification in the bottom panels. **p**, Representative immunofluorescence staining images of 4-HNE. **q**, Schematic of the midbrain section and histological assay with H&E staining for the SN and ST of APROM₂ treated mice. All the data are presented as means \pm s.e.m. Statistical significance was analyzed by one-way ANOVA with multiple comparisons test.

- Supplementary Figure 24

Supplementary Figure 24. Co-immunoprecipitation of α -syn with VMAT2 in the midbrain of APROM₂ treated PD mice.

(5) Figure 5j has not been analyzed quantitatively.

Response: Thank you for your insightful comments. We have performed a quantitative analysis of Fig. 5j through measuring both the total neurite length and the number of primary neurites

in primary neurons after different treatments by using Fiji software (version 1.54f) with the simple neurite tracer plug-in (National Institute of Health, USA). As shown in Supplementary Fig. 17, the length and number of neurites in primary neurons are significantly compromised following MPP⁺ exposure, while APROM₂ substantially protects primary neurons against degeneration.

Our modification to the manuscript: The following sentences were modified on page 14 in the revised manuscript, and the results were added as Supplementary Fig. 20 in the revised supporting information, respectively.

- Page 14

“Importantly, APROM₂ promotes the length and number of neurites, enhancing neuronal connectivity and improving the synaptic plasticity of primary neurons (Fig. 5j, Supplementary Fig. 20).”

- Supplementary Figure 20

Supplementary Figure 20. Quantification of total neurite length (a) and number of primary neurites (b) in primary neurons after different treatments (n = 15 primary neurons). Data are presented as means ± s.e.m.

(6) Figure 6 lends credence to the notion that APROM mitigates the acute toxicity of MPTP, but it is uncertain whether APROM is dephosphorylated, as they propose. Demonstrating the drug's effect following phosphorylation would provide a more rigorous examination of the hypothesis. Alternatively, other animal models can be utilized to test this hypothesis.

Response: Thank you for bringing up this important point. We have comprehensively assessed the protein phosphatase-mimetic activity of APROM₂ using p-α-syn phosphorylated at Serine 129 as the phospho-substrate. As anticipated, APROM₂ exhibited a concentration-dependent dephosphorylation of p-α-syn (Fig. 3n-p), converting it into α-syn. Moreover, quantitative analysis of p-α-syn levels in the substantia nigra (SN) and striatum (ST) of mice after different treatments confirms that APROM₂ significantly reduces p-α-syn levels in PD (Fig. 6i, Supplementary Fig. 22). Furthermore, we conducted co-immunoprecipitation of α-syn or p-α-syn with VMAT2 in the midbrain of APROM₂ treated PD mice. The results reveal that α-syn is

co-immunoprecipitated with VMAT2 rather than p- α -syn, proving that α -syn can bind directly to VMAT2 and this interaction is dependent on dephosphorylation (Fig. 6k,l, Supplementary Fig. 24). Notably, the colocalization of α -syn with VMAT2 and VAMP2 was enhanced (Fig. 6m,n). These observations strongly validate that APROM₂ mitigates the toxicity of MPTP through reprogramming the biological function of α -syn by dephosphorylating p- α -syn.

Moreover, to further test this hypothesis, alternative animal models were employed. Mice were initially treated with MPTP to elevate p- α -syn levels in the brain, followed by the administration of APROM₂ to assess its ability to dephosphorylate p- α -syn. The detailed methods are as follows: mice were treated with MPTP (20 mg/kg body weight) via subcutaneous injection four times at 2 h intervals to induce PD. The control group was treated with the equivalent volume of saline. 24 h later, mice were administered with saline, CeNPs (2 mg/kg body weight), or APROM₂ (2 mg/kg body weight) by tail vein injection. 24 h after the last injection, mice were sacrificed for further study. The quantitative analysis of p- α -syn levels in the SN and ST demonstrates that APROM₂ can significantly dephosphorylate p- α -syn.

We believe that these findings strengthen the manuscript and further support the hypothesis that APROM₂ reprograms the biological function of α -syn through its *de novo* PTM strategy, thereby mitigating MPTP-induced toxicity and rescuing dopaminergic neurodegeneration.

Our modification to the manuscript: The following sentences were modified on pages 11 and 16 in the revised manuscript. The results of APROM₂'s protein phosphatase-mimetic activity by using p- α -syn as the phospho-substrate were added as Fig. 3n-p in the revised manuscript. The quantitative analysis of p- α -syn levels in the SN and ST was added as Supplementary Fig. 23 in the revised supporting information. The results of co-immunoprecipitation were added as Fig. 6k,l in the revised manuscript, and added as Supplementary Fig. 24 in the revised supporting information, respectively.

- Page 11

“.....Inspired by the remarkable dephosphorylation capability of APROM₂, we further investigated its protein phosphatase-mimetic activity using p- α -syn as the phospho-substrate. As expected, APROM₂ readily dephosphorylates p- α -syn in a concentration-dependent manner (Fig. 3n), and the K_m , V_{max} , k_{cat} of APROM₂ are 0.175 mM, 0.0015 μ M s⁻¹, and 0.764 s⁻¹ (Fig. 3o,p, Supplementary Table 3), respectively. This suggests that APROM₂ can efficiently dephosphorylate p- α -syn, converting it into α -syn through *de novo* PTMs.”

- Page 16

“.....More importantly, through APROM₂'s *de novo* PTM strategy, the biological function of α -syn is regained, as verified by the promoted colocalization of α -syn with VMAT2 in dopaminergic neurons that is dependent on dephosphorylation (Fig. 6k-m, Supplementary Fig. 24), as well as colocalization of α -syn with VAMP2 (Fig. 6n).”

• Figure 3

Figure 3. Asymmetric catalytic centers confer protein phosphatase-like characteristics to APROMs. **a,b**, Local coordination environments and corresponding differential charge density maps of OH absorbed CeNPs (**a**) and APROM (**b**). Red and blue represent accumulation and

depletion charge areas, respectively. **c**, Free-energy diagram of phospho-substrate hydrolysis on the APROM and CeNPs. **d,e**, Differential charge density maps of OH and phospho-substrate absorbed CeNPs (**d**) and APROM (**e**). **f**, Bader charge of the O atom of OH before and after phospho-substrate absorption. A positive value indicates that the atom gains electrons. **g**, DOS of CeNPs and the APROM. E = energy level, E_{Fermi} = Fermi level. **h,i**, PDOS of CeNPs (**h**) and the APROM (**i**). **j**, Protein phosphatases-mimetic activity of APROMs and CeNPs by using P-Ser as the phospho-substrate (n = 3 independent experiments). **k**, Raman spectra of APROM₂ with and without incubation of P-Ser. The P-Ser is used as the phospho-substrate for the dephosphorylation reaction. F_{2g} , the Raman-active vibrational mode of the cubic fluorite structure. **l**, The Ce-to-Mn ratio of APROM₂ before and after catalyzing phospho-substrate dephosphorylation. **m**, *In situ* XRD of APROM₂ under dephosphorylation reaction conditions. **n**, Protein phosphatases-mimetic activity of APROM₂ by using p- α -syn as the phospho-substrates (n = 3 independent experiments). **o,p**, Michaelis-Menten kinetics (**o**) and Lineweaver-Burk plotting (**p**) of APROM₂ obtained by adding different concentrations of p- α -syn (n = 3 independent experiments). **q**, Protein phosphatases-mimetic activity of APROM₂ by using P-Tyr and P-Thr as the phospho-substrates (n = 3 independent experiments). All the data are presented as means \pm s.e.m. Statistical significance was analyzed by one-way ANOVA with multiple comparisons test.

• Figure 6

Figure 6. APROM₂ mediated synaptic plasticity improvement *in vivo*. **a**, Schematic illustration of APROM₂ that protects dopaminergic neurons against neurodegeneration for rescuing motor coordination in PD. **b-d**, Behavioral evaluation of pole test (**b**) rotarod test (**c**) and hang test (**d**) of mice after different treatments (n = 9 biologically independent mice). **e,f**, Quantitative analysis (**e**) and representative immunohistochemical staining images (**f**) of TH positive neurons in the SN (n = 3 biologically independent mice). Regions of interest (white square) in the top panels are shown at higher magnification in the bottom panels. **g**, Representative immunohistochemical staining images of dopaminergic fibers in the ST. Regions of interest (white square) in the top panels are shown at higher magnification in the bottom panels. **h**, Schematic illustration of APROM₂ that improves synaptic plasticity of dopaminergic neurons in PD. **I**, hyperphosphorylation of α -syn and oxidative stress impair synaptic function of dopaminergic neurons. **II**, APROM₂ dephosphorylates p- α -syn via protein phosphatase-mimetic activity and scavenges ROS via antioxidant activity. **III**, APROM₂ reprograms α -syn biological function via the *de novo* PTM strategy and alleviates oxidative stress for fueling synaptic function. **i**, Representative immunofluorescence staining images of p- α -syn in the SN and ST. **j**, Representative immunohistochemical staining images of α -syn inclusion in the SN. Inset, higher magnification of dopaminergic neurons. **k,l**, Co-immunoprecipitation of VMAT2 with α -syn (**k**) or p- α -syn (**l**) in the midbrain of APROM₂ treated PD mice. **m,n**, Representative immunofluorescence staining images and colocalization analysis along the white line of α -syn with VMAT2 (**m**) or VAMP2 (**n**). **o**, Bio-TEM images of synaptic vesicles in the SN of mice after different treatments. Regions of interest (red square) in the top panels are shown at a higher magnification in the bottom panels. **p**, Representative immunofluorescence staining images of 4-HNE. **q**, Schematic of the midbrain section and histological assay with H&E staining for the SN and ST of APROM₂ treated mice. All the data are presented as means \pm s.e.m. Statistical significance was analyzed by one-way ANOVA with multiple comparisons test.

- Supplementary Figure 23

Supplementary Figure 23. The quantitative analysis of the p- α -syn level in the SN (**a**) and ST (**b**) after different treatments (n = 3 independent experiment). Data are presented as means \pm s.e.m. Mice were treated with MPTP (20 mg/kg body weight) via subcutaneous injection four times at 2 h intervals to induce PD. The control group was treated with the equivalent volume

of saline. 24 h later, mice were administrated with saline, CeNPs (2 mg/kg body weight), or APROM₂ (2 mg/kg body weight) by tail vein injection. 24 h after the last injection, mice were sacrificed for further quantitative analysis of the p- α -syn level in the SN and ST.

- Supplementary Figure 24

Supplementary Figure 24. Co-immunoprecipitation of α -syn with VMAT2 in the midbrain of APROM₂ treated PD mice.

(7) The immunohistochemical analysis of Figure 6 has not been biochemically or quantitatively analyzed, biochemically insoluble fractions, and the degree of phosphorylation needs to be shown for 6i.

Response: *Thank you for your insightful suggestion. We sincerely appreciate the thorough evaluation of our manuscript. In response to your suggestion, we have conducted quantitative analyses of Fig. 6i, including the assessment of biochemically insoluble fractions and the degree of phosphorylation. As depicted in Supplementary Fig. 22, APROM₂ effectively reverses the hyperphosphorylation state of α -syn at Ser-129 in both the SN and ST, and consequently inhibits the subsequent formation of biochemically insoluble fractions.*

Our modification to the manuscript: *The following sentences were modified on page 16 in the revised manuscript. The quantitative analysis of biochemically insoluble fractions and phosphorylation degree was added as Supplementary Fig. 22 in the revised supporting information.*

- page 16

“.....APROM₂ effectively reverses the hyperphosphorylation state of α -syn at Ser-129 in both the SN and striatum (ST) (Fig. 6i, Supplementary Figs. 22 and 23). This ability highlights the capacity of APROM₂ to rebalance the abnormal PTMs of α -syn and consequently inhibit the subsequent formation of toxic α -syn inclusions (Fig. 6j, Supplementary Fig. 22).”

• Supplementary Figure 22

Supplementary Figure 22. The quantitative analysis of the p-α-syn level in the SN (a) and ST (b) after different treatments (n = 5 independent experiment). The insoluble fraction levels in the SN (c) and ST (d) after different treatments (n = 5 independent experiment). Data are presented as means ± s.e.m.

(8) I cannot confirm the expression of α-syn in Figures 6k and l.

Response: Thank you for your kind comments. We have redone the immunofluorescence staining for Fig. 6k and l, as depicted in Fig. 6m,n in the revised manuscript. The ability of α-syn to bind with VMAT2 and VAMP2 is significantly impaired in PD mice. Through APROM₂'s de novo PTM strategy, the biological function of α-syn is regained, as verified by the promoted colocalization of α-syn with VMAT2 and VAMP2 in dopaminergic neurons. These findings confirm its ability to reprogram the biological function of α-syn.

Our modification to the manuscript: Fig. 6m and n were replaced in the revised manuscript.

• Figure 6

Figure 6. APROM₂ mediated synaptic plasticity improvement *in vivo*. **a**, Schematic illustration of APROM₂ that protects dopaminergic neurons against neurodegeneration for rescuing motor coordination in PD. **b-d**, Behavioral evaluation of pole test (**b**) rotarod test (**c**) and hang test (**d**) of mice after different treatments (n = 9 biologically independent mice). **e,f**, Quantitative analysis (**e**) and representative immunohistochemical staining images (**f**) of TH positive neurons in the SN (n = 3 biologically independent mice). Regions of interest (white square) in the top panels are shown at higher magnification in the bottom panels. **g**, Representative immunohistochemical staining images of dopaminergic fibers in the ST. Regions of interest (white square) in the top panels are shown at higher magnification in the bottom panels. **h**, Schematic illustration of APROM₂ that improves synaptic plasticity of dopaminergic neurons in PD. **I**, hyperphosphorylation of α -syn and oxidative stress impair synaptic function of dopaminergic neurons. **II**, APROM₂ dephosphorylates p- α -syn via protein phosphatase-mimetic activity and scavenges ROS via antioxidant activity. **III**, APROM₂ reprograms α -syn biological function via the *de novo* PTM strategy and alleviates oxidative stress for fueling synaptic function. **i**, Representative immunofluorescence staining images of p- α -syn in the SN and ST. **j**, Representative immunohistochemical staining images of α -syn inclusion in the SN. Inset, higher magnification of dopaminergic neurons. **k,l**, Co-immunoprecipitation of VMAT2 with α -syn (**k**) or p- α -syn (**l**) in the midbrain of APROM₂ treated PD mice. **m,n**, Representative immunofluorescence staining images and colocalization analysis along the white line of α -syn with VMAT2 (**m**) or VAMP2 (**n**). **o**, Bio-TEM images of synaptic vesicles in the SN of mice after different treatments. Regions of interest (red square) in the top panels are shown at a higher magnification in the bottom panels. **p**, Representative immunofluorescence staining images of 4-HNE. **q**, Schematic of the midbrain section and histological assay with H&E staining for the SN and ST of APROM₂ treated mice. All the data are presented as means \pm s.e.m. Statistical significance was analyzed by one-way ANOVA with multiple comparisons test.

(9) While the paper suggests improved synaptic function in Parkinson's disease, this is an oversimplification as the study is conducted in a model system of cultured cells.

Response: Thank you for your comment. The synaptic function of dopaminergic neurons is closely related to the biological function of α -syn, synaptic vesicles and the levels of dopamine. Therefore, we investigated them in PD mice. Through APROM₂'s *de novo* PTM strategy, p- α -syn is substantially converted into α -syn (Fig. 6i, Supplementary Fig. 22), thereby the biological function of α -syn is regained, as verified by the promoted colocalization of α -syn with VMAT2 and VAMP2 in dopaminergic neurons (Fig. 6k,l). In addition, we studied synaptic vesicles in the SN of mice after different treatments via bio-TEM. As shown in Fig. 6o, the synaptic vesicles are impaired upon MPTP treatment, while APROM₂ treatment significantly recovers the synaptic vesicles. Moreover, the dopamine level of PD mice return to normal

physiological levels with the treatment of APROM₂ (Supplementary Fig. 25).

Our modification to the manuscript: *The following sentence was added on page 16, and bio-TEM images were added as Fig. 6o in the revised manuscript, respectively. The dopamine level measurement was added as Supplementary Fig. 25 in the revised supporting information.*

- Page 16

“.....Moreover, synaptic vesicles are significantly recovered with the treatment of APROM₂ (Fig. 6o), and the dopamine level of PD mice return to normal physiological levels (Supplementary Fig. 25). This finding indicates the reestablishment of synaptic function.”

• Figure 6

Figure 6. APROM₂ mediated synaptic plasticity improvement *in vivo*. **a**, Schematic illustration of APROM₂ that protects dopaminergic neurons against neurodegeneration for rescuing motor coordination in PD. **b-d**, Behavioral evaluation of pole test (**b**) rotarod test (**c**) and hang test (**d**) of mice after different treatments (n = 9 biologically independent mice). **e,f**, Quantitative analysis (**e**) and representative immunohistochemical staining images (**f**) of TH positive neurons in the SN (n = 3 biologically independent mice). Regions of interest (white square) in the top panels are shown at higher magnification in the bottom panels. **g**, Representative immunohistochemical staining images of dopaminergic fibers in the ST. Regions of interest (white square) in the top panels are shown at higher magnification in the bottom panels. **h**, Schematic illustration of APROM₂ that improves synaptic plasticity of dopaminergic neurons in PD. **i**, hyperphosphorylation of α -syn and oxidative stress impair synaptic function of dopaminergic neurons. **ii**, APROM₂ dephosphorylates p- α -syn via protein phosphatase-mimetic activity and scavenges ROS via antioxidant activity. **iii**, APROM₂ reprograms α -syn biological function via the *de novo* PTM strategy and alleviates oxidative stress for fueling synaptic function. **iv**, Representative immunofluorescence staining images of p- α -syn in the SN and ST. **v**, Representative immunohistochemical staining images of α -syn inclusion in the SN. Inset, higher magnification of dopaminergic neurons. **vi**, Co-immunoprecipitation of VMAT2 with α -syn (**vii**) or p- α -syn (**viii**) in the midbrain of APROM₂ treated PD mice. **ix**, Representative immunofluorescence staining images and colocalization analysis along the white line of α -syn with VMAT2 (**x**) or VAMP2 (**xi**). **xii**, Bio-TEM images of synaptic vesicles in the SN of mice after different treatments. Regions of interest (red square) in the top panels are shown at a higher magnification in the bottom panels. **xiii**, Representative immunofluorescence staining images of 4-HNE. **xiv**, Schematic of the midbrain section and histological assay with H&E staining for the SN and ST of APROM₂ treated mice. All the data are presented as means \pm s.e.m. Statistical significance was analyzed by one-way ANOVA with multiple comparisons test.

- Supplementary Figure 25

Supplementary Figure 25. The dopamine level in the SN of mice after different treatments (n = 5 biologically independent mice). Data are presented as means \pm s.e.m.

(10) Although the results suggest that neuronal cell death may be inhibited, it is impossible to describe morphology or synaptogenesis from these data.

In line 37, you mention improved synaptic function in Parkinson's disease, but I think this is an oversimplification, as the paper only shows results in a model system of cultured cells.

Lines 425 and 492, but from these results, it is not possible to describe morphology or synaptogenesis, although it is possible that neuronal cell death is inhibited.

Response: *Thank you for your valuable comment. The synaptic function of dopaminergic neurons is intricately linked to the biological role of α -syn, synaptic vesicles, and the dopamine level. Therefore, in our investigation of PD mice, we explored these aspects. Through APROM2's de novo PTM strategy, we observed a substantial conversion of p- α -syn into α -syn (Fig. 6i), thereby restoring the biological function of α -syn. This was corroborated by the enhanced colocalization of α -syn with VMAT2 and VAMP2 in dopaminergic neurons (Fig. 6k,l). Additionally, we examined synaptic vesicles in the substantia nigra of mice after different treatments using bio-TEM. As depicted in Fig. 6o, synaptic vesicles were impaired after MPTP treatment, while APROM₂ treatment significantly restored synaptic vesicles. Moreover, the dopamine level in PD mice returned to normal physiological levels with APROM₂ treatment (Supplementary Fig. 25).*

It is acknowledged that describing the promotion of neural morphogenesis and synaptic plasticity solely through immunohistochemical staining images of TH and terminal deoxynucleotidyl transferase dUTP nick end labeling staining images can be challenging. Therefore, in line 425, we have adjusted the sentence to convey that APROM₂ remarkably rescues dopaminergic neurons. The evidence we present supports the conclusion that APROM₂ enhances synaptic plasticity, as demonstrated through immunofluorescence, colocalization analysis of α -syn with VMAT2 and VAMP2, microscopy images of primary neurons, bio-TEM of synapses, and dopamine level measurements. We hope these additional data clarify and strengthen the accuracy of the statement in line 492.

Our modification to the manuscript: *The following sentence was modified on page 16, and bio-TEM images were added as Fig. 6o in the revised manuscript, respectively. The dopamine level measurement was added as Supplementary Fig. 25 in the revised supporting information.*

- Page 16

“.....Based on these findings, APROM₂ remarkably rescues dopaminergic neurons. In contrast, disappointing neurodegeneration develops in the SN of CeNPs-treated PD mice, despite the protective effect of CeNPs on dopaminergic neurons.”

“.....Moreover, synaptic vesicles are significantly recovered with the treatment of APROM₂ (Fig. 6o), and the dopamine level of PD mice return to normal physiological levels (Supplementary Fig. 25). This finding indicates the reestablishment of synaptic function.”

• Figure 6

Figure 6. APROM₂ mediated synaptic plasticity improvement *in vivo*. **a**, Schematic illustration of APROM₂ that protects dopaminergic neurons against neurodegeneration for rescuing motor coordination in PD. **b-d**, Behavioral evaluation of pole test (**b**) rotarod test (**c**) and hang test (**d**) of mice after different treatments (n = 9 biologically independent mice). **e,f**, Quantitative analysis (**e**) and representative immunohistochemical staining images (**f**) of TH positive neurons in the SN (n = 3 biologically independent mice). Regions of interest (white square) in the top panels are shown at higher magnification in the bottom panels. **g**, Representative immunohistochemical staining images of dopaminergic fibers in the ST. Regions of interest (white square) in the top panels are shown at higher magnification in the bottom panels. **h**, Schematic illustration of APROM₂ that improves synaptic plasticity of dopaminergic neurons in PD. **I**, hyperphosphorylation of α -syn and oxidative stress impair synaptic function of dopaminergic neurons. **II**, APROM₂ dephosphorylates p- α -syn via protein phosphatase-mimetic activity and scavenges ROS via antioxidant activity. **III**, APROM₂ reprograms α -syn biological function via the *de novo* PTM strategy and alleviates oxidative stress for fueling synaptic function. **i**, Representative immunofluorescence staining images of p- α -syn in the SN and ST. **j**, Representative immunohistochemical staining images of α -syn inclusion in the SN. Inset, higher magnification of dopaminergic neurons. **k,l**, Co-immunoprecipitation of VMAT2 with α -syn (**k**) or p- α -syn (**l**) in the midbrain of APROM₂ treated PD mice. **m,n**, Representative immunofluorescence staining images and colocalization analysis along the white line of α -syn with VMAT2 (**m**) or VAMP2 (**n**). **o**, Bio-TEM images of synaptic vesicles in the SN of mice after different treatments. Regions of interest (red square) in the top panels are shown at a higher magnification in the bottom panels. **p**, Representative immunofluorescence staining images of 4-HNE. **q**, Schematic of the midbrain section and histological assay with H&E staining for the SN and ST of APROM₂ treated mice. All the data are presented as means \pm s.e.m. Statistical significance was analyzed by one-way ANOVA with multiple comparisons test.

- Supplementary Figure 25

Supplementary Figure 25. The dopamine level in the SN of mice after different treatments (n = 5 biologically independent mice). Data are presented as means \pm s.e.m.

Reviewer #4 (Remarks to the Author):

Comments to Manuscript NCOMMS-23-36937-T

This manuscript describes interesting and noteworthy results, but the work requires substantial improvement and the manuscript requires revision addressing the specific comments below.

Response: *Thank you for your valuable comments. Based on your kind suggestion, we have made point-to-point responses and modified the manuscript. We believe these revisions have substantially improved the quality of our manuscript.*

1) Page 3, Line 65: This formulation needs to be improved. What is meant by the authors with the term “asymmetric molecular catalysts”?

Response: *Thank you for your comments. We have replaced “asymmetric molecular catalysts” with “small-molecule asymmetric catalysts”, referring to small-molecule catalysts that possess asymmetric catalytic centers.*

Our modification to the manuscript: *The following sentence was modified on page 3 in the revised manuscript.*

- Page 3

“While **small-molecule asymmetric catalysts**²⁷ can selectively generate favorable intermediates for efficient catalytic reactions^{28, 29}, their practical biomedical applications are hindered by challenges such as low solubility and off-target side effects^{30, 31}.”

2) Page 4, Line 66: With the unclear definition of “asymmetric molecular catalysts” the challenges of the low solubility and off-target side effects should be explained in more details.

Response: *Thank you for your kind comments. In our manuscript, the small-molecule asymmetric catalysts refer to small-molecule catalysts that possess asymmetric catalytic centers. Studies have reported that nearly 90% of molecules in the discovery pipeline are poorly water-soluble, which heavily influences their bioavailability (reference: Acta Pharm. Sin. B. 5, 442-453 (2015)). In addition, by virtue of their size, small molecules can rapidly diffuse through biological fluids, across many biological barriers and through cell membranes. These enable them to navigate the complex vasculature and to interact with nearly all tissues and cell types in the body, leading to the off-target side effect (reference: Nat. Biomed. Eng. 5, 951-967 (2021)). We have added the reference in the revised manuscript for more clarity.*

Our modification to the manuscript: *The reference was added in the revised manuscript.*

- Page 3

“While small-molecule asymmetric catalysts²⁷ can selectively generate favorable intermediates for efficient catalytic reactions^{28, 29}, their practical biomedical applications are hindered by challenges such as low solubility and off-target side effects^{30, 31}.”

- Reference

31. Kalepu, S., Nekkanti, V. Insoluble drug delivery strategies: review of recent advances and business prospects. *Acta Pharm. Sin. B.* **5**, 442-453 (2015).

3) Page 4, Line 68: It should be explained in more detail why nanomaterials based heterogeneous catalysts show promise in overcoming the inherent limitations of molecular catalysts.

Response: *Thank you for your thoughtful comments. Indeed, molecular catalysts encounter inherent limitations such as low water solubility and potential off-target side effects. In contrast, nanomaterials-based heterogeneous catalysts offer advantages due to their modifiable surface. These catalysts can be easily tailored with various ligands, including polyethylene glycol (PEG) and targeting ligands, addressing concerns related to biocompatibility, water solubility, and dispersibility. PEGylation enhances these nanoparticles' biocompatibility and improves their water solubility and dispersibility. Moreover, it shields the surface charge, prolonging their circulation half-life in the bloodstream and enhancing cellular uptake (references: *Angew. Chem. Int. Ed.* **50**, 1980-1994 (2011); *Nat. Biomed. Eng.* **5**, 951-967 (2021)). Incorporating targeting ligands equips nanoparticles with the ability to overcome barriers, such as the blood-brain barrier (BBB), and accumulate in specific disease lesions (reference: *Adv. Mater.* **32**, 1902604 (2020)). Examples include modifications with rabies virus glycoprotein (RVG), angiopep-2, or transferrin to facilitate BBB crossing, and 2-methacryloyloxyethyl phosphorylcholine polymer for BBB penetration and targeting of dopaminergic neurons (reference: *Adv. Mater.* **34**, 2105711 (2022)). Consequently, nanomaterials-based heterogeneous catalysts hold promise in addressing the inherent limitations of molecular catalysts. We have revised the manuscript and included relevant references for clarity.*

Our modification to the manuscript: *The sentence was modified on page 4, and the references were added in the revised manuscript, respectively.*

- Page 4

“In this regard, nanomaterials based heterogeneous catalysts show promise in overcoming the inherent limitations of molecular catalysts due to their modifiable surface^{30, 32, 33}.”

- Reference

32. Karakoti, A. S., Das, S., Thevuthasan, S., Seal, S. PEGylated inorganic nanoparticles. *Angew. Chem. Int. Ed.* **50**, 1980-1994 (2011).

33. Mi, P., Cabral, H., Kataoka, K., Ligand-Installed nanocarriers toward precision therapy.

4) Page 13, Line 241: Which classes of phospho-substrates?

Response: *Thank you for your comment. Accumulating evidence demonstrates that cerium ions can bind with the phosphate group and hydrolyze the phosphate monoester bonds of phospho-substrates (references: Adv. Sci. 8, 2004115 (2021); Coord. Chem. Rev. 382 145-159 (2019)). In our manuscript, we used the phospho-substrate that contains the phosphate monoester bond as the model to study the phosphatase-like characteristics of asymmetric catalytic centers through density functional theory calculations. We have revised the sentence in line 241 on page 13 for more clarity.*

Our modification to the manuscript: *The following sentences were modified on page 9 in the revised manuscript.*

- Page 9

“.....The formed bridging μ_3 -hydroxide further influences the subsequent hydrolysis of phospho-substrates that contain phosphate monoester bonds.”

5) Page 13, Line 258: The general statement that “... the APROM exhibits strong catalytic activity in phospho-substrate hydrolysis, resembling the function of protein phosphatases.” needs to be supported by a range of representative phospho-substrates.

Response: *Thank you for your insightful suggestion. As reversible protein phosphorylation primarily occurs in serine, threonine, and tyrosine residues of proteins (references: Cell 127, 635-648 (2006); Proc. Natl. Acad. Sci. USA. 77, 1311-1315 (1980)), we utilized O-phospho-L-serine (P-Ser), O-phospho-L-tyrosine (P-Tyr), and O-phospho-L-threonine (P-Thr) as phospho-substrates to assess the protein phosphatase-mimetic activity of APROM. The results indicate that APROM can effectively dephosphorylate P-Ser, P-Tyr, and P-Thr (Fig. 3). Thus, the APROM exhibits strong catalytic activity in phospho-substrate hydrolysis, resembling the function of protein phosphatases.*

Our modification to the manuscript: *The following sentences were added on page 11, the protein phosphatase-mimetic activity of APROM₂ by using P-Tyr and P-Thr as phospho-substrates was added as Fig. 3q, and the method was added in the revised manuscript.*

- Page 11

“.....Moreover, APROM₂ exhibits pronounced dephosphorylation activity towards O-phospho-L-tyrosine (P-Tyr) and O-phospho-L-threonine (P-Thr), effectively cleaving the phosphate monoester bonds (Fig. 3q). This collective evidence solidifies the notion that APROM₂ faithfully emulates the functionality of protein phosphatases, encompassing both

protein serine/threonine phosphatases and protein tyrosine phosphatases.”

• Figure 3

Figure 3. Asymmetric catalytic centers confer protein phosphatase-like characteristics to APROMs. a,b, Local coordination environments and corresponding differential charge density

maps of OH absorbed CeNPs (a) and APROM (b). Red and blue represent accumulation and depletion charge areas, respectively. c, Free-energy diagram of phospho-substrate hydrolysis on the APROM and CeNPs. d,e, Differential charge density maps of OH and phospho-substrate absorbed CeNPs (d) and APROM (e). f, Bader charge of the O atom of OH before and after phospho-substrate absorption. A positive value indicates that the atom gains electrons. g, DOS of CeNPs and the APROM. E = energy level, E_{Fermi} = Fermi level. h,i, PDOS of CeNPs (h) and the APROM (i). j, Protein phosphatases-mimetic activity of APROMs and CeNPs by using P-Ser as the phospho-substrate (n = 3 independent experiments). k, Raman spectra of APROM₂ with and without incubation of P-Ser. The P-Ser is used as the phospho-substrate for the dephosphorylation reaction. F_{2g} , the Raman-active vibrational mode of the cubic fluorite structure. l, The Ce-to-Mn ratio of APROM₂ before and after catalyzing phospho-substrate dephosphorylation. m, *In situ* XRD of APROM₂ under dephosphorylation reaction conditions. n, Protein phosphatases-mimetic activity of APROM₂ by using p- α -syn as the phospho-substrates (n = 3 independent experiments). o,p, Michaelis-Menten kinetics (o) and Lineweaver-Burk plotting (p) of APROM₂ obtained by adding different concentrations of p- α -syn (n = 3 independent experiments). q, Protein phosphatases-mimetic activity of APROM₂ by using P-Tyr and P-Thr as the phospho-substrates (n = 3 independent experiments). All the data are presented as means \pm s.e.m. Statistical significance was analyzed by one-way ANOVA with multiple comparisons test.

- Methods

“Protein phosphatases-mimetic activity assay. Additionally, the protein phosphatase-mimetic activity of APROM₂ (20 $\mu\text{g/mL}$) was also detected by using p- α -syn, O-phospho-L-tyrosine (P-Tyr) and O-phospho-L-threonine (P-Thr) as the phospho-substrates. The kinetic assays of APROM₂ (20 $\mu\text{g/mL}$) were carried out at 37 °C using a series of P-Ser concentrations (6.25, 12.5, 25, 50, 100 μM) or a series of p- α -syn concentrations (6.25, 12.5, 25, 50 μM). The Michaelis-Menten constant was calculated by using GraphPad Prism 8.0 (GraphPad Software).”

6) Page 13, Line 263: Have any other phospho-substrates than O-phospho-L-serine (P-Ser) been tested? If only O-phospho-L-serine (P-Ser) was used, then the term “ phospho-substrates” should be replaced by the term “phospho-substrate”.

Response: Thank you for your valuable suggestion. In response to your comment, we have extended our investigation to include P-Tyr and P-Thr as phospho-substrates for testing the protein phosphatase-mimetic activity of APROM₂. As illustrated in Fig. 3q of the revised manuscript, APROM₂ effectively dephosphorylates both P-Tyr and P-Thr. It is important to note that, as a proof-of-concept, we initially focused on P-Ser as the phospho-substrate for an in-depth analysis, given that phosphorylation at Ser129 is a predominant pathological post-translational modification of p- α -syn.

Our modification to the manuscript: *The following sentences were modified on page 10, and the protein phosphatase-mimetic activity of APROM₂ by using P-Tyr and P-Thr as phospho-substrates was added as Fig. 3q in the revised manuscript.*

- Page 10

“.....As a proof-of-concept, the protein phosphatase-mimetic activity of APROMs was tested using O-phospho-L-serine (P-Ser) as the phospho-substrate, as phosphorylation at Ser-129 is a predominant pathological PTM of p- α -syn⁴².”

• Figure 3

Figure 3. Asymmetric catalytic centers confer protein phosphatase-like characteristics to APROMs. **a,b**, Local coordination environments and corresponding differential charge density maps of OH absorbed CeNPs (**a**) and APROM (**b**). Red and blue represent accumulation and

depletion charge areas, respectively. **c**, Free-energy diagram of phospho-substrate hydrolysis on the APROM and CeNPs. **d,e**, Differential charge density maps of OH and phospho-substrate absorbed CeNPs (**d**) and APROM (**e**). **f**, Bader charge of the O atom of OH before and after phospho-substrate absorption. A positive value indicates that the atom gains electrons. **g**, DOS of CeNPs and the APROM. E = energy level, E_{Fermi} = Fermi level. **h,i**, PDOS of CeNPs (**h**) and the APROM (**i**). **j**, Protein phosphatases-mimetic activity of APROMs and CeNPs by using P-Ser as the phospho-substrate (n = 3 independent experiments). **k**, Raman spectra of APROM₂ with and without incubation of P-Ser. The P-Ser is used as the phospho-substrate for the dephosphorylation reaction. F_{2g} , the Raman-active vibrational mode of the cubic fluorite structure. **l**, The Ce-to-Mn ratio of APROM₂ before and after catalyzing phospho-substrate dephosphorylation. **m**, *In situ* XRD of APROM₂ under dephosphorylation reaction conditions. **n**, Protein phosphatases-mimetic activity of APROM₂ by using p- α -syn as the phospho-substrates (n = 3 independent experiments). **o,p**, Michaelis-Menten kinetics (**o**) and Lineweaver-Burk plotting (**p**) of APROM₂ obtained by adding different concentrations of p- α -syn (n = 3 independent experiments). **q**, Protein phosphatases-mimetic activity of APROM₂ by using P-Tyr and P-Thr as the phospho-substrates (n = 3 independent experiments). All the data are presented as means \pm s.e.m. Statistical significance was analyzed by one-way ANOVA with multiple comparisons test.

7) Page 14 Line 265: The statement that “Comparative analysis reveals that APROMs exhibit superior protein phosphatase-mimetic activity” is not sufficiently supported by figures 3j and the supplementary figure 10. The kinetic characterization, statements on the kinetics and catalytic efficiency of the APROMS with regard to their protein phosphatase-mimetic activity, and a figure on the dependence of initial rates on the concentration of O-phospho-L-serine (P-Ser) are missing.

Response: *Thank you for your valuable suggestion. In response to your comment, we conducted kinetic assays for APROM₂, which contains the highest number of asymmetric catalytic centers and exhibits the most potent protein phosphatase-mimetic activity (Fig. 3j). The initial rates of the hydrolysis reaction catalyzed by APROM₂ were measured at different P-Ser concentrations (6.25, 12.5, 25, 50, 100 μ M). We employed Michaelis-Menten kinetics, depicting the dependence of initial rates on the concentration of P-Ser, and Lineweaver-Burk plotting of APROM₂ using GraphPad Prism 8.0 (GraphPad Software). Subsequently, the K_m , V_{max} , and k_{cat} were calculated. The kinetic and catalytic efficiency of APROM₂ with regard to its protein phosphatase-mimetic activity have been included in the revised manuscript. These additional details reinforce our assertion that APROMs exhibit superior protein phosphatase-mimetic activity.*

Our modification to the manuscript: *The following sentences were modified on pages 10-11,*

and the method was added in the revised manuscript. The steady-state kinetics results of APROM₂ were added as Supplementary Fig. 11 and Supplementary Table 3 in the revised supporting information, respectively.

- Pages 10-11

“.....Comparative analysis reveals that APROMs exhibit superior protein phosphatase-mimetic activity (Fig. 3j, Supplementary Fig. 10), with APROM₂, containing the highest number of asymmetric catalytic centers, exhibiting the most potent protein phosphatase-mimetic activity. According to the Michaelis-Menten plot and Lineweaver-Burk plot, the Michaelis-Menten constant (K_m), maximum velocity (V_{max}) and turnover number (k_{cat}) of APROM₂ are 39.17 mM, 0.49 $\mu\text{M s}^{-1}$, and $2.47 \times 10^2 \text{ s}^{-1}$, respectively, indicating APROM₂ can effectively dephosphorylate P-Ser (Supplementary Fig. 11, Supplementary Table 3).”

- Methods

“Protein phosphatases-mimetic activity assay.Additionally, the protein phosphatase-mimetic activity of APROM₂ (20 $\mu\text{g/mL}$) was also detected by using p- α -syn, O-phospho-L-tyrosine (P-Tyr) and O-phospho-L-threonine (P-Thr) as the phospho-substrates. The kinetic assays of APROM₂ (20 $\mu\text{g/mL}$) were carried out at 37 °C using a series of P-Ser concentrations (6.25, 12.5, 25, 50, 100 μM) or a series of p- α -syn concentrations (6.25, 12.5, 25, 50 μM). The Michaelis-Menten constant was calculated by using GraphPad Prism 8.0 (GraphPad Software).”

- Supplementary Figure 11

Supplementary Figure 11. Michaelis-Menten kinetics (a) and Lineweaver-Burk plotting (b) of APROM₂ obtained by adding different concentrations of P-Ser (n = 3 independent experiments). Data are presented as means \pm s.e.m.

- Supplementary Table 3

Supplementary Table 3. Kinetics parameters of APROM₂ for protein phosphatase-mimetic activity.

Catalyst	Substrate	K_m (mM)	V_{max} ($\mu\text{M s}^{-1}$)	k_{cat} (s^{-1})
APROM ₂	P-Ser	39.17	0.49	2.47×10^2
APROM ₂	p- α -syn	0.175	1.5×10^{-3}	0.764

K_m is the Michaelis-Menten constant, V_{max} is the maximal reaction velocity, and k_{cat} is the catalytic constant.

8) Page 14, Line 267: It is not entirely clear what the authors mean by “optimal protein phosphatase-mimetic activity”.

Response: *Thank you for your comments. In fact, when referring to “optimal protein phosphatase-mimetic activity”, we intended to convey that APROM₂ exhibits the most potent protein phosphatase-mimetic activity in comparison to APROM₁ and APROM₃. To enhance the precision of our statement, we have replaced “optimal” to “most potent”.*

Our modification to the manuscript: *The following sentences were modified on page 10 in the revised manuscript.*

- Page 10

“.....with APROM₂, containing the highest number of asymmetric catalytic centers, exhibiting the most potent protein phosphatase-mimetic activity.”

9) Page 14, Line 268: Have any experimental tests been done to check whether APROM₂ can efficiently dephosphorylate p- α -syn, converting it into α -syn?

Response: *Thank you for your kind comment. We tested the protein phosphatase-mimetic activity of APROM₂ by using p- α -syn as the phospho-substrate. As shown in Fig. 3n, APROM₂ efficiently dephosphorylates p- α -syn in a concentration-dependent manner through catalytically cleaving its phosphate monoester bonds, thereby converting p- α -syn into α -syn and free phosphate.*

Our modification to the manuscript: *The following sentences were added on page 11, the results were added as Fig. 3n, and the method was added in the revised manuscript, respectively.*

- Page 11

“.....Inspired by the remarkable dephosphorylation capability of APROM₂, we further investigated its protein phosphatase-mimetic activity using p- α -syn as the phospho-substrate. As expected, APROM₂ readily dephosphorylates p- α -syn in a concentration-dependent manner (Fig. 3n), and the K_m , V_{max} , k_{cat} of APROM₂ are 0.175 mM, 0.0015 $\mu\text{M s}^{-1}$, and 0.764 s^{-1} (Fig. 3o,p, Supplementary Table 3), respectively. This suggests that APROM₂ can efficiently dephosphorylate p- α -syn, converting it into α -syn through *de novo* PTMs.”

• Figure 3

Figure 3. Asymmetric catalytic centers confer protein phosphatase-like characteristics to APROMs. **a,b**, Local coordination environments and corresponding differential charge density maps of OH absorbed CeNPs (**a**) and APROM (**b**). Red and blue represent accumulation and

depletion charge areas, respectively. **c**, Free-energy diagram of phospho-substrate hydrolysis on the APROM and CeNPs. **d,e**, Differential charge density maps of OH and phospho-substrate absorbed CeNPs (**d**) and APROM (**e**). **f**, Bader charge of the O atom of OH before and after phospho-substrate absorption. A positive value indicates that the atom gains electrons. **g**, DOS of CeNPs and the APROM. E = energy level, E_{Fermi} = Fermi level. **h,i**, PDOS of CeNPs (**h**) and the APROM (**i**). **j**, Protein phosphatases-mimetic activity of APROMs and CeNPs by using P-Ser as the phospho-substrate (n = 3 independent experiments). **k**, Raman spectra of APROM₂ with and without incubation of P-Ser. The P-Ser is used as the phospho-substrate for the dephosphorylation reaction. F_{2g} , the Raman-active vibrational mode of the cubic fluorite structure. **l**, The Ce-to-Mn ratio of APROM₂ before and after catalyzing phospho-substrate dephosphorylation. **m**, *In situ* XRD of APROM₂ under dephosphorylation reaction conditions. **n**, Protein phosphatases-mimetic activity of APROM₂ by using p- α -syn as the phospho-substrates (n = 3 independent experiments). **o,p**, Michaelis-Menten kinetics (**o**) and Lineweaver-Burk plotting (**p**) of APROM₂ obtained by adding different concentrations of p- α -syn (n = 3 independent experiments). **q**, Protein phosphatases-mimetic activity of APROM₂ by using P-Tyr and P-Thr as the phospho-substrates (n = 3 independent experiments). All the data are presented as means \pm s.e.m. Statistical significance was analyzed by one-way ANOVA with multiple comparisons test.

- Methods

“**Protein phosphatases-mimetic activity assay.** Additionally, the protein phosphatase-mimetic activity of APROM₂ (20 $\mu\text{g/mL}$) was also detected by using p- α -syn, O-phospho-L-tyrosine (P-Tyr) and O-phospho-L-threonine (P-Thr) as the phospho-substrates. The kinetic assays of APROM₂ (20 $\mu\text{g/mL}$) were carried out at 37 °C using a series of P-Ser concentrations (6.25, 12.5, 25, 50, 100 μM) or a series of p- α -syn concentrations (6.25, 12.5, 25, 50 μM). The Michaelis-Menten constant was calculated by using GraphPad Prism 8.0 (GraphPad Software).”

10) Page 14, Line 269: What was the phospho-substrate of the dephosphorylation reaction shown in figure 3k?

Response: *Thank you for your comments. In the dephosphorylation reaction shown in Fig. 3k, O-phospho-L-serine was employed as the phospho-substrate. To enhance clarity and address your concern, we have included an explanation of the phospho-substrate in Fig. 3k, the related text and the figure legend.*

Our modification to the manuscript: *The following sentences were modified on page 11, and Fig. 3k and figure legend were modified in the revised manuscript, respectively.*

- Page 11

“Raman spectra further confirm that the dephosphorylation of P-Ser occurred on asymmetric catalytic centers (Fig. 3k), validating the enhancement of catalytic performance by asymmetric

catalytic centers.”

- Figure 3

Figure 3. Asymmetric catalytic centers confer protein phosphatase-like characteristics to APROMs. a,b, Local coordination environments and corresponding differential charge density

maps of OH absorbed CeNPs (a) and APROM (b). Red and blue represent accumulation and depletion charge areas, respectively. c, Free-energy diagram of phospho-substrate hydrolysis on the APROM and CeNPs. d,e, Differential charge density maps of OH and phospho-substrate absorbed CeNPs (d) and APROM (e). f, Bader charge of the O atom of OH before and after phospho-substrate absorption. A positive value indicates that the atom gains electrons. g, DOS of CeNPs and the APROM. E = energy level, E_{Fermi} = Fermi level. h,i, PDOS of CeNPs (h) and the APROM (i). j, Protein phosphatases-mimetic activity of APROMs and CeNPs by using P-Ser as the phospho-substrate (n = 3 independent experiments). k, Raman spectra of APROM₂ with and without incubation of P-Ser. The P-Ser is used as the phospho-substrate for the dephosphorylation reaction. F_{2g} , the Raman-active vibrational mode of the cubic fluorite structure. l, The Ce-to-Mn ratio of APROM₂ before and after catalyzing phospho-substrate dephosphorylation. m, *In situ* XRD of APROM₂ under dephosphorylation reaction conditions. n, Protein phosphatases-mimetic activity of APROM₂ by using p- α -syn as the phospho-substrates (n = 3 independent experiments). o,p, Michaelis-Menten kinetics (o) and Lineweaver-Burk plotting (p) of APROM₂ obtained by adding different concentrations of p- α -syn (n = 3 independent experiments). q, Protein phosphatases-mimetic activity of APROM₂ by using P-Tyr and P-Thr as the phospho-substrates (n = 3 independent experiments). All the data are presented as means \pm s.e.m. Statistical significance was analyzed by one-way ANOVA with multiple comparisons test.

11) Page 14, Line 274: This should be experimentally demonstrated.

Response: *Thank you for your kind comment. According to your valuable suggestion, we tested the protein phosphatase-mimetic activity of APROM₂ by using p- α -syn as the phospho-substrate. As shown in Fig. 3n, APROM₂ efficiently dephosphorylates p- α -syn in a concentration-dependent manner through catalytically cleaving its phosphate monoester bonds, thereby converting p- α -syn into α -syn and free phosphate.*

Our modification to the manuscript: *The following sentences were added on page 11, and the results were added as Fig. 3n in the revised manuscript, respectively.*

• Page 11

“.....Inspired by the remarkable dephosphorylation capability of APROM₂, we further investigated its protein phosphatase-mimetic activity using p- α -syn as the phospho-substrate. As expected, APROM₂ readily dephosphorylates p- α -syn in a concentration-dependent manner (Fig. 3n), and the K_m , V_{max} , k_{cat} of APROM₂ are 0.175 mM, 0.0015 $\mu\text{M s}^{-1}$, and 0.764 s^{-1} (Fig. 3o,p, Supplementary Table 3), respectively. This suggests that APROM₂ can efficiently dephosphorylate p- α -syn, converting it into α -syn through *de novo* PTMs.”

• Figure 3

Figure 3. Asymmetric catalytic centers confer protein phosphatase-like characteristics to APROMs. **a,b**, Local coordination environments and corresponding differential charge density maps of OH absorbed CeNPs (**a**) and APROM (**b**). Red and blue represent accumulation and

depletion charge areas, respectively. **c**, Free-energy diagram of phospho-substrate hydrolysis on the APROM and CeNPs. **d,e**, Differential charge density maps of OH and phospho-substrate absorbed CeNPs (**d**) and APROM (**e**). **f**, Bader charge of the O atom of OH before and after phospho-substrate absorption. A positive value indicates that the atom gains electrons. **g**, DOS of CeNPs and the APROM. E = energy level, E_{Fermi} = Fermi level. **h,i**, PDOS of CeNPs (**h**) and the APROM (**i**). **j**, Protein phosphatases-mimetic activity of APROMs and CeNPs by using P-Ser as the phospho-substrate (n = 3 independent experiments). **k**, Raman spectra of APROM₂ with and without incubation of P-Ser. The P-Ser is used as the phospho-substrate for the dephosphorylation reaction. F_{2g} , the Raman-active vibrational mode of the cubic fluorite structure. **l**, The Ce-to-Mn ratio of APROM₂ before and after catalyzing phospho-substrate dephosphorylation. **m**, *In situ* XRD of APROM₂ under dephosphorylation reaction conditions. **n**, Protein phosphatases-mimetic activity of APROM₂ by using p- α -syn as the phospho-substrates (n = 3 independent experiments). **o,p**, Michaelis-Menten kinetics (**o**) and Lineweaver-Burk plotting (**p**) of APROM₂ obtained by adding different concentrations of p- α -syn (n = 3 independent experiments). **q**, Protein phosphatases-mimetic activity of APROM₂ by using P-Tyr and P-Thr as the phospho-substrates (n = 3 independent experiments). All the data are presented as means \pm s.e.m. Statistical significance was analyzed by one-way ANOVA with multiple comparisons test.

- Supplementary Table 3

Supplementary Table 3. Kinetics parameters of APROM₂ for protein phosphatase-mimetic activity.

Catalyst	Substrate	K_m (mM)	V_{max} ($\mu\text{M s}^{-1}$)	k_{cat} (s^{-1})
APROM ₂	P-Ser	39.17	0.49	2.47×10^2
APROM ₂	p- α -syn	0.175	1.5×10^{-3}	0.764

K_m is the Michaelis-Menten constant, V_{max} is the maximal reaction velocity, and k_{cat} is the catalytic constant.

12) Page 16, Line 296: Figure 4a is giving inhibition rate in per cent and not the superoxide dismutase (SOD)-mimetic activity of APROMs mentioned in the text. A comparison of the k_{cat} of the APROMS with the ones of the very efficient superoxide dismutases should be provided.

Response: Thank you for your suggestion. Figure 4a was done by the Total Superoxide Dismutase Assay Kit with WST (Dojindo, Japan). Firstly, 20 μL of catalysts with a final concentration of 20 $\mu\text{L mL}^{-1}$ or H_2O was mixed with 200 μL a 2-(4-iodophenyl)-3-(4-nitrophenyl)-5-(2,4-disulfophenyl)-2H tetrazolium sodium salt (WST-1) working solution in a microplate well. Then, 20 μL of the enzyme working solution was added. After incubation at 37 $^\circ\text{C}$ for 20 min, the absorbance at 450 nm was measured on a microplate reader (Bio Tech, USA). The inhibition rate of WST-1 was calculated by measuring the decrease in color

development, which also demonstrates the elimination rate of superoxide radicals (O_2^-). This method has been widely utilized in various articles to evaluate the SOD-mimetic activity of nanoparticles (references: *Nat. Commun.* 14, 160 (2023); *Adv. Funct. Mater.* 30, 2004692 (2020); *Adv. Funct. Mater.* 31, 2007130 (2021)). We have revised the Y-axis title of Figure 4a for more clarity, replacing “Inhibition rate (%)” with “ O_2^- Elimination rate of (%)”. Furthermore, we carried out steady-state kinetic assays for APROM₂ and natural SOD (Beyotime, China). As results, the k_{cat} for APROM₂ and natural SOD are 1.63×10^5 and $1.57 \times 10^2 \text{ s}^{-1}$, respectively. This finding demonstrates the significantly superior catalytic activity of APROM compared to that of natural SOD.

Our modification to the manuscript: The following sentence was modified on page 12, Fig. 4a was modified, the steady-state kinetics results of APROM₂ were added as Fig. 4b,c, and the method was added in the revised manuscript, respectively. Supplementary Table 4 was added in the revised supporting information.

- Page 12

“.....Remarkably, the SOD-mimetic activity of APROMs is enhanced, with APROM₂ performing best in scavenging superoxide radicals (O_2^-), consistent with the high Ce^{3+} -to- Ce^{4+} ratio induced by asymmetric catalytic centers (Fig. 4a). More importantly, the SOD-mimetic activity of APROM₂ is superior to that of natural SOD (Fig. 4b,c, Supplementary Table 4).”

• Figure 4

Figure 4. Asymmetric catalytic centers endow APROMs with exceptional antioxidant activity. a, SOD-mimetic activity of APROMs and CeNPs (n = 4 independent experiments).

b,c, Michaelis-Menten kinetics (**b**) and Lineweaver-Burk plotting (**c**) of APROM₂ obtained by adding different concentrations of xanthine (n = 3 independent experiments). **d**, CAT-mimetic activity of APROMs and CeNPs (n = 3 independent experiments). **e,f**, Michaelis-Menten kinetics (**e**) and Lineweaver-Burk plotting (**f**) of APROM₂ obtained by adding different concentrations of H₂O₂ (n = 3 independent experiments). **g**, The free-energy diagrams for H₂O₂ decomposition on the APROM and CeNPs. **h,i**, Differential charge density maps of H₂O₂ adsorbed CeNPs (**h**) and APROM (**i**). Red and blue represent accumulation and depletion charge areas, respectively. The isosurface level is 0.005 a.u. **j**, Schematic illustration of the catalytic mechanism of H₂O₂ decomposition in the asymmetric catalytic center. H₂O₂ is decomposed at the Ce site of the asymmetric catalytic center, and then the formed O₂ is transformed to the Mn site, so that the Ce site is regenerated for a new round of H₂O₂ decomposition. Moreover, O₂ can be readily desorbed at the Mn site with a low energy barrier. All the data are presented as means ± s.e.m. Statistical significance was analyzed by one-way ANOVA with multiple comparisons test.

- Methods

“**Superoxide dismutase (SOD)-mimetic activity assay.** The SOD-mimetic activity of APROMs and CeNPs (20 µg/mL) was tested by the Total Superoxide Dismutase Assay Kit with WST (Dojindo, Japan). Nitrotetrazolium blue chloride (NBT), an O₂⁻ sensitive probe, was used to study the kinetic assay of APROM₂ and natural SOD (Beyotime, China). APROM₂ or natural SOD were mixed with NBT (100 µg/mL), xanthine oxidase (0.3 U/mL), and various concentrations of xanthine (0.01875, 0.0375, 0.075, 0.15, 0.3 mM) in tris-HCl buffer (0.1 M). The mixed solution was continuously monitored the absorbance at 550 nm by using a UV-Vis spectrophotometer UV-2600 (Shimadzu, Japan). The Michaelis-Menten constant was calculated by using GraphPad Prism 8.0 (GraphPad Software).”

- Supplementary Table 4

Supplementary Table 4. Kinetics parameters of APROM₂ and natural SOD with xanthine as the substrate for SOD-mimetic activity.

Catalyst	K_m (mM)	V_{max} (µM s ⁻¹)	k_{cat} (s ⁻¹)
APROM ₂	50.70	32.46	1.63×10^5
Natural SOD	7.63	4.20	1.57×10^2

K_m is the Michaelis-Menten constant, V_{max} is the maximal reaction velocity, and k_{cat} is the catalytic constant.

13) Page 16, Line 300: A definition of CAT-mimetic activity is missing and for the CAT-mimetic activity of APROM₂ k_{cat} should be provided, in addition to v_{max} (also for CeNP and natural CAT)

Response: Thank you for your valuable comments. According to your kind suggestions, we have added the definition of CAT-mimetic activity, which refers to the decomposition of H₂O₂

into H₂O and O₂. Per your suggestion, we also calculated the kinetics parameters of natural CAT, CeNPs, and APROM₂ with H₂O₂ as the substrate for CAT-mimetic activity, including the V_{max} and k_{cat} . As shown in supplement table 5, the V_{max} for natural CAT, CeNPs and APROM₂ are 18.88, 0.82, 3.87 $\mu\text{M s}^{-1}$, respectively. The k_{cat} for natural CAT, CeNPs and APROM₂ are 7.90×10^4 , 1.81×10^3 , and $3.88 \times 10^3 \text{ s}^{-1}$ respectively.

Our modification to the manuscript: The following sentences were modified on page 12 in the revised manuscript. The steady-state kinetics results of CeNPs were added as Supplementary Fig. 12, and Supplementary Table 5 was modified in the revised supporting information, respectively.

- Page 12

“Notably, different from the downregulated CAT-mimetic activity affected by decreasing surface Ce⁴⁺ levels⁴⁴, APROMs exhibit superior CAT-mimetic activity in decomposing H₂O₂ into H₂O and O₂ compared to CeNPs, despite a lower Ce⁴⁺-to-Ce³⁺ ratio, which possibly due to the presence of asymmetric catalytic centers (Fig. 4d). It's worth mentioning that APROM₂ demonstrates a CAT-mimetic activity approximately 14.8-fold higher than that of CeNPs, with an affinity for H₂O₂ approximately 3.34-fold higher than that of natural CAT enzyme (Fig. 4e,f, Supplementary Figs. 12 and 13, Supplementary Table 5), as confirmed by steady-state kinetics.”

- Supplementary Figure 12

Supplementary Figure 12. Michaelis-Menten kinetics (a) and Lineweaver-Burk plotting (b) of CeNPs obtained by adding different concentrations of H₂O₂ (n = 3 independent experiments). Data are presented as means \pm s.e.m.

- Supplementary Table 5

Supplementary Table 5. Kinetics parameters of APROM₂ and CeNPs with H₂O₂ as the substrate for CAT-mimetic activity.

Catalyst	K_m (mM)	V_{max} ($\mu\text{M s}^{-1}$)	k_{cat} (s^{-1})
Natural CAT	51.57	18.88	7.90×10^4
CeNPs	53.53	0.82	1.81×10^3
APROM ₂	15.42	3.87	3.88×10^3

K_m is the Michaelis-Menten constant, V_{max} is the maximal reaction velocity, and k_{cat} is the catalytic constant.

14) Page 19, Line 346: This statement should be improved and be made more specific.

Response: *Thank you for your kind suggestion. The C-terminal region (residues 96–140) of α -syn, which contains motifs of charged amino acids, has the ability to bind to metal ions, including Mn, Fe and Cu (reference: *J. Am. Chem. Soc.* 128, 9893-9901 (2006)). Importantly, phosphorylation at Ser 129 alters the charge, hydrogen-binding patterns, and backbone angles of the C-terminal region, increasing the affinity between metal ions and the C-terminal region of α -syn (reference: *J. Parkinson. Dis.* 6, 39-51 (2016)). Therefore, heterogeneous catalysts are more likely to bind with p- α -syn. We have incorporated these specific improvements into the statement on page 19 in the revised manuscript.*

Our modification to the manuscript: *The following sentences were modified on page 13, and the reference is added in the revised manuscript.*

- Page 13

“.....The C-terminal region of α -syn, which contains motifs of charged amino acids, is capable of binding to metal ions⁴⁶. Notably, phosphorylation of Ser 129 within the C-terminal region significantly improves this affinity⁴⁷. In line with these observations, APROM₂ remarkably dephosphorylates p- α -syn with decreasing the p- α -syn level approximate to the physiological level via effective interaction with p- α -syn, thereby rebalancing the aberrant PTMs.”

- Reference

46. Binolfi, A. et al. Interaction of α -synuclein with divalent metal ions reveals key differences: A link between structure, binding specificity and fibrillation enhancement. *J. Am. Chem. Soc.* 128, 9893-9901 (2006).

15) Page 19, Line 348: A figure for the APROM₂-catalyzed dephosphorylation kinetics of p- α -syn should be added.

Response: *Thank you for your valuable suggestion. According to your suggestion, we carried out the kinetic assays for APROM₂. The initial rates of hydrolysis reaction catalyzed by APROM₂ were measured at different p- α -syn concentrations (6.25, 12.5, 25, 50 μ M). The Michaelis-Menten kinetics and Lineweaver-Burk plotting of APROM₂ were obtained by using GraphPad Prism 8.0 (GraphPad Software), and then the K_m , V_{max} , and k_{cat} were calculated. The results show that the K_m , V_{max} , and k_{cat} for APROM₂ were 0.175 mM, 0.0015 μ M s⁻¹, and 0.764 s⁻¹, respectively.*

Our modification to the manuscript: *The following sentences were added on page 11, the steady-state kinetics results of APROM₂ were added as Fig. 3o,p, and the method was added in the revised manuscript. The steady-state kinetics results of APROM₂ were added as*

Supplementary Table 3 in the revised supporting information.

- Page 11

“.....Inspired by the remarkable dephosphorylation capability of APROM₂, we further investigated its protein phosphatase-mimetic activity using p- α -syn as the phospho-substrate. As expected, APROM₂ readily dephosphorylates p- α -syn in a concentration-dependent manner (Fig. 3n), and the K_m , V_{max} , k_{cat} of APROM₂ are 0.175 mM, 0.0015 $\mu\text{M s}^{-1}$, and 0.764 s^{-1} (Fig. 3o,p, Supplementary Table 3), respectively. This suggests that APROM₂ can efficiently dephosphorylate p- α -syn, converting it into α -syn through *de novo* PTMs.”

- Methods

“**Protein phosphatases-mimetic activity assay.**Additionally, the protein phosphatase-mimetic activity of APROM₂ (20 $\mu\text{g/mL}$) was also detected by using p- α -syn, O-phospho-L-tyrosine (P-Tyr) and O-phospho-L-threonine (P-Thr) as the phospho-substrates. The kinetic assays of APROM₂ (20 $\mu\text{g/mL}$) were carried out at 37 °C using a series of P-Ser concentrations (6.25, 12.5, 25, 50, 100 μM) or a series of p- α -syn concentrations (6.25, 12.5, 25, 50 μM). The Michaelis-Menten constant was calculated by using GraphPad Prism 8.0 (GraphPad Software).”

• Figure 3

Figure 3. Asymmetric catalytic centers confer protein phosphatase-like characteristics to APROMs. **a,b**, Local coordination environments and corresponding differential charge density maps of OH absorbed CeNPs (**a**) and APROM (**b**). Red and blue represent accumulation and

depletion charge areas, respectively. **c**, Free-energy diagram of phospho-substrate hydrolysis on the APROM and CeNPs. **d,e**, Differential charge density maps of OH and phospho-substrate absorbed CeNPs (**d**) and APROM (**e**). **f**, Bader charge of the O atom of OH before and after phospho-substrate absorption. A positive value indicates that the atom gains electrons. **g**, DOS of CeNPs and the APROM. E = energy level, E_{Fermi} = Fermi level. **h,i**, PDOS of CeNPs (**h**) and the APROM (**i**). **j**, Protein phosphatases-mimetic activity of APROMs and CeNPs by using P-Ser as the phospho-substrate (n = 3 independent experiments). **k**, Raman spectra of APROM₂ with and without incubation of P-Ser. The P-Ser is used as the phospho-substrate for the dephosphorylation reaction. F_{2g} , the Raman-active vibrational mode of the cubic fluorite structure. **l**, The Ce-to-Mn ratio of APROM₂ before and after catalyzing phospho-substrate dephosphorylation. **m**, *In situ* XRD of APROM₂ under dephosphorylation reaction conditions. **n**, Protein phosphatases-mimetic activity of APROM₂ by using p- α -syn as the phospho-substrates (n = 3 independent experiments). **o,p**, Michaelis-Menten kinetics (**o**) and Lineweaver-Burk plotting (**p**) of APROM₂ obtained by adding different concentrations of p- α -syn (n = 3 independent experiments). **q**, Protein phosphatases-mimetic activity of APROM₂ by using P-Tyr and P-Thr as the phospho-substrates (n = 3 independent experiments). All the data are presented as means \pm s.e.m. Statistical significance was analyzed by one-way ANOVA with multiple comparisons test.

- Supplementary Table 3

Supplementary Table 3. Kinetics parameters of APROM₂ for protein phosphatase-mimetic activity.

Catalyst	Substrate	K_m (mM)	V_{max} ($\mu\text{M s}^{-1}$)	k_{cat} (s^{-1})
APROM ₂	P-Ser	39.17	0.49	2.47×10^2
APROM ₂	p- α -syn	0.175	1.5×10^{-3}	0.764

K_m is the Michaelis-Menten constant, V_{max} is the maximal reaction velocity, and k_{cat} is the catalytic constant.

16) Page 24, Line 435: What about the effect of APROM₂ on the dephosphorylation of α -syn phosphorylated at other phosphorylation sites, such as S87, Y125, S129, Y133, and Y136?

Response: Thank you for your comment. We have confirmed that APROM₂ can dephosphorylate α -syn that is phosphorylated at S129 both *in vitro* and *in vivo* (Fig. 5b,c and Fig. 6i). Given APROM₂'s ability to dephosphorylate P-Ser (Fig. 3j-m), it has potential to dephosphorylate α -syn phosphorylated at S87. In order to investigate the effect of APROM₂ on the dephosphorylation of α -syn phosphorylated at tyrosine residues, such as Y125, Y133 and Y136, we have further studied the protein phosphatase-mimetic activity of APROM₂ by using O-phospho-L-tyrosine (P-Tyr) as the substrate. The results show that APROM₂ effectively dephosphorylates P-Tyr, indicating its potential to dephosphorylate α -syn phosphorylated at tyrosine residues.

Protein phosphorylation plays a pivotal role in regulating protein functions by modulating protein-protein interactions. Accumulating evidence has demonstrated that phosphorylation of α -syn affects its functions (references: Mol. Neurodegener. 12, 45 (2017); J Neurosci. 30, 3184-3198 (2010); Int. J. Mol. Sci., 24, 13270 (2023); ACS Chem. Neurosci. 5, 1203-1208 (2014)). For example, phosphorylated at S87 may influence the normal function of α -syn, by perturbing its membrane-bound conformation or modulating its protein-protein interactions (reference: J Neurosci. 30, 3184-3198 (2010)). Phosphorylation of α -syn at S129 and S87 reduces its binding capacity to the membrane (reference: Mol. Neurodegener. 12, 45 (2017)). Additionally, phosphorylation of α -syn at Y125 argues S129 phosphorylation under physiological in vivo conditions (reference: ACS Chem. Neurosci. 5, 1203-1208 (2014)). Since phosphorylation of S129 is the dominant pathological modification of α -synuclein in PD, we focus our study on this particular phosphorylation site.

Our modification to the manuscript: *The result of protein phosphatase-mimetic activity of APROM₂ by using P-Tyr as the substrate was added as Fig. 3q in the revised manuscript.*

• Figure 3

Figure 3. Asymmetric catalytic centers confer protein phosphatase-like characteristics to APROMs. **a,b**, Local coordination environments and corresponding differential charge density maps of OH absorbed CeNPs (**a**) and APROM (**b**). Red and blue represent accumulation and

depletion charge areas, respectively. **c**, Free-energy diagram of phospho-substrate hydrolysis on the APROM and CeNPs. **d,e**, Differential charge density maps of OH and phospho-substrate absorbed CeNPs (**d**) and APROM (**e**). **f**, Bader charge of the O atom of OH before and after phospho-substrate absorption. A positive value indicates that the atom gains electrons. **g**, DOS of CeNPs and the APROM. E = energy level, E_{Fermi} = Fermi level. **h,i**, PDOS of CeNPs (**h**) and the APROM (**i**). **j**, Protein phosphatases-mimetic activity of APROMs and CeNPs by using P-Ser as the phospho-substrate (n = 3 independent experiments). **k**, Raman spectra of APROM₂ with and without incubation of P-Ser. The P-Ser is used as the phospho-substrate for the dephosphorylation reaction. F_{2g} , the Raman-active vibrational mode of the cubic fluorite structure. **l**, The Ce-to-Mn ratio of APROM₂ before and after catalyzing phospho-substrate dephosphorylation. **m**, *In situ* XRD of APROM₂ under dephosphorylation reaction conditions. **n**, Protein phosphatases-mimetic activity of APROM₂ by using p- α -syn as the phospho-substrates (n = 3 independent experiments). **o,p**, Michaelis-Menten kinetics (**o**) and Lineweaver-Burk plotting (**p**) of APROM₂ obtained by adding different concentrations of p- α -syn (n = 3 independent experiments). **q**, Protein phosphatases-mimetic activity of APROM₂ by using P-Tyr and P-Thr as the phospho-substrates (n = 3 independent experiments). All the data are presented as means \pm s.e.m. Statistical significance was analyzed by one-way ANOVA with multiple comparisons test.

17) Page 27, Line 486: Which of these multiple protein phosphatase functions can be performed by APROMs?

Response: Thank you for your comment. Based on the amino acidic residues they dephosphorylate, protein phosphatases are traditionally classified into two major groups, protein serine/threonine (Ser/Thr) phosphatases (PSPs) and protein tyrosine phosphatases (PTPs) (references: *Moll. Cell* 65, 347-360 (2017); *Biochim. Biophys. Acta Rev. Cancer* 1876, 188562 (2021); *Cancer Lett.* 335, 9-18 (2013)). All of them exert function through dephosphorylating phospho-Ser residues, phospho-Thr residues or phospho-Tyr residues by cleaving phosphate monoester bonds, thereby working with kinases to regulate reversible protein phosphorylation (references: *Cell* 139, 468-484 (2009); *Cell* 117, 699-711 (2004)). Accumulating evidence demonstrates that ceria-based catalysts can bind with both hydroxyl groups (or H_2O) and phospho-substrates, and promote the nucleophilic attack of hydroxyl groups (or H_2O) toward the phosphate group, thereby catalytically cleaving phosphate monoester bonds (references: *Adv. Sci.* 8, 2004115 (2021); *ACS Catal.* 13, 504-514 (2023)). We investigated the protein phosphatase-mimetic activity of the APROM by using O-phospho-L-serine (P-Ser), O-phospho-L-tyrosine (P-Tyr) and O-phospho-L-threonine (P-Thr) as phospho-substrates (Fig. 3j,q). As results, the APROM catalytically cleaves the phosphate monoester bonds of P-Ser, P-Tyr and P-Thr, exhibiting strong catalytic activity in dephosphorylation. Collectively, these findings demonstrate that the APROM resembles the

dephosphorylation function of protein phosphatases (PSPs and PTPs) in dephosphorylating phosphorylated amino acidic residues, including phospho-Ser residues, phospho-Thr residues and phospho-Tyr residues.

Our modification to the manuscript: *The following sentences were added on page 11, and the protein phosphatase-mimetic activity of APROM₂ by using O-phospho-L-tyrosine (P-Tyr) and O-phospho-L-threonine (P-Thr) as phospho-substrates was added as Fig. 3q in the revised manuscript.*

- Page 11

“.....Moreover, APROM₂ exhibits pronounced dephosphorylation activity towards O-phospho-L-tyrosine (P-Tyr) and O-phospho-L-threonine (P-Thr), effectively cleaving the phosphate monoester bonds (Fig. 3q). This collective evidence solidifies the notion that APROM₂ faithfully emulates the functionality of protein phosphatases, encompassing both protein serine/threonine phosphatases and protein tyrosine phosphatases.”

• Figure 3

Figure 3. Asymmetric catalytic centers confer protein phosphatase-like characteristics to APROMs. **a,b**, Local coordination environments and corresponding differential charge density maps of OH absorbed CeNPs (**a**) and APROM (**b**). Red and blue represent accumulation and

depletion charge areas, respectively. **c**, Free-energy diagram of phospho-substrate hydrolysis on the APROM and CeNPs. **d,e**, Differential charge density maps of OH and phospho-substrate absorbed CeNPs (**d**) and APROM (**e**). **f**, Bader charge of the O atom of OH before and after phospho-substrate absorption. A positive value indicates that the atom gains electrons. **g**, DOS of CeNPs and the APROM. E = energy level, E_{Fermi} = Fermi level. **h,i**, PDOS of CeNPs (**h**) and the APROM (**i**). **j**, Protein phosphatases-mimetic activity of APROMs and CeNPs by using P-Ser as the phospho-substrate (n = 3 independent experiments). **k**, Raman spectra of APROM₂ with and without incubation of P-Ser. The P-Ser is used as the phospho-substrate for the dephosphorylation reaction. F_{2g} , the Raman-active vibrational mode of the cubic fluorite structure. **l**, The Ce-to-Mn ratio of APROM₂ before and after catalyzing phospho-substrate dephosphorylation. **m**, *In situ* XRD of APROM₂ under dephosphorylation reaction conditions. **n**, Protein phosphatases-mimetic activity of APROM₂ by using p- α -syn as the phospho-substrates (n = 3 independent experiments). **o,p**, Michaelis-Menten kinetics (**o**) and Lineweaver-Burk plotting (**p**) of APROM₂ obtained by adding different concentrations of p- α -syn (n = 3 independent experiments). **q**, Protein phosphatases-mimetic activity of APROM₂ by using P-Tyr and P-Thr as the phospho-substrates (n = 3 independent experiments). All the data are presented as means \pm s.e.m. Statistical significance was analyzed by one-way ANOVA with multiple comparisons test.

18) Page 28, Line 491: The statement on the “efficient rebalancing of aberrant PTMs of phospho-proteins” is too general and should be made more specific.

Response: *Thank you for your kind suggestion. Accumulating evidence demonstrates that cerium ions can bind with the phosphate group and hydrolyze the phosphate monoester bonds of phospho-substrates (references: Adv. Sci. 8, 2004115 (2021); Coord. Chem. Rev. 382 145-159 (2019)). In addition, our experimental results demonstrate that APROM₂ is capable of dephosphorylating P-Ser, P-Tyr, and P-Thr by cleaving the phosphate monoester bonds. Therefore, APROM enables efficient rebalancing of aberrant PTMs of phospho-proteins that possess phosphate monoester bonds at amino acid residues like serine, tyrosine and threonine, through a de novo PTM strategy. We have revised the statement on page 28 to make it more specific.*

Our modification to the manuscript: *The following sentence was modified on page 18 in the revised manuscript.*

- Page 18

“These APROMs enable efficient rebalancing of aberrant PTMs of phospho-proteins that possess phosphate monoester bonds at amino acid residues like serine, tyrosine and threonine, through a *de novo* PTM strategy.”

19) Page 28, Line 495: This “avenue for reprogramming protein function using the de novo PTM strategy” should be discussed in more detail.

Response: *Thank you for your helpful suggestion. The de novo PTM strategy involves the dephosphorylation of hyperphosphorylated proteins by APROMs. We have modified the sentence for more clarity.*

Our modification to the manuscript: *The following sentence was modified on page 18 in the revised manuscript.*

- Page 18

“Our findings offer a promising avenue for reprogramming protein function using the *de novo* PTM strategy, which involves the dephosphorylation of hyperphosphorylated proteins by APROMs, with the potential to revolutionize the treatment of neurological disorders.”

20) Page 28, Line 498: The “precise rebalancing of aberrant PTMs” should be specified.

Response: *Thank you for your kind suggestion. We have refined the phrase “precise rebalancing of aberrant PTMs” to “precise rebalancing of aberrant PTMs of the proteins that contain phosphate monoester bonds at amino acid residues like serine, tyrosine and threonine” in the revised manuscript.*

Our modification to the manuscript: *The following sentence was modified on page 19 in the revised manuscript.*

- Page 19

“.....Future advancements focusing on specific protein-targeted APROMs enable precise rebalancing of aberrant PTMs of the proteins that contain phosphate monoester bonds at amino acid residues like serine, tyrosine and threonine within affected areas, thereby reprogramming protein function and maintaining proteostasis for effective disease therapy.”

REVIEWER COMMENTS

Reviewer #1 (Remarks to the Author):

Comments to authors:

The development of APROM remains an excellent achievement, suitable for a chemically oriented journal. It is also appreciated that the authors made an effort to revise, but the key request – to strengthen the aSyn biology part – was unfortunately largely ignored, and from the responses it is not clear if the authors even understood the concerns fully.

The current manuscript that implies that there is a realistic chance that APROM can be of therapeutic use for the treatment of synucleinopathies can in our opinion not be published in Nature Communications:

- The role of pS129 aSyn in health and disease is not understood and there is no clear evidence that reducing pS129 is the way to go. The authors have cherry-picked some publications that claim this and have settled on an ill-defined interplay between VMAT and VAMP2, but this will not be convincing to the field of synucleinopathies – a field that we know very well.
- The pleiotropic nature of APROM remains an unresolved issue despite the authors' referral to heatmaps of different pathways. The proposed relative specificity of the interaction with pS129 because aSyn can bind metal ions in its C-terminus is not convincing. The authors themselves state: “APROM reprograms the biological function of neuronal proteins to fuel synaptic function via de novo PTMs” – this implies a plethora of “de novo PTMs” caused by APROM, beyond aSyn. And we should simply assume that all these “de novo PTMs” are beneficial?
- The authors have been reluctant to improve the manuscript concerning the synuclein parts. Reading through the rebuttal, the authors' knowledge about synuclein pathophysiology appears too shallow. Otherwise, they would not make tall claims as they did in their manuscript. In their rebuttal, for the most part, the authors seem to explain the concerns (raised by this reviewer) in detail to the reviewer rather than respond to the concerns convincingly. Our strong suggestion to collaborate with someone in the aSyn field was ignored. In the current form, the aSyn claims will not be suitable to be presented to the aSyn field.
- On many occasions the experimental evidence provided does not justify the claims made, and the scientific reasoning, despite citing lots of literature, seems

The achievement of having developed APROM remains as a strong achievement and the authors are encouraged to have a fresh look at presenting its value to the field of chemical biology – without implying a direct use for the treatment of synucleinopathies.

Reviewer #2 (Remarks to the Author):

Reviewer #3 (Remarks to the Author):

The revised paper thoroughly addresses my previous feedback. I have no further comments.

Reviewer #4 (Remarks to the Author):

The authors have addressed the reviewer comments and have improved the manuscript considerably. The requested kinetic data have been determined by performing additional experiments. Some statements (see below) in the revised manuscript still need to be corrected and improved.

Comments to Manuscript NCOMMS-23-36937-T REVISION

1) Page 3, Line 66: There are still some issues with the revised formulation “While small molecule asymmetric catalysts can selectively generate favorable intermediates for efficient catalytic reactions, their practical biomedical applications are hindered by challenges such as low solubility and off-target side effects.” Do the authors mean chiral organocatalysts with the term “small molecule asymmetric catalysts” as they cite reference 27, or do they also include under this term “metal complexes with chiral ligands”? The statement that their practical biomedical applications are hindered by challenges such as low solubility and off-target side effects is too general and not sufficiently detailed, as well-known chiral organocatalysts are very water-soluble, and an example of off-target side effects of chiral organocatalysts should be cited.

2) Page 4, Line 70: While in the previous sentence the term “asymmetric molecular catalysts” has been replaced with “small-molecule asymmetric catalysts”, this statement that “... nanomaterials based heterogeneous catalysts show promise in overcoming the inherent limitations of molecular catalysts due to their modifiable surface” is not the appropriate comparison and should be reformulated.

REVIEWER COMMENTS

Reviewer #1 (Remarks to the Author):

Comments to authors:

The development of APROM remains an excellent achievement, suitable for a chemically oriented journal. It is also appreciated that the authors made an effort to revise, but the key request – to strengthen the aSyn biology part – was unfortunately largely ignored, and from the responses it is not clear if the authors even understood the concerns fully.

The current manuscript that implies that there is a realistic chance that APROM can be of therapeutic use for the treatment of synucleinopathies can in our opinion not be published in Nature Communications:

- The role of pS129 aSyn in health and disease is not understood and there is no clear evidence that reducing pS129 is the way to go. The authors have cherry-picked some publications that claim this and have settled on an ill-defined interplay between VMAT and VAMP2, but this will not be convincing to the field of synucleinopathies – a field that we know very well.

- The pleiotropic nature of APROM remains an unresolved issue despite the authors' referral to heatmaps of different pathways. The proposed relative specificity of the interaction with pS129 because aSyn can bind metal ions in its C-terminus is not convincing. The authors themselves state: ““APROM reprograms the biological function of neuronal proteins to fuel synaptic function via de novo PTMs” – this implies a plethora of “de novo PTMs” caused by APROM, beyond aSyn. And we should simply assume that all these “de novo PTMs” are beneficial?

- The authors have been reluctant to improve the manuscript concerning the synuclein parts. Reading through the rebuttal, the authors' knowledge about synuclein pathophysiology appears too shallow. Otherwise, they would not make tall claims as they did in their manuscript. In their rebuttal, for the most part, the authors seem to explain the concerns (raised by this reviewer) in detail to the reviewer rather than respond to the concerns convincingly. Our strong suggestion to collaborate with someone in the aSyn field was ignored. In the current form, the aSyn claims will not be suitable to be presented to the aSyn field.

- On many occasions the experimental evidence provided does not justify the claims made, and the scientific reasoning, despite citing lots of literature, seems

The achievement of having developed APROM remains as a strong achievement and the authors are encouraged to have a fresh look at presenting its value to the field of chemical biology – without implying a direct use for the treatment of synucleinopathies.

Response: *Thank you very much for your valuable comments. Following your suggestions, we carefully reconsidered how we present APROM's significance in the field of chemical biology and removed any implication that it can be directly applied to treat synucleinopathies. We understand and appreciate your concerns about the synuclein biology aspect, and tried our best to discuss with experts in the aSyn field at SJTU. In the revised manuscript, we have made several adjustments to address the key issues raised. Here are the specific actions we have taken:*

Role of p- α -syn: *Phosphorylation at S129 of α -syn increases from ~4% under physiological conditions to ~90% in neurodegenerative disorders (reference: Nat. Cell Biol. 4, 160-164 (2002)). Therefore, there is a reasonable basis to associate pS129 α -synuclein (p- α -syn) with the progression of PD. Indeed, although the role of p- α -syn has been extensively investigated, the precise role of pS129 α -synuclein remains unclear and controversial due to conflicting results in the literatures (references: Proc. Natl. Acad. Sci. U. S. A. 119, e2109617119 (2022); J. Biol. Chem. 283, 23179-23188 (2008); J. Neurosci. 32, 1536-1544 (2012)). These discrepancies may arise from differences in experimental models. In our manuscript, we used 1-methyl-4-phenylpyridinium (MPP⁺, a metabolite of MPTP) and MPTP to establish the PD phenotype cell and PD mice models, respectively, as MPTP is the only known dopaminergic neurotoxin capable of causing a clinical picture in both humans and monkeys indistinguishable from PD (reference: Nat. Protoc. 2, 141-151 (2007)). We found that the p- α -syn levels increase upon MPP⁺ exposure, while APROM remarkably dephosphorylates intracellular p- α -syn. In addition, we noted a restoration of synaptic transmission of APROM-treated neurons. Considering the diverse roles of α -syn in synaptic maintenance and plasticity, neurotransmitter release and homeostasis, and the regulation of synaptic vesicle pools, we hypothesize that the dephosphorylation of p- α -syn by APROM attributes to the restoration of synaptic transmission. Numerous studies have proved that α -syn can bind to vesicular monoamine transporter 2 (VMAT2) (reference: Cell Mol. Neurobiol. 28, 35-47 (2008); Aging Cell 18, e13031 (2019)), facilitating the incorporation of dopamine into synaptic vesicles, and interacts with vesicle-associated membrane protein 2 (VAMP2) (reference: Science 329, 1663-1667 (2010)), promoting vesicle-membrane fusion for dopamine release. To prove our hypothesis, we tested the colocalization of α -syn with VMAT2 and VAMP2. Surprisingly, we found that the colocalization of α -syn with VMAT2 and VAMP2 increases with APROM treatment, and the formation of α -syn-VMAT2 complex dependent on dephosphorylation. We agree with the reviewer that while our data support the hypothesis that dephosphorylation of p- α -syn can restore its biological function in MPP⁺/MPTP models, its applicability to all PD models and for the completely treatment of synucleinopathies require further investigation. Brain science is enigmatic,*

and exploration continues in the realm of the unknown. Although the role of p- α -syn is not fully elucidated, we strongly believe our findings hold significance and may offer a novel avenue for future PD treatment. Accordingly to your kind comments and suggestions, we have further modified the Results and Discussion sections for more clarity and accuracy.

The pleiotropic nature of APROM: *We acknowledge and fully agree with the reviewer. Since APROM demonstrates the ability to dephosphorylate O-phospho-L-serine (P-Ser), O-phospho-L-tyrosine (P-Tyr), and O-phospho-L-threonine (P-Thr), it possesses pleiotropic effects. In our work, APROM does not specifically catalyze the dephosphorylation of p- α -syn. We observed a significant dephosphorylation of p- α -syn in vitro and in vivo by APROM through mimetic activity assessment, immunofluorescence, and Western blot analysis, probably due to the substantial accumulation of p- α -syn and the heightened affinity of p- α -syn to metal ions. In fact, phosphoproteomic analysis results show that the phosphorylation levels of some metabolism-related proteins in the midbrain of PD mice treated with APROM are lower than those in normal mice, likely attributed to the pleiotropic effects of APROM, damage caused by MPTP or other factors. The plethora of “de novo PTMs” caused by APROM may potentially be detrimental to neurons. Nevertheless, compared to PD mice, there is no discernible reduction in phosphorylation levels of aforementioned proteins in APROM-treated PD mice. Collective results indicate that APROM holds potential in disease treatments. Indeed, the pleiotropic effect mentioned by the reviewer is a critical issue in the context of therapeutic drug development for PD. Although APROM, as a potent artificial protein modulator, demonstrates potential in protein reprogramming, it still has a long way to go in terms of practical application in clinical disease therapy. Despite our preliminary in vitro and in vivo data showing the reprogramming of α -syn by APROM, future advancements in the development of specific protein-targeted APROM, leveraging its modifiable characteristics, are imperative for precisely targeting and rebalancing aberrant PTMs of this individual protein within affected areas. We have added more discussions regarding the deficiency and the further improvement of APROM in the revised Discussion section, and have modified the analysis of heatmaps in the revised manuscript for more clarity and accuracy.*

Improvement of the manuscript concerning the synuclein parts: *According to your suggestions, we have revised the claims both of α -syn and p- α -syn in the revised manuscript for more clarity and accuracy.*

Improvement of the manuscript concerning the claims and the scientific reasoning: *According to your suggestions, we have revised the claims and the scientific reasoning*

as described above in the revised manuscript, aligning them more closely with the experimental results.

We appreciate your careful and thoughtful review, and we believe that these 2nd revisions improved the overall quality of our manuscript. Hopefully we may have chance to collaborate with you to solve the real challenges in the aSyn field in our future work.

Our modification to the manuscript: The following sentences were modified and added on pages 2, 4, 13, 14, 15, 17, 19, and the references were added in the revised manuscript, respectively.

- Page 2

“.....Our findings offer insights into in situ protein reprogramming and provide a promising avenue for developing **potential** therapeutic strategies targeting aberrant PTMs and treating neurological disorders.”

- Page 4

“.....These findings highlight the potential for developing asymmetric heterogeneous catalysts as artificial protein modulators, and provide a **promising** platform for the treatment of neurological disorders through reprogramming neuronal protein functions.”

- Page 13

“APROM₂ reprograms the biological function of α -syn to fuel synaptic function via *de novo* PTMs. Building upon the remarkable protein phosphatase-like characteristics of APROM₂, we further investigated its capacity to reprogram neuronal protein functions through *de novo* PTMs (Fig. 5a). **As a proof-of-concept, we chose α -syn as the neuronal protein of interest.**”

- Page 14

“.....Subsequently, the impact of APROM₂ on synaptic transmission was assessed using the fluorescent dye FM1-43, which is incorporated into synaptic vesicles during endocytosis and released during vesicle-membrane fusion (Fig. 5d,e). Surprisingly, APROM₂ significantly enhances synaptic transmission. Given that α -syn is essential for maintaining synaptic homeostasis^{11,48}, we hypothesize that the dephosphorylation of p- α -syn by APROM₂ through *de novo* PTMs attributes to the restoration of synaptic transmission. Emerging evidence suggests that α -syn, primarily localized at presynaptic terminals, binds to vesicular monoamine transporter 2 (VMAT2)¹¹, facilitating the incorporation of dopamine into synaptic vesicles, and interacts with vesicle-associated membrane protein 2 (VAMP2)⁴⁸, promoting vesicle-membrane fusion for dopamine release. **To test our hypothesis,** the colocalization of α -syn with VMAT2 and VAMP2 was examined (Fig. 5f-i). The ability of α -syn to bind with VMAT2 and VAMP2 is significantly impaired following MPP⁺ exposure, which is substantially restored by

treatment with APROM₂, confirming APROM₂'s ability to reprogram the biological function of α -syn in MPP⁺-induced cell model of PD.”

- Page 15

“.....Collectively, APROM₂ fuels synaptic function through *de novo* PTMs of p- α -syn and protects mitochondria in MPP⁺-induced cell model of PD, thereby safeguarding neurons against degeneration (Fig. 5o).”

- Page 17

“These collective results strongly support the notion that APROM₂ can promote the synaptic plasticity of dopaminergic neurons in the MPTP mouse model of PD.”

“.....As indicated in phosphoproteomic analysis, the phosphorylation levels of some metabolism-related proteins in the midbrain of APROM₂-treated PD mice are lower than those in normal mice, probably attributed to the pleiotropic effects of APROM₂, damage caused by MPTP or other factors (Supplementary Figs. 30-34). However, it is worth noting that, compared to PD mice, there is no discernible reduction in phosphorylation levels of aforementioned proteins in APROM₂-treated PD mice. Above all, these results demonstrate that APROM₂ exhibits excellent biocompatibility and does not induce local or systemic toxicity.”

- Page 18

“.....As a proof-of-concept, we found the restoration of α -syn biological function using APROMs, resulting in enhanced synaptic plasticity in MPP⁺-treated primary neurons and the rescue of dopaminergic neurons in MPTP mouse model of PD. While our data support the hypothesis that the dephosphorylation of p- α -syn can restore its biological function in MPP⁺/MPTP models of PD, a more in-depth investigation in other PD models is still needed, given the ongoing controversy regarding the pathogenic relevance of p- α -syn⁵⁴⁻⁵⁶.”

- Page 19

“.....APROMs, as potent tool for protein programming, still have a long way to go in terms of practical application in disease therapy, primarily due to its pleiotropic nature. While our preliminary in vitro and in vivo data demonstrate the direct reprogramming of α -syn by APROMs, future advancements in developing specific protein-targeted APROMs, capitalizing on its modifiable characteristics, are essential. These advancements will enable precise targeting and rebalancing of aberrant PTMs, particularly those involving phosphate monoester bonds at amino acid residues like serine, tyrosine, and threonine within affected areas. This targeted approach holds great promise for selectively reprogramming protein function and maintaining proteostasis, offering potential therapeutic strategies for various diseases.”

- Reference

54. Ghanem, S. S. et al. α -Synuclein phosphorylation at serine 129 occurs after initial protein deposition and inhibits seeded fibril formation and toxicity. *Proc.*

Natl. Acad. Sci. U. S. A. **119**, e2109617119 (2022).

55. Sugeno, N. et al. Serine 129 phosphorylation of alpha-synuclein induces unfolded protein response-mediated cell death. *J. Biol. Chem.* **283**, 23179-23188 (2008).
56. Oueslati, A., Paleologou, K. E., Schneider, B. L., Aebischer, P., Lashuel, H. A. Mimicking phosphorylation at serine 87 inhibits the aggregation of human α -synuclein and protects against its toxicity in a rat model of Parkinson's disease. *J. Neurosci.* **32**, 1536-1544 (2012).

Reviewer #2 (Remarks to the Author):

Response: *We thank the reviewer for their thoughtful and constructive previous review and attention to this revision.*

Reviewer #3 (Remarks to the Author):

The revised paper thoroughly addresses my previous feedback. I have no further comments.

Response: *Thank you very much for carefully assessing and supporting our work.*

Reviewer #4 (Remarks to the Author):

The authors have addressed the reviewer comments and have improved the manuscript considerably. The requested kinetic data have been determined by performing additional experiments. Some statements (see below) in the revised manuscript still need to be corrected and improved.

Response: *We truly appreciate your positive feedback and acknowledgment of the revisions made in response to your earlier comments. Thank you for your consideration and insightful comments on our work. Based on your constructive suggestions, we have meticulously made point-to-point responses and revised the manuscript. We believe that your comments have significantly improved the quality of our manuscript.*

Comments to Manuscript NCOMMS-23-36937-T REVISION

1) Page 3, Line 66: There are still some issues with the revised formulation “While small molecule asymmetric catalysts can selectively generate favorable intermediates for efficient catalytic reactions, their practical biomedical applications are hindered by challenges such as low solubility and off-target side effects.” Do the authors mean chiral organocatalysts with the term “small molecule asymmetric catalysts” as they cite reference 27, or do they also include under this term “metal complexes with chiral ligands”? The statement that their practical biomedical applications are hindered by challenges such as low solubility and off-target side effects is too general and not sufficiently detailed, as well-known chiral organocatalysts are very water-soluble, and an example of off-target side effects of chiral organocatalysts should be cited.

Response: *Thank you for your valuable suggestion. We referred to “small-molecule asymmetric catalysts” as small molecule catalysts with asymmetric catalysts, including chiral organocatalysts and metal complexes with chiral ligands. To avoid potential confusion, we have removed reference 27. Due to their small size, usually small molecules can rapidly diffuse through biological fluids, across many biological barriers, and through cell membranes. This enables them to navigate the complex vasculature and to interact with most tissues and cell types in the body (Physiological Pharmaceutics: Barriers to Drug Absorption (CRC Press, 2000); Nat. Biomed. Eng. 5, 951-967 (2021)), leading to off-target side effects and untargeted biodistribution. As far as we know, specific examples of off-target effects related to chiral organocatalysts are lacking because most of them are mainly used to synthesize compounds in vitro, and research on off-target side effects is primarily focused on clinical drugs or drug candidates undergoing clinical trials (references: Nat. Rev. Drug Discov. 10, 111-126*

(2011); *Toxicol. Pathol.* **41**, 310-314 (2013)). Given that off-target effects largely stem from the untargeted biodistribution of small molecules, it is reasonable to foresee similar outcomes with small molecule asymmetric catalysts in vivo. Leveraging nanomaterials' surface-modifiable characteristics, nanomaterials-based heterogeneous catalysts can be readily endowed with targeting capability through surface engineering, thereby overcoming certain inherent limitations associated with small molecules (references: *Angew. Chem. Int. Ed.* **50**, 1980-1994 (2011); *Adv. Mater.* **32**, 1902604 (2020)). In addition, according to your comments, the term "low solubility" has been removed in the revised manuscript for more accuracy, and more references regarding off-target side effects of small molecular drugs have been added in the revised manuscript.

Our modification to the manuscript: The following sentences were modified on page 4, and the relevant references were added in the revised manuscript, respectively.

- Page 4

".....While small-molecule asymmetric catalysts can selectively generate favorable intermediates for efficient catalytic reactions^{27,28}, their practical biomedical applications are hindered by inherent challenges associated with small molecules^{29,30}, such as off-target side effects³¹ and untargeted biodistribution^{29,32}."

- Reference

30. Washington, N., Washington, C., Wilson, C. *Physiological Pharmaceutics: Barriers to Drug Absorption* (CRC Press, 2000).

31. Rudmann, D. G. On-target and off-target-based toxicologic effects. *Toxicol. Pathol.* **41**, 310-314 (2013).

32. Veronese, F. M., Pasut, G. PEGylation, successful approach to drug delivery. *Drug Discov. Today* **10**, 1451-1458 (2005).

2) Page 4, Line 70: While in the previous sentence the term "asymmetric molecular catalysts" has been replaced with "small-molecule asymmetric catalysts", this statement that "... nanomaterials based heterogeneous catalysts show promise in overcoming the inherent limitations of molecular catalysts due to their modifiable surface" is not the appropriate comparison and should be reformulated.

Response: Thank you for your valuable suggestion. We have reformulated the phrase "... nanomaterials based heterogeneous catalysts show promise in overcoming the inherent limitations of molecular catalysts due to their modifiable surface" to "... nanomaterials based heterogeneous catalysts with asymmetric catalytic centers show promise in overcoming the inherent limitations of small-molecule asymmetric catalysts due to their modifiable surface" in the revised manuscript.

Our modification to the manuscript: *The following sentences were modified on page 4 in the revised manuscript.*

- Page 4

“.....In this regard, nanomaterials based heterogeneous catalysts with asymmetric catalytic centers show promise in overcoming the inherent limitations of small-molecule asymmetric catalysts due to their modifiable surface^{29, 33}.”